# BioDiscoveryAgent: An AI Agent for Designing Genetic Perturbation Experiments

**Yusuf Roohani** [*1,2]   **Andrew Lee**[*1]   **Qian Huang**[*1]   **Jian Vora**[1]
**Zachary Steinhart**[3,4]   **Kexin Huang**[1]   **Alexander Marson**[3,4,5,6]
**Percy Liang**[1]   **Jure Leskovec**[1]

[1]Department of Computer Science, Stanford University [2]Arc Institute
[3]Gladstone-UCSF Institute of Genomic Immunology
[4]Department of Medicine, University of California, San Francisco
[5]Department of Microbiology and Immunology, University of California, San Francisco
[6]UCSF Helen Diller Family Comprehensive Cancer Center

Correspondence to: yhr@cs.stanford.edu

## Abstract

Agents based on large language models have shown great potential in accelerating scientific discovery by leveraging their rich background knowledge and reasoning capabilities. In this paper, we introduce *BioDiscoveryAgent*, an agent that designs new experiments, reasons about their outcomes, and efficiently navigates the hypothesis space to reach desired solutions. We demonstrate our agent on the problem of designing genetic perturbation experiments, where the aim is to find a small subset out of many possible genes that, when perturbed, result in a specific phenotype (e.g., cell growth). Utilizing its biological knowledge, BioDiscoveryAgent can uniquely design new experiments without the need to train a machine learning model or explicitly design an acquisition function as in Bayesian optimization. Moreover, BioDiscoveryAgent using Claude 3.5 Sonnet achieves an average of 21% improvement in predicting relevant genetic perturbations across six datasets, and a 46% improvement in the harder task of non-essential gene perturbation, compared to existing Bayesian optimization baselines specifically trained for this task. Our evaluation includes one dataset that is unpublished, ensuring it is not part of the language model's training data. Additionally, BioDiscoveryAgent predicts gene combinations to perturb more than twice as accurately as a random baseline, a task so far not explored in the context of closed-loop experiment design. The agent also has access to tools for searching the biomedical literature, executing code to analyze biological datasets, and prompting another agent to critically evaluate its predictions. Overall, BioDiscoveryAgent is interpretable at every stage, representing an accessible new paradigm in the computational design of biological experiments with the potential to augment scientists' efficacy.

## 1 Introduction

Scientific discovery often resembles a search problem, where multiple hypotheses are tested to find one that leads to informative outcomes (Simon et al., 1981; Langley et al., 1987). AI agents, particularly those built using large language models (LLMs), show promising capabilities for this task (Wang et al., 2023). Previous studies have demonstrated that LLMs can effectively learn from the scientific literature (Wang et al., 2023; Boiko et al., 2023; Moor et al., 2023), compare various experimental plans (Liu et al., 2024; 2023; Huang et al., 2023b) and use tools such as executing code (Schick et al., 2024) or accessing databases (Jin et al., 2024).

---

[*]Equal contribution.
Code is available at: `www.github.com/snap-stanford/BioDiscoveryAgent`

Biomedical research, which heavily relies on iterative experimentation, stands to benefit significantly from such agents. One key challenge in this field is identifying drug targets—typically proteins that, when engaged by a drug, lead to a desired change in disease phenotype. Misidentification of these drug targets is a major cause of failure in clinical trials (Nelson et al., 2015). CRISPR-based genetic perturbation experiments are instrumental in addressing this challenge. These experiments involve the repression or activation of genes that code for proteins, followed by the measurement of the resulting biological effects, helping to identify drug targets that could reverse disease effects (Przybyla & Gilbert, 2022). These *perturbation screens* or *forward genetics screens* (Schneeberger, 2014; Moresco et al., 2013), have been transformative in areas such as drug target discovery (Wang & Doudna, 2023), elucidating disease mechanisms (Mamedov et al., 2023), cell engineering (Lim, 2022), gene therapy (Kalos et al., 2011), and immunotherapy (Goodman et al., 2022).

However, experimentally perturbing every single gene is costly. A perturbation screen typically perturbs around 19,000 protein-coding genes, yet anywhere between a handful of genes to a few thousand may exhibit the desired phenotype. When perturbing combinations of genes, this search space is even larger. By strategically designing these experiments in smaller batches that prioritize genes likely to result in meaningful phenotypic effects, it is possible to enhance the efficiency of the search process (King et al., 2004; Cleary et al., 2017; Huang et al., 2023a; Roohani et al., 2023).

Recent work has benchmarked Bayesian optimization algorithms for this task, but this requires training bespoke machine learning models that are often difficult to interpret, on small datasets (Mehrjou et al., 2021; Lyle et al., 2023) (Figure 1a). Moreover, these models are unable to leverage the vast biological prior knowledge contained within the scientific literature that is valuable during early experimentation. In contrast, LLMs have been exposed to the scientific literature and can recall biomedical knowledge (Gao et al., 2024). Such models have shown state of the art performance in information retrieval for clinical question answering (Moor et al., 2023), patient matching for clinical trials (Wornow et al., 2024), gene set identification (Hu et al., 2023), gene function prediction (Chen & Zou, 2023) and cell type annotation using gene expression data (Hou & Ji, 2024). However, LLM-based agents have not yet been used for closed-loop biological experiment design.

For effective experiment design, agents require both domain-specific knowledge and the ability to interpret and reason over experimental results. While LLMs demonstrate strong capabilities in these areas, their full applicability across diverse biological contexts requires access to not only the literature but also external sources like tabular datasets. A careful balance must be struck between granting the LLM freedom to explore the action space of genes to perturb and ensuring the selected genes are biologically valid and well-motivated. The LLM must carry information across successive prompts to maintain a consistent experimental strategy. Additionally, the agent's decision-making should be interpretable, ideally including literature citations and mechanisms for human feedback.

Here, we overcome some of these challenges and introduce *BioDiscoveryAgent*, an agent that designs genetic perturbation experiments using only an LLM paired with a suite of tools (Figure 1b). In each round, the agent constructs a prompt that includes both the task description and experimental results from previous rounds. This prompt is fed into the LLM, and the response identifies genes to perturb for the next round of experiments. BioDiscoveryAgent can also leverage different tools: it can search the scientific literature for relevant articles, execute code to analyze datasets that are inaccessible through textual sources and prompt another agent to critique the predictions of the initial agent.

BioDiscoveryAgent uniquely designs genetic perturbation experiments without relying on a specifically trained machine learning model or an explicitly defined acquisition function. It accomplishes this by leveraging an LLM that can effectively integrate information from both its prior knowledge and experimental results. Moreover, BioDiscoveryAgent using Claude 3.5 Sonnet outperforms baseline methods for experiment design, identifying 21% more experimental *hits* after five experimental rounds of 128 genes each, which is approximately 17 additional phenotypically relevant genes per dataset. On the harder task of only predicting non-essential genes, it identifies 46% more hits than baselines. BioDiscoveryAgent (Claude 3.5 Sonnet) also shows more than twice as high performance in predicting responses to combinatorial gene perturbations compared to a random baseline, exploring a new setting not previously considered. Unlike conventional approaches that are entirely black-box, the agent's decision-making is fully transparent at every stage and can be enhanced through using tools such as LLM-based model critique. Overall, BioDiscoveryAgent utilizes its vast biological knowledge along with the ability to reason over insights from previous experimental results to offer an accessible and interpretable method for designing genetic perturbation experiments.

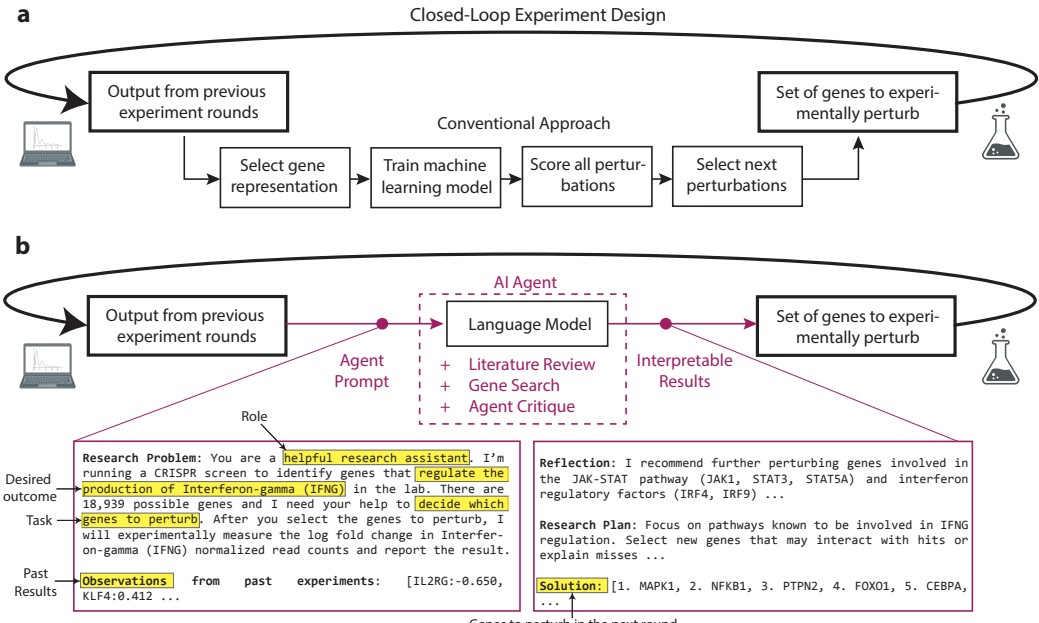

Figure 1: **An AI agent for closed-loop experiment design.** **(a)** Conventional Bayesian optimization approach for experiment design involves training a machine learning model in every experimental round, scoring all perturbations and defining an acquisition function for selecting genes to perturb in the next round. **(b)** Overview of BioDiscoveryAgent. In every round, the agent is given a prompt that describes the research problem, its role and task as well as experimental results from previous rounds. It generates a set of genes to perturb in the next round and provides reasoning for its prediction. The agent primarily makes use of a large language model for its predictions but also has access to additional tools such as the ability to search for relevant scientific papers on the internet.

## 2 PROBLEM FORMULATION

We study the following closed-loop experimental design task: an agent determines which genes to perturb in each experimental round, using the results to plan subsequent rounds. The objective is to maximize the number of *hit* genes or gene combinations that are detected over the full course of rounds. In particular, we test our model on two real-world experimental settings:

**Single-gene perturbation**: Let $\mathcal{G}$ represent the set of all protein-coding genes in humans, where $|\mathcal{G}| \approx 19{,}000$. The effect of perturbing a gene $g \in \mathcal{G}$, is denoted by a phenotypic response function $f(g)$, where $f : \mathcal{G} \to \mathbb{R}$ maps each gene to a real-valued phenotypic response. In this setting, we define an experimental round as the perturbation of a set of genes $\{g_1, g_2, \ldots, g_B\} \subseteq \mathcal{G}$. Each gene in this set is perturbed individually within separate cells. The goal of the experiment is to identify those genes that upon perturbation produce a desired phenotype, $f(g) > \tau$ for some threshold $\tau$. These genes are referred to as *hits* for that phenotype.

The goal for the agent is to guide the design of these experiments, such that over $t = 1, 2, \ldots, T$ experimental rounds, the total number of hits identified can be maximized (Mehrjou et al., 2021). The experiment design procedure involves selecting $b$ genes to perturb at each round $t$. Let $B_t$ refer to the set of genes selected at round $t$. Following this selection, the phenotypic response of perturbing each of these genes $f(g)$ for each $g \in B_t$ is revealed. In the next round $(t + 1)$, the agent has access to all phenotypic responses for genes tested in previous rounds: $1, 2, \ldots, t$.

At the end of $T$ rounds, the total hit ratio is computed as the fraction of true hits that were discovered cumulatively through the series of experiments (similar to recall). Let $\mathcal{G}_a = \cup_{t=1}^{T} B_t^+$, where $B_t^+ = \{g \in B_t : f(g) > \tau\}$ represents the hits identified in round $t$. Thus, $\mathcal{G}_a$ represents the cumulative set of hits across all rounds up to $t$. The hit ratio can then be formally expressed as hit ratio $= \frac{|\mathcal{G}_a|}{|\mathcal{G}_p|}$, where $\mathcal{G}_p$ is the set of all true hits for the phenotype, $\mathcal{G}_p = \{g \in \mathcal{G} : f(g) > \tau\}$.

**Two-gene perturbation**: We also consider a new problem setting of predicting gene pairs (2-gene combinations) to perturb. Here, each query consists of two genes perturbed in a single cell simultaneously. This formulation is both more computationally challenging given the larger search space as well as biologically impactful. Let $\mathcal{G}^2 = \mathcal{G} \times \mathcal{G}$ denote the set of all possible gene pairs that can be perturbed. Each element in $\mathcal{G}^2$ is a pair of genes $(g_a, g_b)$ where $g_a, g_b \in \mathcal{G}$. The perturbation experiment in this context is represented by a function $f_c : \mathcal{G}^2 \to \mathbb{R}$. This function $f_c$ maps each gene pair to a real-valued phenotypic response, which measures the combined effect of perturbing both genes simultaneously. The goal for the agent in this setting is to identify *pairs* of genes whose combined perturbation results in the desired phenotypic results, i.e. $f_c(g_a, g_b) > \tau$. Thus, *hits* also correspond to gene pairs instead of single genes.

## 3   BioDiscoveryAgent

We now present BioDiscoveryAgent which uses an LLM to automate the scientific discovery process for this biological setting. This includes accessing scientific knowledge, generating hypotheses, planning experiments and interpreting results (Algorithm 1). In the single gene setting, at each step $t$, the agent's objective is to select a batch of $B$ genes for testing in the next step. The agent receives a prompt that describes general information about the experimental setup and the biological hypothesis being tested (Figure 1b, Appendix A, B). The results from each experiment are incorporated into the next prompt, along with the same information about the experimental setup. This prompt creation draws from established methods in developing other LLM-based agents, such as pre-action reasoning (Yao et al., 2022), reflective thinking (Shinn et al., 2023b), and stepwise planning (AutoGPT, 2023).

We tested 9 different LLMs across varying levels of complexity for use in BioDiscoveryAgent (Claude v1 (Anthropic, 2023), Claude 3 Haiku, Claude 3 Sonnet, Claude 3 Opus (Anthropic, 2024b), Claude 3.5 Sonnet (Anthropic, 2024a), GPT-3.5-Turbo (OpenAI, 2023), GPT-4o (OpenAI, 2024a), o1-mini (OpenAI, 2024b), o1-preview (OpenAI, 2024c)). Due to the lack of transparent parameter counts from most organizations, we used the price per token as a proxy for model size. Unless otherwise noted, results are presented for the best performing high and low cost models which are Claude 3.5 Sonnet ($15 per 1M output token) and Claude 3 Haiku ($1.25 per 1M output token) respectively.

**Processing gene list and experimental observations**: When choosing genes for perturbation, it is not always feasible to include all possible genes to choose from. At the same time, we want to avoid arbitrarily narrowing down the gene list based on our existing knowledge. Therefore, we adopt a two-step approach: Initially, we allow BioDiscoveryAgent to suggest genes without restrictions (Appendix Figure 4a). This enables it to draw on its comprehensive understanding of biology freely. If the agent is unable to produce the required gene list after many trials due to invalid or repeated sampling of genes (Appendix Figure 4b,c), BioDiscoveryAgent summarizes the list of all remaining genes and adds it to the prompt to aid in gene selection (Appendix Figure 4d). The summarized gene list is designed to cover a broad range of biological pathways and functions. A similar summarization technique is used for the experimental observations when they exceed the LLM's context.

**Agent Response Format**: To ensure interpretability and to guide the agent's thought process, a consistent response format is defined across all prompts. We direct the LLM to structure its responses into several parts: `Reflection`, `Research Plan`, `Solution` (Appendix A, Figure 1b), similar to (Huang et al., 2023b). `Research Plan` helps in effective planning and monitoring progress. Through the `Reflection` and `Research Plan` entries, the model is able to provide additional reasoning behind a particular prediction. This also helps to rule out predictions that may be hallucinations or not well-motivated. `Solution` contains a formatted list of genes to perturb next.

### 3.1   Agent tools

The primary mechanism by which BioDiscoveryAgent interacts with the user is through natural language (Appendix Figure 5a). To aid its decision-making it has access to additional tools.

**Literature search**: As part of the input, the user can choose to have the agent query the literature at each experimental cycle to inform its predictions (Appendix Figure 5b,c). In this case, the agent uses the PubMed API (Wobben, 2020) to search for papers containing the most pertinent literature. The search terms are chosen by the agent. It then attaches the summarization of the paper to the prompt and use it to identify additional genes to perturb while retaining citations.

**Gene search based on biological databases**: We provide the agent with the ability to query databases to search for other genes with similar biological properties as hit genes from previous experimental rounds (Appendix Figure 5d). First, the API is called to perform enrichment analysis for biological processes on the Reactome 2022 database (Gillespie et al., 2022) to identify the most relevant biological pathways. For each identified pathway, the agent queries other genes that participate in the same pathway. The top K genes that appear most frequently in these pathways are then selected and concatenated to the prompt for the main agent. Gene search based on other criteria did not perform as well as Reactome (Appendix Table 5).

**AI critic**: To benefit from contrastive prompting strategies, we make use of an AI critic (Weng et al., 2023). The goal is to identify mistakes and enhance the quality of the final prediction made by the agent (Appendix Figure 5e). At every round, a critic agent (which is also an LLM) is prompted to critique the choice of the main agent i.e. it can change some or all the genes in the batch and come up with a new set of genes.

## 4 EXPERIMENTS

We assess model performance using data from past genetic perturbation experiments. We simulate the perturbation of a gene $g$ by retrieving the relevant observation of the perturbation-induced phenotype $f(g)$ from this dataset. In every experimental round we perturb 128 genes, representing a reasonably sized small-scale biological screen. Since each round of experimentation can incur additional costs and introduce unwanted experimental variation, we focus our evaluations on fewer experimental rounds (5) to more accurately reflect a real biological setting. For each dataset, after each round, we calculate the hit ratio as the proportion of discovered hits out of the total true hits for that dataset.

### 4.1 DATASETS AND BASELINES

For the single-gene perturbation setting, we make use of six different datasets spread across different cell types, publication dates and data generation sites. Each of the datasets contains the phenotypic response of knocking-down over 18,000 individual genes in distinct cells, with the exception of Scharenberg et al. (2023) which contains data for 1061 perturbations. All datasets were released after 2021, apart from one dataset (CAR-T [1]) which is so far unpublished.

Each of the datasets measure a distinct biological process. The Schmidt et al. (2022) dataset measures the changes in the production of two key cytokines involved in immune signaling: Interferon-$\gamma$ (IFNG) and Interleukin-2 (IL-2) under different genetic perturbations performed in primary human T-cells. The Carnevale et al. (2022) dataset includes perturbation screens for identifying genes that render T cells resistant to inhibitory signals encountered in the tumor microenvironment. Unpublished data (CAR-T dataset) studies the impact of genome-wide perturbations on CAR-T cell proliferation. The Scharenberg et al. (2023) dataset measures the effect of perturbation on mediating lysosomal choline recycling in pancreatic cells, and the Sanchez et al. (2021) dataset studies the change in expression of endogenous tau protein levels in neurons.

For the two-gene perturbation task, we use a dataset from a screen that knocked down 100,576 gene pairs in K562 cells (Horlbeck et al., 2018). For each gene pair, we are interested in the synergistic effects on cell fitness upon combinatorial knockdown. Synergy is determined by the deviation between the observed cell fitness and the expected fitness, which is calculated by summing the average impact of knocking down each gene in the combination individually (Horlbeck et al., 2018). For baseline models, we use a multi-layer perceptron combined with one of seven different Bayesian optimization acquisition functions (Mehrjou et al., 2021; Lyle et al., 2023). We also include a human baseline (Appendix E).

## 5 RESULTS

**BioDiscoveryAgent outperforms baselines based on hit ratio for 1-gene perturbation experiments**: We evaluate the performance of BioDiscoveryAgent without any tools (*No-Tools*) against a range of baselines, including random sampling. When measuring hit ratios across all genes at

---

[1]generated by the authors of this paper

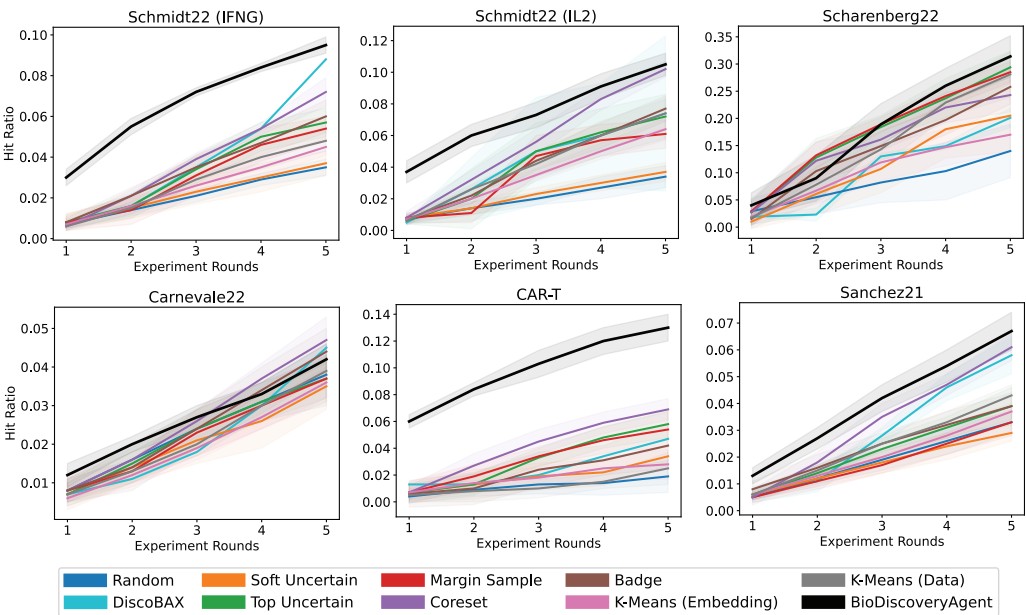

Figure 2: **Per-round performance comparison to machine learning baselines for 1-gene perturbation experiments**: Each line averages the hit ratio from 10 runs, with error bars indicating 1 standard deviation. 128 genes were predicted per round over 5 rounds. For Scharenberg et al. (2023), a batch size of 32 was used due to its smaller size of 1061 perturbations.

experimental round 5, BioDiscoveryAgent consistently demonstrates superior performance compared to the best baseline approach, across 7 out of 9 LLMs tested (Table 1, Appendix Table 7). The best performing LLM, Claude 3.5 Sonnet (Appendix Table 8), outperforms the best baseline for each dataset by 21% on average. Performance improvement is observed across 5 of the 6 datasets. This gap is especially large at earlier rounds, where the LLM can leverage its biological knowledge to select genes, in contrast to baseline methods that suffer from the cold start problem (Figure 2). However, it persists into much longer runs as well, when considering non-essential gene hits over 30 rounds of experimentation (Appendix Tables 12, 13, 14, 15). We also tested the scenario (Sonnet + Coreset) where the best performing baseline (Coreset) uses predictions from BioDiscoveryAgent (Sonnet) in its first round. Results indicate that while the agent adds non-redundant information that consistently improves performnace over the baseline, the combination does not outperform the agent by itself.

To verify that the model isn't relying on uninformative hits, we measured model performance in predicting hits that are non-essential genes. Essential genes are likely to be detected as hits under any perturbation screen given the strong phenotypic effect of perturbing these genes. On the other hand, the response of non-essential genes to perturbation, is harder to predict and often more useful biologically. When filtering for non-essential genes, BioDiscoveryAgent (Claude 3.5 Sonnet) shows an improvement across all 6 datasets with an even higher average performance improvement of 46% over baselines (Table 1, Appendix Table 7).

Additionally, we tested the model's tendency to predict distinct genes when prompted differently and upon observing different experimental results. We observe a low Jaccard similarity between all predicted genes after five rounds of experiments for any pair of datasets (Appendix Figure 6), suggesting that gene selection is not invariant to the task prompt and experimental observations.

**BioDiscoveryAgent can guide 2-gene combinatorial perturbation experiments**: In addition to 1-gene perturbation experiments, we also demonstrate that BioDiscoveryAgent can guide 2-gene combinatorial perturbation experiments, which is significantly more difficult due to the much larger combinatorial search space (100,576 gene pairs considered in Horlbeck et al. (2018)). As shown in Table 2, BioDiscoveryAgent (Claude 3.5 Sonnet) significantly outperforms the random sampling baseline by 170% on average.

| **Model** | Schmidt1 | | Schmidt2 | | CAR-T[†] | | Scharen.[*] | | Carnev. | | Sanchez | |
|---|---|---|---|---|---|---|---|---|---|---|---|---|
| | All | N/E | All | N/E | All | N/E | All | N/E | All | N/E | All | N/E |
| Random | 0.037 | 0.035 | 0.031 | 0.031 | 0.033 | 0.020 | 0.160 | 0.136 | 0.036 | 0.039 | 0.034 | 0.034 |
| Human | 0.032 | 0.036 | 0.029 | 0.034 | 0.033 | 0.034 | 0.113 | 0.122 | 0.042 | **0.049** | 0.025 | 0.026 |
| **Baseline Models** | | | | | | | | | | | | |
| Soft Uncertain | 0.037 | 0.037 | 0.037 | 0.035 | 0.034 | 0.032 | 0.205 | 0.161 | 0.031 | 0.035 | 0.029 | 0.029 |
| Top Uncertain | 0.057 | 0.042 | 0.072 | 0.050 | 0.058 | 0.044 | 0.294 | 0.236 | 0.037 | 0.033 | 0.039 | 0.028 |
| Margin Sample | 0.054 | 0.040 | 0.061 | 0.047 | 0.054 | 0.045 | 0.285 | 0.227 | 0.036 | 0.032 | 0.033 | 0.028 |
| Coreset | 0.072 | 0.066 | 0.102 | 0.084 | 0.069 | 0.059 | 0.243 | 0.197 | **0.047** | 0.038 | 0.061 | 0.054 |
| Badge | 0.060 | 0.050 | 0.077 | 0.058 | 0.042 | 0.038 | 0.258 | 0.211 | 0.044 | 0.036 | 0.039 | 0.035 |
| K-Means (E) | 0.045 | 0.035 | 0.064 | 0.053 | 0.028 | 0.024 | 0.170 | 0.154 | 0.036 | 0.029 | 0.037 | 0.032 |
| K-Means (D) | 0.048 | 0.035 | 0.074 | 0.060 | 0.025 | 0.021 | 0.281 | 0.240 | 0.039 | 0.030 | 0.043 | 0.037 |
| DiscoBax | 0.088 | 0.069 | 0.074 | 0.057 | 0.047 | 0.021 | 0.201 | 0.200 | 0.045 | 0.038 | 0.058 | 0.049 |
| **BioDiscoveryAgent** (No-Tools) | | | | | | | | | | | | |
| Claude 3 Haiku | 0.064 | 0.072 | 0.072 | 0.083 | 0.116 | 0.119 | 0.209 | 0.200 | 0.032 | 0.034 | 0.056 | 0.056 |
| GPT-3.5-Turbo | 0.044 | 0.048 | 0.061 | 0.073 | 0.064 | 0.066 | 0.230 | 0.188 | 0.032 | 0.034 | 0.039 | 0.038 |
| Claude v1 | 0.067 | 0.086 | 0.089 | 0.110 | 0.110 | 0.126 | 0.292 | 0.222 | 0.038 | 0.045 | 0.053 | 0.055 |
| o1-mini | 0.081 | 0.092 | 0.096 | 0.120 | 0.127 | 0.139 | 0.279 | 0.286 | 0.040 | 0.041 | **0.074** | **0.073** |
| Claude 3 Sonnet | 0.076 | 0.082 | 0.088 | 0.111 | 0.115 | 0.118 | 0.302 | 0.265 | 0.041 | 0.042 | 0.064 | 0.063 |
| Claude 3.5 Sonnet | **0.095** | **0.107** | 0.104 | 0.122 | 0.130 | 0.133 | **0.326** | **0.292** | 0.042 | 0.044 | 0.066 | 0.063 |
| GPT-4o | 0.059 | 0.068 | 0.074 | 0.090 | 0.116 | 0.120 | 0.311 | 0.286 | 0.037 | 0.040 | 0.030 | 0.043 |
| o1-preview | 0.081 | 0.091 | 0.091 | 0.114 | **0.141** | **0.145** | 0.283 | 0.259 | 0.041 | 0.043 | 0.068 | 0.063 |
| Claude 3 Opus | 0.094 | 0.106 | 0.104 | **0.132** | 0.126 | 0.129 | 0.247 | 0.206 | 0.043 | 0.043 | 0.059 | 0.058 |
| Sonnet + Coreset | 0.092 | 0.079 | **0.133** | 0.092 | 0.111 | 0.101 | 0.215 | 0.160 | 0.047 | 0.038 | 0.063 | 0.054 |

Table 1: **Performance comparison to machine learning baselines for 1-gene perturbation experiments**. Hit ratio for experimental round 5 averaged over 10 runs, with 128 genes predicted in each round. 'All' refers to results across all genes, 'N/E' only considers non-essential genes. The best performing model in each class is highlighted in gray. [*]For Scharenberg et al. (2023), a batch size of 32 was used due to its smaller size of 1061 perturbations. Schmidt1 refers to the screen measuring Interferon-$\gamma$ (IFNG) and Schmidt2 measures Interleukin-2 (IL-2) following perturbation (Schmidt et al., 2022). [†]CAR-T refers to an unpublished dataset. LLMs are sorted by price per 1M output tokens (Table 4). See Appendix Table 7 for error intervals.

| Model | Round 1 | Round 2 | Round 3 | Round 4 | Round 5 |
|---|---|---|---|---|---|
| Random | $2.6 \pm 1.43$ | $5.7 \pm 2.83$ | $8.9 \pm 3.67$ | $12.8 \pm 3.74$ | $16.4 \pm 3.8$ |
| **BioDiscoveryAgent** (No-Tools) | | | | | |
| Claude 3 Haiku | $5.1 \pm 2.2$ | $14.8 \pm 2.9$ | **$23.6 \pm 4.1$** | **$34.1 \pm 5.8$** | **$45.1 \pm 5.3$** |
| Claude 3.5 Sonnet | **$8.4 \pm 1.36$** | **$15.7 \pm 1.1$** | $23.1 \pm 2.59$ | $31.7 \pm 2.9$ | $40.0 \pm 3.77$ |

Table 2: **Performance evaluation for 2-gene perturbation experiments.** 32 gene pairs out of 100,576 pairs predicted per round averaged over 10 runs (Horlbeck et al., 2018). Evaluation using cumulative number of hits across 5 experimental rounds. Error intervals correspond to 1SD.

**The performance of BioDiscoveryAgent can be augmented with tool-use, but the improvement varies by choice of LLM**: We evaluated the impact of integrating three different tools—literature search, gene search, and AI critic into BioDiscoveryAgent (Table 3, Appendix Table 9). For the agent using Claude 3.5 Sonnet, tools applied individually or in combination did not lead to significant changes in performance and in some cases hurt performance. In contrast, for Claude 3 Haiku, incorporating all tools consistently enhanced performance.

The effects of different tools varied. In the case of Claude 3 Haiku, using only the literature search tool often resulted in fixation on a few simple keywords and irrelevant papers, leading to less effective searches. However, the literature search still provided verifiable citations for the gene prediction process, which is beneficial for scientists. Performance significantly improved with the use of a gene search based on shared biological pathways. This improvement likely stems from the gene search relying on tabular datasets not fully represented in text-based sources used for language model

| Tools Used | Schmidt1 | Schmidt2 | CAR-T | Scharen. | Carnev. | Sanchez |
|---|---|---|---|---|---|---|
| Random | 0.037 | 0.031 | 0.033 | 0.160 | 0.036 | 0.034 |
| **BioDiscoveryAgent (Claude 3.5 Sonnet)** (Does not benefit from these tools) | | | | | | |
| No-Tools | 0.095 | **0.104** | 0.130 | 0.326 | 0.042 | 0.066 |
| Literature | **0.096** (+1%) | 0.098 (-6%) | **0.138** (+6%) | 0.309 (-5%) | 0.042 (+0%) | **0.069** (+5%) |
| AI Critic | 0.088 (-7%) | 0.092 (-12%) | 0.126 (-3%) | 0.309 (-5%) | 0.042 (+0%) | 0.059 (-11%) |
| Gene Search | **0.096** (+1%) | 0.100 (-4%) | 0.123 (-5%) | **0.348** (+7%) | **0.043** (+2%) | 0.062 (-6%) |
| All-Tools | **0.096** (+1%) | 0.090 (-13%) | 0.121 (-6%) | 0.234 (-28%) | **0.043** (+2%) | 0.054 (-18%) |
| **BioDiscoveryAgent (Claude 3 Haiku)** (Benefits from these tools) | | | | | | |
| No-Tools | 0.064 | 0.072 | 0.116 | 0.209 | 0.032 | 0.056 |
| Literature | 0.053 (-17%) | 0.069 (-4%) | 0.091 (-22%) | 0.164 (-22%) | 0.035 (+9%) | 0.057 (+2%) |
| AI Critic | 0.061 (-5%) | 0.070 (-3%) | 0.113 (-3%) | 0.219 (+5%) | 0.043 (+34%) | 0.054 (-4%) |
| Gene Search | 0.080 (+25%) | 0.098 (+36%) | 0.114 (-2%) | 0.249 (+14%) | **0.046** (+44%) | **0.065** (+16%) |
| All-Tools | **0.084** (+31%) | **0.099** (+38%) | **0.128** (+10%) | **0.259** (+24%) | 0.043 (+34%) | 0.058 (+4%) |

Table 3: **Agent performance improvements when using different tools for 1-gene perturbation experiments**. Results show hit ratio for experimental round 5 averaged over 10 runs. See Table 1 caption for notes on specific datasets. See Appendix Table 9 for error intervals.

| Tools Used | Schmidt1 | Schmidt2 | CAR-T | Scharen. | Carnev. | Sanchez | Price per 1M Token ($) |
|---|---|---|---|---|---|---|---|
| Claude 3 Haiku | +31% | +38% | +10% | +24% | +34% | +4% | 1.25 |
| GPT-3.5-Turbo | +41% | +64% | -2% | -5% | +16% | +15% | 3.00 |
| Claude v1 | +42% | +37% | +4% | +14% | +42% | +9% | 11.00 |
| o1-mini | +2% | -22% | -10% | -5% | 0% | -3% | 12.00* |
| Claude 3 Sonnet | -3% | +3% | -9% | 0% | +15% | +14% | 15.00 |
| Claude 3.5 Sonnet | +1% | -13% | -6% | -28% | +2% | -18% | 15.00 |
| GPT-4o | -17% | -14% | -11% | -8% | -5% | -13% | 15.00 |
| o1-preview | -36% | -13% | -30% | +2% | -41% | -54% | 60.00* |
| Claude 3 Opus | +2% | -3% | -8% | +18% | -5% | -7% | 75.00 |

Table 4: **Effect of tools on different LLMs.** Results show increase in hit ratio when using all-tools as compared to results for the same agent when using no-tools. Models are sorted by increasing price per 1M output tokens. *For o1-mini and o1-preview, output tokens include internal reasoning tokens generated by the models that are not visible in API responses (OpenAI, 2024b). Results are shown for experimental round 5 averaged over 10 runs. See Appendix Table 10 for the actual hit rates.

training, which the *No-Tools* agent cannot access. Using only the AI critic tool slightly improved performance over the No-Tools model by diversifying predictions or focusing on specific gene sets.

We further evaluated the impact of incorporating tools across a broader range of LLMs (Table 4, Appendix Table 10). We observe that while some models such as Claude v1, Claude 3 Haiku and GPT3.5-Turbo showed a significant improvement in performance with the incorporation of all tools (24.7%, 23.5% and 21.5% respectively), others such as Claude 3.5 Sonnet, Claude 3.5 Opus and GPT4o showed a decrease in performance or no significant change (-10%, -1% and -11% respectively). This contrast suggests a potential relationship between model size and the benefit of tool use—smaller models appear to benefit more from tool-assisted retrieval, whereas larger models may derive this information directly from their trained weights. Another reason could be the difference in the level of prediction confidence between smaller and larger models (Wu et al., 2024).

We further investigated this relationship by measuring the percentage of new genes predicted by Claude 3 Haiku only when using tools, that are also predicted by Claude 3.5 Sonnet with no-tools (Appendix Figure 7). For some datasets, we see a large proportion of such tool-derived genes being predicted by Claude 3.5 Sonnet without any tool use (14-28%). This suggests that larger models are capable of intrinsically retrieving additional biological information that is otherwise provided by the tools. However, this wasn't the case in all datasets indicating that some orthogonal information from the tools remains uncaptured by training on scientific text alone. Therefore, a more strategic design and selection of tools could lead to further improvements in agent performance for larger models.

**BioDiscoveryAgent accounts for prior knowledge and observations in decision-making** : Next, we investigate the use of prior knowledge versus observations from previous experiments in the agent's decision-making. We examine three scenarios using BioDiscoveryAgent (Claude 3.5 Sonnet): 1) *Prompt + Observation*, where the agent utilizes both previous experiment results and detailed information about the experiment's goal; 2) *Prompt Only*, where the agent ignores all experiment results; 3) *Observation Only*, where the agent is unaware of the current experiment's goal and only conditioned on observations. To accurately capture these trends at fine resolution, we use a larger number of experimental rounds with smaller number of perturbations in each round: 30 rounds of experimentation with 32 genes in each round on the IFNG dataset from Schmidt et al. (2022).

Results show that *Prompt + Observation* outperforms the other two scenarios, highlighting the significance of integrating prior knowledge and observations (Figure 3a). Interestingly, *Prompt + Observation* and *Prompt Only* benefit from prior knowledge early on, unlike *Observation Only*, which lacks the experiment's goal, underscoring the vital role of prior knowledge in the initial experiment phases. However, as experiments progress, *Observation Only* surpasses *Prompt Only*, showcasing the agent's capacity to adapt based on observations. For further validation, we also perform the same experiment using a different LLM and observe a similar trend (Appendix Figure 9, Table 16)

Additionally, we find that access to observations results in more similar gene predictions across different trials compared to experiments without access to observations (Figure 3b). This consistency was quantified using the Jaccard similarity index between all predicted genes after 30 rounds of experimentation. This suggests that observations significantly influence BioDiscoveryAgent's decision-making, leading to more uniform choices across separate trials.

**BioDiscoveryAgent provides interpretable predictions with references to the literature.** BioDiscoveryAgent provides interpretable predictions at various stages. In one such example (Appendix G, Figure 5), the agent is tasked with identifying genes regulating the production of Interferon-gamma (IFNG). The agent explicitly reasons that it will focus on genes involved in mitochondrial respiration and the electron transport chain (Appendix Figure 5c). Utilizing the literature search tool, it accesses relevant literature to support its predictions, citing specific papers and line numbers (Appendix Figure 5b). For example, the agent highlights *STUB1* as a gene for potential perturbation, supported by references to specific lines in a relevant scientific paper. Alternatively, the agent is also able to indicate when the paper being summarized does not contain actionable information: "In this segment, I cannot find specific gene targets or pathways to focus on for the initial experiments" (Appendix G).

The LLM critic tool provides valuable insights into the predicted set of genes, identifying potential issues such as randomness or too narrow of a focus. In this example, the critic LLM suggests diversifying the selected genes by also considering pathways involved in interferon regulation, such as NF-kB signaling and MAPK signaling (Appendix Figure 5e). Thus, not only does the critic LLM provide clear reasons for selecting specific genes but also further opens avenues for human-in-the-loop feedback by a subject-matter expert. In another example, the critic LLM guides the agent to limit the randomness in gene selection: "The selection of genes seems somewhat random and not focused enough on likely candidates based on known lysosomal and endolysosomal genes..." (Appendix G).

## 6 RELATED WORK

Artificial intelligence has demonstrated significant potential across various scientific domains, from realistic simulations of human behavior (Park et al., 2023b) to exploring mathematical function spaces (Romera-Paredes et al., 2023). Research has shown the utility of these models in mining and querying the scientific literature (Lála et al., 2023; Schick et al., 2023), as well as in general research tasks such as analyzing large datasets, reasoning about data, and generating reports (Shakked & Zhang, 2023; Lehr et al., 2024). Additionally, closed-loop AI-driven lab experimentation has made notable advances, particularly in chemical synthesis (Boiko et al., 2023; M. Bran et al., 2024) and materials discovery (Tom et al., 2024).

In the biological domain, research has shown that LLMs can capture significant information about biological pathways and processes (Park et al., 2023a), and are valuable in simulating biological processes at various scales (Schaefer et al., 2023). There have been benchmarking efforts for answering questions in genomics (Hou & Ji, 2023), performing bioinformatic analyses (Sarwal et al., 2023) as well as assessing broader biological research capabilities (Laurent et al., 2024). Despite long-

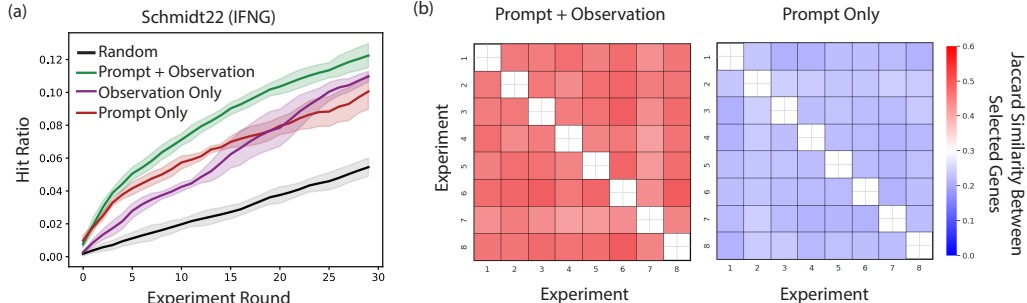

Figure 3: **Agent accounts for both prior knowledge and observations in decision-making** Three scenarios are considered: BioDiscoveryAgent (Claude 3.5 Sonnet) has access to task description and experimental observations (Prompt + Observation); only has access to experimental observations (Observation Only); only has access to the task description (Prompt Only). **(a)** Hit ratio at each experimental round across 30 rounds with 32 genes predicted in each. Each line corresponds to the average over 8 runs with error bars representing 1SD. **(b)** Jaccard similarity index between all predicted genes at 30 rounds of experimentation. Each cell corresponds to a different model run.

standing interest in developing autonomous AI systems for genetic perturbation experiment design (King et al., 2004), the use of LLM-based agents has yet to be explored for closed-loop biological experiment design. Some work has focused on designing individual gene editing experiments (Huang et al., 2024), primarily by optimizing experimental parameters and protocols. In contrast, our work, BioDiscoveryAgent, aims to generate and refine biological hypotheses through a series of perturbation experiments, leveraging the agent's biological understanding and reasoning capabilities. Although agents have not been used in this setting, past research has investigated machine learning and Bayesian optimization techniques for similar purposes (Mehrjou et al., 2021; Lyle et al., 2023).

# 7 DISCUSSION

BioDiscoveryAgent represents a new paradigm in the design of biological experiments, aimed at augmenting scientists' capabilities. Traditionally, this process employs a multi-stage pipeline using an acquisition function combined with a machine learning model. This model, often challenging to interpret, requires manual design and periodic retraining with handpicked gene features. In contrast, we demonstrate that an agent powered by an LLM can simplify the entire experimental design process into a single prompt from the researcher. Through its enhanced interpretability at every stage, the researcher can meaningfully engage with the model and augment its predictions.

Moreover, the agent is equipped with prior biological knowledge, solving the cold start problem at the beginning of a new round of experiments while efficiently utilizing observational data as the rounds progress. By using various tools, it can explicitly search for and integrate information from the scientific literature and existing tabular datasets. This capability results in a research assistant capable of speeding up biological research through utilizing information from diverse sources.

While BioDiscoveryAgent shows strong capabilities in enhancing experiment design, it still has room for improvement. It tends to perform better with certain cell types over others, likely due to variations in scientific literature coverage. Most performance benefits are observed in the early stages of experimentation rather than in prolonged rounds. Additionally, there is a need to develop better tools or fine-tuning methods to improve information extraction from non-text-based sources.

Overall, BioDiscoveryAgent offers a complementary approach to existing experimental design methods, significantly improving model performance in the low data regime while enhancing overall model interpretability. Amid increasing interest in active experimental design for biological research, BioDiscoveryAgent provides compelling evidence that language models could become essential components of such systems. By facilitating improved reasoning and interpretability and leveraging a broad understanding of the scientific literature, AI tools like BioDiscoveryAgent are poised to become integral to experimental design strategies in the near future.

## ETHICS STATEMENT

This study did not involve any animal or human subjects. All experiments were conducted using cell cultures under controlled laboratory conditions.

## REPRODUCIBILITY STATEMENT

Our results are fully reproducible with the code provided. The required prompts are included in the repository, and all datasets are available in the supplementary materials. The only exception is the CAR-T dataset since it is so far unpublished. Authors are willing to provide an anonymized version of that dataset upon request.

## ACKNOWLEDGEMENTS

We thank Yanay Rosen, Hamed Nilforoshan, Charilaos Kanatsoulis, Rok Sosic, Hanchen Wang, Ayush Agrawal, Charlotte Bunne, Michael Moor, Michael Bereket, Yash Dalmia, Jens Magnusson, Julia Carnevale, Mineto Ota, Ralf Schmidt, Carl Ward and attendees of the ICLR Workshop on Machine Learning for Genomics Explorations, 2024 for discussions and for providing feedback on our manuscript. J.L. gratefully acknowledge the support of DARPA under Nos. N660011924033 (MCS); NSF under Nos. OAC-1835598 (CINES), CCF-1918940 (Expeditions), DMS-2327709 (IHBEM); Stanford Data Applications Initiative, Wu Tsai Neurosciences Institute, Stanford Institute for Human-Centered AI, Chan Zuckerberg Initiative, Amazon, Genentech, GSK, Hitachi, SAP, and UCB. The content is solely the responsibility of the authors and does not necessarily represent the official views of the funding entities.

## CONFLICTS OF INTEREST

A.M. is a cofounder of Site Tx, Arsenal Biosciences, Spotlight Therapeutics and Survey Genomics; serves on the boards of directors at Site Tx, Spotlight Therapeutics and Survey Genomics; is a member of the scientific advisory boards of Site Tx, Arsenal Biosciences, Cellanome, Spotlight Therapeutics, Survey Genomics, NewLimit, Amgen and Tenaya; owns stock in Arsenal Biosciences, Site Tx, Cellanome, Spotlight Therapeutics, NewLimit, Survey Genomics, Tenaya and Lightcast; has received fees from Site Tx, Arsenal Biosciences, Cellanome, Spotlight Therapeutics, NewLimit, Gilead, Pfizer, 23andMe, PACT Pharma, Juno Therapeutics, Tenaya, Lightcast, Trizell, Vertex, Merck, Amgen, Genentech, GLG, ClearView Healthcare, AlphaSights, Rupert Case Management, Bernstein and ALDA; is an investor in and informal advisor to Offline Ventures; and a client of EPIQ. The Marson laboratory has received research support from the Parker Institute for Cancer Immunotherapy, the Emerson Collective, Arc Institute, Juno Therapeutics, Epinomics, Sanofi, GlaxoSmithKline, Gilead and Anthem and reagents from Genscript and Illumina. The Krogan Laboratory has received research support from Vir Biotechnology, F. Hoffmann-La Roche and Rezo Therapeutics. N.J.K. has a financially compensated consulting agreement with Maze Therapeutics. N.J.K. is the President and on the Board of Directors of Rezo Therapeutics; and is a shareholder in Tenaya Therapeutics, Maze Therapeutics, Rezo Therapeutics, GEn1E Lifesciences and Interline Therapeutics. J.W.F. was a consultant for NewLimit; is an employee of Genentech; and has equity in Roche. A.T.S. is a founder of Immunai, Cartography Biosciences, Santa Ana Bio and Prox Biosciences; is an advisor to Zafrens and Wing Venture Capital; and receives research funding from Astellas and Merck Research Laboratories. Patent applications have been filed based on the findings described here. The other authors declare no competing interests.

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

## A  PROMPT

The prompt includes the task information and response format as this example shown below.

```
You are a scientist working on problems in drug discovery.

Research Problem: I'm planning to run a CRISPR screen to
    identify genes that regulate the production of
    Interleukin-2 (IL-2). There are 18,939 possible  genes to
     perturb and I can only perturb 128 genes at a time. For
    each perturbation, I'm able to measure out the log fold
    change in Interleukin-2 (IL-2) normalized read counts
    which will be referred to as the score. I can only do a
    few rounds of experimentation.

Always respond in this format exactly:

1. Reflection: Thoughts on previous results and next steps.
2. Research Plan: The full high level research plan, with
    current status and reasoning behind each proposed
    approach. It should be at most 5 sentences.
3. Solution: Propose a list of predicted genes to test
    separated by commas in this format: 1. <Gene name 1>, 2.
    <Gene name 2> ...
Do not include any genes from this prompt (since they're
    already tested).
```

## B  DATASET SPECIFIC PROMPTS

For each dataset, the research problem and the type of measurement outcome are used to create the prompt.

```
IFNG Task: identify genes that regulate the production of
            Interferon-gamma (IFNG)
IFNG Measurement: the log fold change in Interferon-gamma (IFNG)
            normalized read counts

IL2 Task: identify genes that regulate the production of
            Interleukin-2 (IL-2)
IL2 Measurement: the log fold change in Interleukin-2 (IL-2)
            normalized read counts

CAR-T Task: identify genes that upon inhibition allow
            cells to resist T-cell
            exhaustion, under the HA GD2 CAR
            (chimeric-antigenic receptor) condition
CAR-T Measurement: the log fold change in normalized
            sgRNA read counts
            compared to the non-targeting control,
            22 days after perturbation

Scharenberg Task: identify genes mediating lysosomal choline
            recycling using an
            endolysosome-focused CRISPR-Cas9 screen
Scharenberg Measurement: enrichment or depletion of targeting
            sgRNAs (indicated by a high score) in the
            culture medium lacking free choline

Carnevale Task: identify genes that, upon being knocked out,
            would boost the efficacy of engineered T cells in
            the presence of an adenosine agonist that
            creates an immunosuppressive condition
Carnevale Measurement: the change in T cell proliferation
```

## C  ALGORITHM FOR BIODISCOVERYAGENT

---

**Algorithm 1** BioDiscoveryAgent: AI Agent for Biological Experiment Design (using all tools)

---

**Input:** Experiment description, Number of rounds $T$, Number of genes to perturb in each round $b$
**Output:** Set of genes to perturb
**for** $t = 1$ to $T$ **do**
    Search and retrieve literature using LLM-generated search terms
    Summarize articles using LLM and attach results to the main gene selection prompt
    Perform LLM-requested gene search and attach results to the main gene selection prompt
    Prompt LLM to select $b$ new genes (or gene pairs)
    **Output**: LLM generates a structured response with `Reflection`, `Research Plan`, `Gene Search`, and `Solution` entries
    **while** any predicted genes are invalid **do**
        Prompt LLM to select new genes
    **end while**
    Prompt LLM to critique the prediction made by the main agent
    Get phenotypic score $f(g)$ for each gene $g$ and add to gene selection prompt for the next step
    **if** context window is too large **then**
        Prompt LLM to summarize text for context window management
    **end if**
**end for**

---

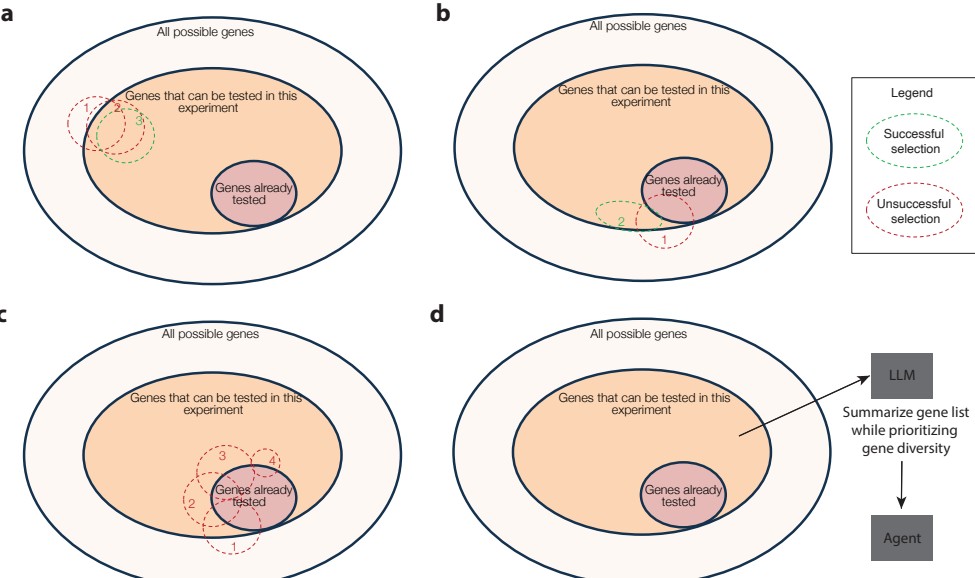

Figure 4: **Gene selection strategy**: **(a)** The space of genes that can be tested in a given experiment is constrained by expeirmental limitations. BioDiscoveryAgent can take a few tries to select genes within this limited space. **(b)** A common error is repeating previously tested genes. **(c)** Often this will result in the agent getting stalled and unable to make successful selections, especially in the case of large batch sizes. **(d)** After several failed attempts, we summarize the space of genes that can be tested while prioritizing gene diversity.

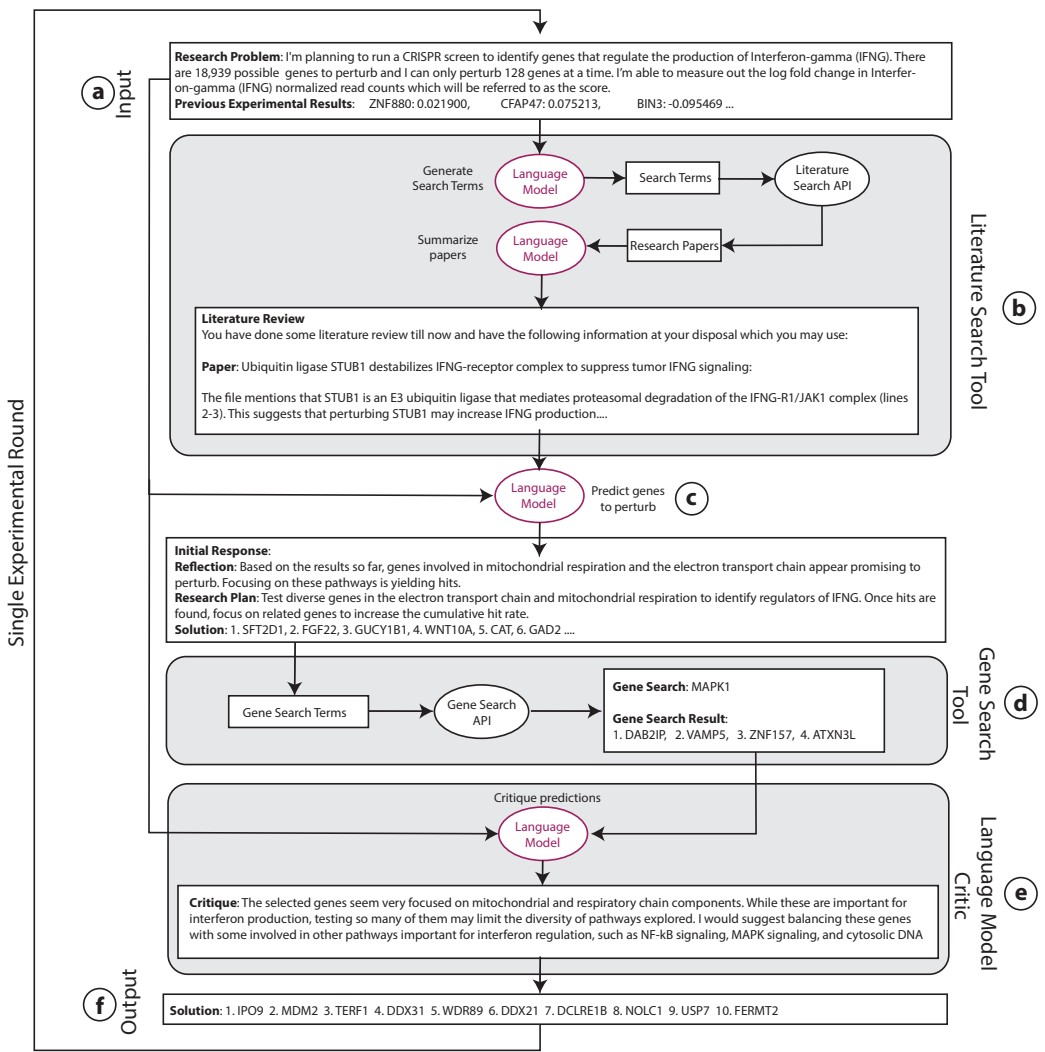

Figure 5: **BioDiscoveryAgent workflow with all tools over a single experimental round**. Prompts and agent responses have been summarized. See Appendix G for full trace. **(a)** The input to the agent is the description of the problem. **(b)** In case of the literature search tool, the LLM first determines appropriate search terms for finding relevant scientific papers. Top ranked retrieved papers are summarized by the LLM (along with line numbers that reference directly to text in the paper) and provided as additional context to the agent when predicting the set of genes to perturb.**(c)** The agent predicts the set of genes to perturb in the next experiment. Response is structured containing Reflection, Research Plan and Solution. **(d)** In case of using the gene search tool, the LLM identifies a query gene with which to search for other genes **(e)** The LLM can also be prompted differently to function as a critic and analyze the predictions so far from a different perspective **(f)** Final output gene list after using different tools. We used Claude v1 for this analysis since it showed high absolute score as well as high performance gain through tool use for this dataset (Schmidt et al., 2022).

## D   TOOL DESCRIPTIONS FOR BIODISCOVERYAGENT

In this section, we provide some more details about the tools provided to BioDiscoveryAgent to aid it in making its predictions, along with the reasoning for adding those tools.

1. **AI critic**: LLMs are very sensitive to the prompt they are presented with. Past research has shown varying performance depending on the setting in which the LLM is queried. Thus, an LLM prompted to behave as an agent for a human researcher may behave very differently from one that is prompted to mainly critique the prediction made by another LLM. To benefit from these contrastive prompting strategies, we make use of an AI critic similar to the idea by Weng et al. (2023) to identify mistakes and enhance the quality of the final prediction made by the agent.

   Research into AI-based self critique and reinforcement has shown a significant impact in LLM responses. For example, Bai et al. (2022) introduce RL from AI feedback (RLAIF) where they used AI-based feedback on the responses from the same AI model to successfully finetune a different model. Another example is Shinn et al. (2023a) where agents verbally reflect on task feedback, then maintain their own reflections in memory to enable better decision-making in subsequent trials.

   At every round, once BioDiscoveryAgent comes up with a batch of genes to be tested, a critic agent (which is also an LLM) is prompted to critique the choice of the main agent and it can change some or all the genes in the batch and come up with a new set of genes (Figure 5e). Having such an agent improved performance giving the system additional opportunities to reflect on its reasoning. The system prompt for the critic agent was as follows:

   ```
   As an advisor, please critique this plan and suggest
       some changes to it. Use this format:
   1. Critique: include all relevant details of the
       critique.
   2. Updated Solution: Give an updated selection of {args
       .num_genes} genes based on the critique separated
       by commas in this format:: 1. <Gene name 1>, 2. <
       Gene name 2> ... \n

   Try to first focus a lot on trying very diverse genes
       to get a sense of which types of genes affect the
       research problem the most. From the observations,
       update your beliefs quickly and smartly and then
       double down on genes that you think shall be hits
       in order to increase the cumulative hit ratio.

   Please do not critique/make a lot of changes if there
       is no need to make a change.
   ```

   In addition to the above prompt, the critic agent was also provided with a list of all genes that were tested in the previous rounds along with genes that were identified as hits.

2. **Literature Search**: Scientific literature captures prior knowledge that can be leveraged to design experiments. A scientist typically reads literature relevant to a problem, builds a hypothesis, and cites relevant past work. We aimed to provide BioDiscoveryAgent with similar capabilities that would allow it to search for relevant papers on the web, learn from them, and incorporate its learnings in designing the experiments.

   As part of the input, the user can choose to have the agent query the literature at each experimental cycle to inform its predictions. In this case, the agent uses the PubMed API (Wobben, 2020) to search for papers containing the most pertinent literature for the experiments that the agent was asked to design. The search terms are chosen by the agent. Once the top 5 papers have been identified, the agent summarizes the information within

these papers including parsed title, abstract, methods, results, and conclusion sections. It then attaches the summarization to the prompt and use it to identify additional genes to perturb for the given experimental round (Figure 5b). The citations to these papers are retained and returned along with the model predictions. Over time, the agent accumulated these summaries, granting it access to all literature surveys conducted in previous rounds to propose a set of genes for each specific round.

The inclusion of a literature review tool enhanced interpretability and improved grounding as the agent frequently cited papers it had previously surveyed when predicting genes for the batch. Despite these benefits, the use of a literature review tool had its drawbacks. It tended to make the agent less exploratory, heavily biasing its reasoning towards the papers retrieved. The agent struggled to generate innovative queries for the literature survey API, and the lack of additional re-ranking on API outputs further limited the diversity of papers presented to the agent. This raised some important questions about the most effective use of scientific literature by an AI agent, a topic for future work.

3. **Gene Search**: LLMs are trained on text-based data and do not have access to many biological databases that are stored in the form of tables. We provide the agent with the ability to search for top 10 genes based on different criteria. The selected genes are concatenated to the prompt given to the main agent (Figure 5d). The results of augmenting the agent with these different gene search methods are summarized in Table 5.

   - **Similar/Dissimilar Genes**: The agent selects a gene for which it would like to search for similar or dissimilar genes. This is computed using the cosine similarity between the provided gene features (gene co-essentiality profiles (Tsherniak et al., 2017)). This process first computes the inner product of gene features with the queried gene's features and sorts the results based on the desired similarity or dissimilarity.
   - **Correlated Genes**: The agent accesses ARCHS4 (Lachmann et al., 2018), a public database that provides RNA-seq expression data from human and mouse samples. The API call ranks genes based on the Pearson correlation coefficient, a linear relationship between two genes' expression levels across multiple samples. Then the call returns top 10 most correlated genes to a gene that the agent selects for query.
   - **Genes in Common Tissues**: The agent accesses ARCHS4 to retrive a list of tissue types where the gene is expressed. Then the API call looks for top 10 genes that also show strong expression in these tissues, based on the median TPM (transcripts-per-million) value.
   - **KEGG Enrichment Analysis**: The agent accesses the KEGG (Kanehisa et al., 2017) enrichment database, which provide curated information on biological pathways and functions associated with genes. For hit genes from previous rounds, the top 10 pathways that are over-represented by statistical significance are returned. For these pathways, the API call will generate the top 10 genes that are associated with them.
   - **Reactome Enrichment Analysis**: The agent perform similar enrichment analysis to the above, but uses the Reactome database (Gillespie et al., 2022) instead.

## D.1 RECOMMENDATIONS ON TOOL USE FOR BIODISCOVERYAGENT

In this section, we provide some more details about the tools provided to BioDiscoveryAgent to aid it in making its predictions, along with the reasoning for adding those tools.

1. When using lower cost LLMs like Claude v1, Claude 3 Haiku and GPT3.5, tool use consistently improves performance. Based on Appendix Table 10, the all-tools setting should give the best result.

2. When using larger LLMs like Claude 3.5 Sonnet or GPT-o1-mini, then tool use can hurt performance and should not be used

3. If literature citations is important, then including the literature search tool can be beneficial with a minor cost in performance

4. If further understanding of model decision-making is helpful then the conversation with the critique agent can provide deeper context and critical analysis.

| Tools Used | Schmidt1 | Schmidt2 | CAR-T | Scharen. | Carnev. | Sanchez |
|---|---|---|---|---|---|---|
| Random | 0.037 | 0.031 | 0.033 | 0.160 | 0.036 | 0.034 |
| **Claude 3.5 Sonnet** | | | | | | |
| No-Tools | 0.095 | **0.104** | 0.130 | 0.326 | 0.042 | 0.066 |
| Similar | 0.091 (-4%) | 0.098 (-6%) | **0.145** (+12%) | 0.342 (+5%) | 0.043 (+2%) | 0.061 (-8%) |
| Dissimilar | 0.091 (-4%) | 0.096 (-8%) | 0.123 (-5%) | 0.302 (-7%) | 0.042 (+0%) | **0.069** (+5%) |
| Correlated | 0.088 (-7%) | 0.094 (-10%) | 0.138 (+6%) | **0.358** (+10%) | 0.044 (+4%) | 0.058 (-12%) |
| Tissues | 0.091 (-4%) | 0.097 (-7%) | 0.130 (+0%) | 0.310 (-5%) | 0.042 (+0%) | 0.066 (+0%) |
| KEGG | 0.089 (-6%) | 0.091 (-13%) | 0.122 (-6%) | 0.326 (+0%) | **0.045** (+6%) | 0.057 (-14%) |
| Reactome | **0.096** (+1%) | 0.100 (-4%) | 0.123 (-5%) | 0.348 (+7%) | 0.043 (+2%) | 0.062 (-6%) |
| **Claude 3 Haiku** | | | | | | |
| No-Tools | 0.064 | 0.072 | 0.116 | 0.209 | 0.032 | 0.056 |
| Similar | 0.065 (+2%) | 0.093 (+29%) | 0.095 (-18%) | 0.206 (-1%) | 0.043 (+34%) | 0.054 (-4%) |
| Dissimilar | 0.066 (+3%) | 0.073 (+1%) | 0.081 (-30%) | 0.215 (+3%) | **0.046** (+44%) | 0.054 (-18%) |
| Correlated | 0.068 (+6%) | 0.089 (+24%) | 0.108 (-7%) | **0.259** (+19%) | 0.043 (34%) | 0.047 (-16%) |
| Tissues | 0.063 (-2%) | 0.081 (+13%) | **0.119** (+3%) | 0.226 (+8%) | 0.037 (+16%) | 0.053 (-5%) |
| KEGG | 0.070 (+9%) | 0.074 (+3%) | 0.099 (-15%) | 0.232 (+11%) | 0.036 (+13%) | 0.047 (-16%) |
| Reactome | **0.080** (+25%) | **0.098** (+36%) | 0.114 (-2%) | 0.249 (+14%) | **0.046** (+44%) | **0.065** (+16%) |

Table 5: **Effect of different gene search tools.** Results show hit ratio for experimental round 5 averaged over 10 runs, with error intervals showing 1 standard deviation. *For Scharenberg et al. (2023), a batch size of 32 was used due to its smaller pool of 1061 relevant genes. Schmidt1 refers to the screen measuring Interferon-$\gamma$ (IFNG) and Schmidt2 measures Interleukin-2 (IL-2) following perturbation (Schmidt et al., 2022). †CAR-T refers to an unpublished dataset.

# E BASELINES

For baseline models, we use the methods implemented in the GeneDisco benchmark (Mehrjou et al. (2021), Lyle et al. (2023)). Every baseline includes a multi-layer perceptron $M$ for predicting experimental outcomes using gene features. This is then combined with one of seven different acquisition functions for designing each round of experiments:

- **Soft Uncertain**: Prioritizes genes with higher uncertainty under $M$, using a softmax function with temperature.

- **Top Uncertain**: Selects genes with the highest uncertainty under model $M$.

- **Margin Sample**: Selects genes for which the model $M$ has the smallest margins between different classes.

- **Coreset**: Selects genes which are the most distant from previously selected genes based on their embedding representation in $M$.

- **Badge**: Uses a modified k-means algorithm on the gradient embeddings of the data points to select genes. The aim is to diversify the batch based on the model's gradients.

- **Kmeans**: Selects genes that are closest to the cluster centers determined by K-means. Two baselines apply K-means either to an embedding of the data or the raw data directly.

- **DiscoBax** (Lyle et al., 2023): Selects genes with high expected change to the phenotype of interest as well as high diversity. Implemented as a set-value maximization problem.

.

In addition to published baselines, we include a Human baseline. The human baseline uses pathways and traditional enrichment analysis to sample genes. In the first round, we select the genes most active in pathways related to each test. These pathways are listed in Table 6. Then in subsequent rounds, enrichment analysis is performed to previous samples using Reactome (Gillespie et al., 2022) and KEGG (Kanehisa et al., 2017) databases.

Table 6 contains the pathways used to sample initial genes for each dataset during human baseline.

| Category | Pathway Name | Reactome ID | KEGG ID |
|---|---|---|---|
| IFNG | Interferon Gamma Signaling | R-HSA-877300 | hsa04060 |
| | Cytokine Signaling | R-HSA-1280215 | hsa04060 |
| IL-2 | Interleukin-2 Signaling | R-HSA-451927 | hsa04060 |
| | Cytokine Signaling | R-HSA-1280215 | hsa04060 |
| Carnevale | PD-1 Signaling | R-HSA-389948 | hsa05235 |
| | T-Cell Receptor Signaling | R-HSA-202433 | hsa04660 |
| | Immune Checkpoints | R-HSA-389957 | hsa05235 |
| CAR-T Proliferation | IL-2 Signaling Pathway | R-HSA-451927 | hsa04060 |
| | PI3K-Akt Signaling Pathway | R-HSA-110021 | hsa04151 |
| | mTOR Signaling Pathway | R-HSA-165159 | hsa04150 |
| Scharenberg | Lysosomal Transport | R-HSA-3229371 | hsa04142 |
| | Choline Metabolism | R-HSA-6798163 | hsa00564 |
| | Autophagy | R-HSA-1632852 | hsa04140 |
| Sanchez | MAPK Signaling Pathway | R-HSA-5683057 | hsa04010 |
| | Protein Processing in ER | R-HSA-381119 | hsa04141 |
| | Ubiquitin-Proteasome Pathway | R-HSA-983168 | hsa04120 |

Table 6: **Reactome and KEGG Pathways Used to Sample Genes during Human Baseline.** These pathways were used to sample initial batch of genes. In subsequent rounds, enrichment analysis using the Reactome database (Gillespie et al., 2022) was performed to sample the next batch. If the sampled genes were not sufficient in number, KEGG enrichment (Kanehisa et al., 2017) was also performed to fill the remaining samples.

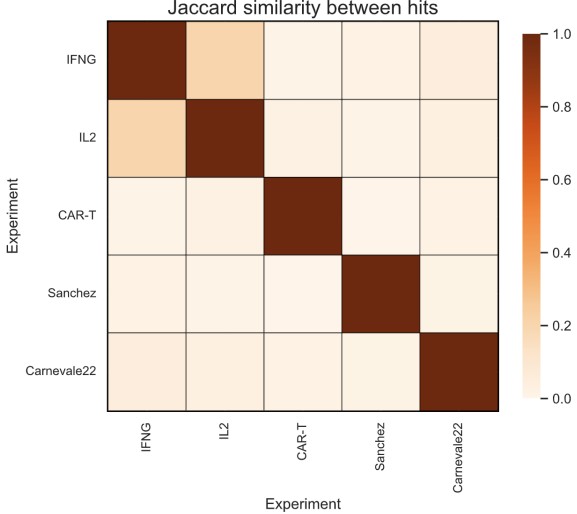

Figure 6: **Jaccard similarity index between all predicted hits at 5 rounds of experimentation** Each cell corresponds to the union of predicted hits across 10 model runs for each dataset.

## F  COMPUTATIONAL COST

Computational cost associated with these tool use is another important aspect to consider. We observe that even considering the API cost increase, the tools designed in this paper works especially well for smaller models like Claude 3 Haiku. From Table 4, we observe that tool usage results in 21% performance improvement in average. This only comes with the average of $0.14 increase in cost per trial (30%) - which is much less than what would be expected from literature review and critic agent. This is because tools help models to produce the required gene list in fewer number of turns. However, the tool usage with Claude 3.5 Sonnet neither reduces the cost or improves the performance. Table 11 shows the average number of input and output tokens along with API cost for these two models with each dataset.

| Model | Schmidt1 | Schmidt2 | CAR-T[†] | Scharen.[*] | Carnev. |
|---|---|---|---|---|---|
| Random | $0.037 \pm 0.013$ | $0.031 \pm 0.002$ | $0.033 \pm 0.003$ | $0.160 \pm 0.028$ | $0.036 \pm 0.001$ |
| **Baseline Models** | | | | | |
| Soft Uncertain | $0.037 \pm 0.006$ | $0.037 \pm 0.006$ | $0.034 \pm 0.007$ | $0.205 \pm 0.006$ | $0.031 \pm 0.006$ |
| Top Uncertain | $0.057 \pm 0.007$ | $0.072 \pm 0.014$ | $0.058 \pm 0.010$ | $0.294 \pm 0.030$ | $0.037 \pm 0.005$ |
| Margin Sample | $0.054 \pm 0.006$ | $0.061 \pm 0.009$ | $0.054 \pm 0.013$ | $0.285 \pm 0.019$ | $0.036 \pm 0.003$ |
| Coreset | $0.072 \pm 0.007$ | $0.102 \pm 0.005$ | $0.069 \pm 0.008$ | $0.243 \pm 0.031$ | $\mathbf{0.047 \pm 0.006}$ |
| Badge | $0.060 \pm 0.008$ | $0.077 \pm 0.008$ | $0.042 \pm 0.017$ | $0.258 \pm 0.032$ | $0.044 \pm 0.006$ |
| Kmeans Embed. | $0.045 \pm 0.004$ | $0.064 \pm 0.007$ | $0.028 \pm 0.011$ | $0.170 \pm 0.032$ | $0.036 \pm 0.004$ |
| Kmeans Data | $0.048 \pm 0.005$ | $0.074 \pm 0.009$ | $0.025 \pm 0.012$ | $0.281 \pm 0.042$ | $0.039 \pm 0.004$ |
| DiscoBAX | $0.088 \pm 0.000$ | $0.074 \pm 0.049$ | $0.047 \pm 0.000$ | $0.201 \pm 0.018$ | $0.045 \pm 0.000$ |
| **BioDiscoveryAgent** (No-Tools) | | | | | |
| Claude 3 Haiku | $0.064 \pm 0.005$ | $0.071 \pm 0.018$ | $0.116 \pm 0.014$ | $0.209 \pm 0.030$ | $0.032 \pm 0.004$ |
| GPT-3.5-turbo | $0.044 \pm 0.007$ | $0.061 \pm 0.009$ | $0.064 \pm 0.012$ | $0.230 \pm 0.032$ | $0.032 \pm 0.004$ |
| Claude v1 | $0.067 \pm 0.010$ | $0.089 \pm 0.014$ | $0.110 \pm 0.023$ | $0.292 \pm 0.064$ | $0.038 \pm 0.005$ |
| o1-mini | $0.081 \pm 0.007$ | $0.096 \pm 0.013$ | $0.127 \pm 0.016$ | $0.279 \pm 0.040$ | $0.040 \pm 0.004$ |
| Claude 3 Sonnet | $0.076 \pm 0.006$ | $0.088 \pm 0.007$ | $0.115 \pm 0.012$ | $0.302 \pm 0.036$ | $0.041 \pm 0.003$ |
| Claude 3.5 Sonnet | $\mathbf{0.095 \pm 0.004}$ | $\mathbf{0.104 \pm 0.004}$ | $0.130 \pm 0.010$ | $\mathbf{0.326 \pm 0.034}$ | $0.042 \pm 0.003$ |
| GPT-4o | $0.059 \pm 0.006$ | $0.074 \pm 0.006$ | $0.116 \pm 0.031$ | $0.311 \pm 0.037$ | $0.037 \pm 0.004$ |
| o1-preview | $0.081 \pm 0.003$ | $0.091 \pm 0.004$ | $\mathbf{0.141 \pm 0.022}$ | $0.283 \pm 0.015$ | $0.041 \pm 0.001$ |
| Claude 3 Opus | $0.094 \pm 0.009$ | $\mathbf{0.104 \pm 0.007}$ | $0.126 \pm 0.011$ | $0.247 \pm 0.037$ | $0.043 \pm 0.005$ |

| Model | Sanchez |
|---|---|
| Random | $0.033 \pm 0.004$ |
| **Baseline Models** | |
| Soft Uncertain | $0.029 \pm 0.003$ |
| Top Uncertain | $0.039 \pm 0.007$ |
| Margin Sample | $0.033 \pm 0.005$ |
| Coreset | $\mathbf{0.061 \pm 0.005}$ |
| Badge | $0.039 \pm 0.005$ |
| Kmeans Embed. | $0.037 \pm 0.003$ |
| Kmeans Data | $0.043 \pm 0.004$ |
| DiscoBAX | $0.058 \pm 0.007$ |
| **BioDiscoveryAgent** (No-Tools) | |
| Claude 3 Haiku | $0.056 \pm 0.005$ |
| GPT-3.5-turbo | $0.039 \pm 0.004$ |
| Claude v1 | $0.058 \pm 0.007$ |
| o1-mini | $\mathbf{0.074 \pm 0.005}$ |
| Claude 3 Sonnet | $0.064 \pm 0.012$ |
| Claude 3.5 Sonnet | $0.066 \pm 0.007$ |
| GPT-4o | $0.030 \pm 0.003$ |
| o1-preview | $0.068 \pm 0.006$ |
| Claude 3 Opus | $0.059 \pm 0.009$ |

Table 7: **Performance comparison to machine learning baselines for 1-gene perturbation experiments**. Results show hit ratio for experimental round 5 averaged over 10 runs, with error intervals showing 1 standard deviation. [*]For Scharenberg22, a batch size of 32 was used due to its smaller pool of 1061 relevant genes. [†]CAR-T refers to an unpublished dataset. BDAgent stands for BioDiscoveryAgent.

| Model | Schmidt1 | | Schmidt2 | | CAR-T[†] | | Scharen.[*] | | Carnev. | | Sanchez | | Average |
|---|---|---|---|---|---|---|---|---|---|---|---|---|---|
| | All | N/E | All | N/E | All | N/E | All | N/E | All | N/E | All | N/E | Rank |
| Claude 3 Haiku | 7 | 7 | 8 | 8 | 5 | 7 | 9 | 8 | 8 | 8 | 6 | 6 | 7.25 |
| GPT-3.5-Turbo | 9 | 9 | 9 | 9 | 9 | 9 | 8 | 9 | 8 | 8 | 8 | 9 | 8.67 |
| Claude v1 | 6 | 5 | 5 | 6 | 8 | 5 | 4 | 6 | 6 | **1** | 7 | 7 | 6.00 |
| o1-mini | 3 | 3 | 3 | 3 | 3 | 2 | 6 | **2** | 5 | 5 | **1** | **1** | 3.08 |
| Claude 3 Sonnet | 5 | 6 | 6 | 5 | 7 | 8 | 3 | 4 | 3 | 2 | 4 | 2 | 4.58 |
| Claude 3.5 Sonnet | **1** | **1** | **1** | 2 | 2 | 3 | **1** | 1 | 2 | 2 | 3 | 2 | **1.75** |
| GPT-4o | 8 | 8 | 7 | 7 | 5 | 6 | 2 | **2** | 7 | 7 | 9 | 8 | 6.33 |
| o1-preview | 3 | 4 | 4 | 4 | **1** | **1** | 5 | 5 | 3 | 2 | 2 | 2 | 3.00 |
| Claude 3 Opus | 2 | 2 | **1** | **1** | 4 | 4 | 7 | 7 | **1** | 5 | 5 | 5 | 3.67 |

Table 8: Rankings of models for each dataset, with ties assigned the lower rank. Bold indicates the best performance; underlined indicates the second-best performance.

| Tools Used | Schmidt1 | Schmidt2 | CAR-T | Scharen. | Carnev. | Sanchez |
|---|---|---|---|---|---|---|
| Random | $0.037 \pm 0.013$ | $0.031 \pm 0.002$ | $0.033 \pm 0.003$ | $0.160 \pm 0.028$ | $0.036 \pm 0.001$ | $0.034 \pm 0.004$ |
| **BioDiscoveryAgent (Claude 3.5 Sonnet)** | | | | | | |
| No-Tools | $0.095 \pm 0.004$ | $\mathbf{0.104 \pm 0.004}$ | $0.130 \pm 0.010$ | $\mathbf{0.326 \pm 0.034}$ | $0.042 \pm 0.003$ | $0.066 \pm 0.007$ |
| Literature | $\mathbf{0.096 \pm 0.005}$ | $0.098 \pm 0.010$ | $\mathbf{0.138} \pm 0.019$ | $0.309 \pm 0.041$ | $0.042 \pm 0.002$ | $\mathbf{0.069 \pm 0.006}$ |
| AI Critic | $0.088 \pm 0.004$ | $0.092 \pm 0.010$ | $0.126 \pm 0.019$ | $0.309 \pm 0.037$ | $0.042 \pm 0.002$ | $0.059 \pm 0.006$ |
| Gene Search | $\mathbf{0.096 \pm 0.003}$ | $0.100 \pm 0.009$ | $0.123 \pm 0.009$ | $\mathbf{0.348 \pm 0.025}$ | $\mathbf{0.043 \pm 0.004}$ | $0.062 \pm 0.007$ |
| All-Tools | $\mathbf{0.096} \pm 0.005$ | $0.090 \pm 0.003$ | $0.121 \pm 0.020$ | $0.234 \pm 0.080$ | $\mathbf{0.043} \pm 0.001$ | $0.054 \pm 0.004$ |
| **BioDiscoveryAgent (Claude 3 Haiku)** | | | | | | |
| No-Tools | $0.064 \pm 0.005$ | $0.072 \pm 0.018$ | $0.116 \pm 0.014$ | $0.209 \pm 0.030$ | $0.032 \pm 0.004$ | $0.056 \pm 0.005$ |
| Literature | $0.053 \pm 0.005$ | $0.069 \pm 0.011$ | $0.091 \pm 0.031$ | $0.164 \pm 0.068$ | $0.035 \pm 0.006$ | $0.054 \pm 0.009$ |
| AI Critic | $0.061 \pm 0.009$ | $0.070 \pm 0.009$ | $0.113 \pm 0.012$ | $0.219 \pm 0.046$ | $0.043 \pm 0.006$ | $0.054 \pm 0.009$ |
| Gene Search | $0.080 \pm 0.013$ | $0.098 \pm 0.025$ | $0.114 \pm 0.016$ | $0.249 \pm 0.067$ | $\mathbf{0.046 \pm 0.006}$ | $\mathbf{0.065 \pm 0.009}$ |
| All-Tools | $\mathbf{0.084 \pm 0.006}$ | $\mathbf{0.099 \pm 0.019}$ | $\mathbf{0.128 \pm 0.031}$ | $\mathbf{0.259 \pm 0.039}$ | $0.043 \pm 0.008$ | $0.058 \pm 0.011$ |

Table 9: **Agent performance improvements when using different tools for 1-gene perturbation experiments**. Results show hit ratio for experimental round 5 averaged over 10 runs, with the error intervals. See Table 1 caption for notes on specific datasets.

| Tools Used | Schmidt1 | Schmidt2 | CAR-T | Scharen. | Carnev. | Sanchez |
|---|---|---|---|---|---|---|
| Random | 0.037 | 0.031 | 0.033 | 0.160 | 0.036 | 0.034 |
| **Claude 3 Haiku** | | | | | | |
| No-Tools | 0.064 | 0.072 | 0.116 | 0.209 | 0.032 | 0.056 |
| All-Tools | 0.084 (+31%) | 0.099 (+38%) | 0.128 (+10%) | 0.259 (+24%) | 0.043 (+34%) | 0.058 (+4%) |
| **GPT-3.5-Turbo** | | | | | | |
| No-Tools | 0.044 | 0.061 | 0.064 | 0.230 | 0.032 | 0.039 |
| All-Tools | 0.062 (+41%) | 0.100 (+64%) | 0.063 (-2%) | 0.218 (-5%) | 0.037 (+16%) | 0.045 (+15%) |
| **Claude v1** | | | | | | |
| No-Tools | 0.067 | 0.089 | 0.110 | 0.292 | 0.038 | 0.053 |
| All-Tools | 0.095 (+42%) | 0.122 (+37%) | 0.114 (+4%) | 0.333 (+14%) | 0.054 (+42%) | 0.058 (+9%) |
| **GPT o1-mini** | | | | | | |
| No-Tools | 0.081 | 0.096 | 0.127 | 0.279 | 0.041 | 0.074 |
| All-Tools | 0.083 (+2%) | 0.075 (-22%) | 0.114 (-10%) | 0.264 (-5%) | 0.041 (+0%) | 0.072 (-3%) |
| **Claude 3 Sonnet** | | | | | | |
| No-Tools | 0.076 | 0.088 | 0.115 | 0.302 | 0.041 | 0.064 |
| All-Tools | 0.074 (-3%) | 0.091 (+3%) | 0.105 (-9%) | 0.302 (+0%) | 0.047 (+15%) | 0.073 (+14%) |
| **Claude 3.5 Sonnet** | | | | | | |
| No-Tools | 0.095 | 0.104 | 0.130 | 0.326 | 0.042 | 0.066 |
| All-Tools | 0.096 (+1%) | 0.090 (-13%) | 0.121 (-6%) | 0.234 (-28%) | 0.043 (+2%) | 0.054 (-18%) |
| **GPT-4o** | | | | | | |
| No-Tools | 0.059 | 0.074 | 0.116 | 0.311 | 0.037 | 0.030 |
| All-Tools | 0.049 (-17%) | 0.064 (-14%) | 0.103 (-11%) | 0.285 (-8%) | 0.035 (-5%) | 0.026 (-13%) |
| **GPT o1-preview** | | | | | | |
| No-Tools | 0.081 | 0.091 | 0.141 | 0.283 | 0.041 | 0.068 |
| All-Tools | 0.052 (-36%) | 0.079 (-13%) | 0.098 (-30%) | 0.289 (+2%) | 0.024 (-41%) | 0.031 (-54%) |
| **Claude 3 Opus** | | | | | | |
| No-Tools | 0.094 | 0.104 | 0.126 | 0.247 | 0.043 | 0.059 |
| All-Tools | 0.096 (+2%) | 0.101 (-3%) | 0.116 (-8%) | 0.291 (+18%) | 0.041 (-5%) | 0.055 (-7%) |

Table 10: **Effect of tools on different LLMs.** Results show hit ratio for experimental round 5 averaged over 10 runs, with error intervals showing 1 standard deviation. *For Scharenberg et al. (2023), a batch size of 32 was used due to its smaller pool of 1061 relevant genes. Schmidt1 refers to the screen measuring Interferon-$\gamma$ (IFNG) and Schmidt2 measures Interleukin-2 (IL-2) following perturbation (Schmidt et al., 2022). †CAR-T refers to an unpublished dataset.

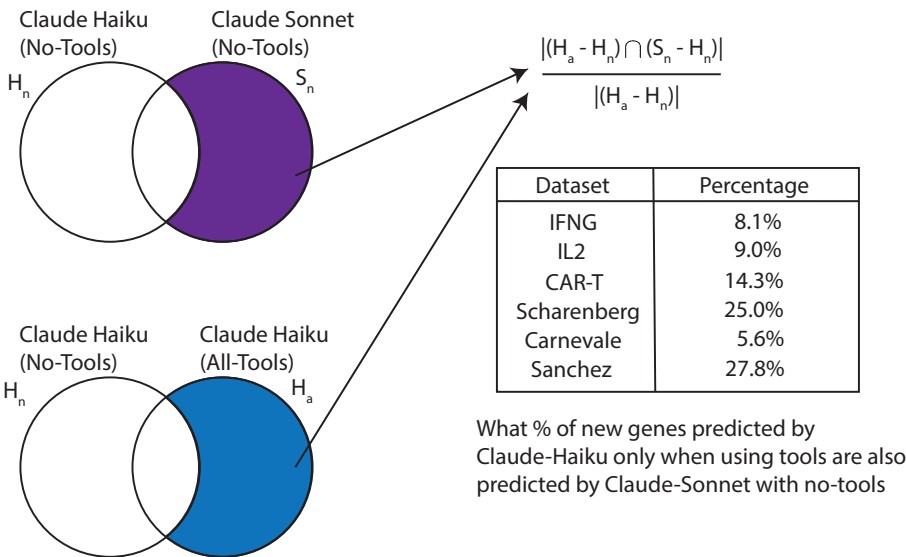

Figure 7: Percentage of new genes predicted by Claude Haiku only when using tools $H_a - H_n$ that are also predicted by Claude Sonnet with no-tools $(S_n)$, where $H_n$ is the set of genes predicted by Claude Haiku with no-tools and $H_a$ is the set of genes predicted by Claude Haiku with all-tools

| Model | Dataset | Avg. Input Tokens | Avg. Cost per Trial ($/trial) |
|---|---|---|---|
| Claude 3.5 Sonnet (No-Tool) | Schmidt1 | 130409 | 0.61 |
| | Schmidt2 | 120494 | 0.58 |
| | CAR-T | 99867 | 0.49 |
| | Scharen. | 92594 | 0.59 |
| | Carnev. | 126206 | 0.60 |
| | Sanchez | 88802 | 0.42 |
| Claude 3.5 Sonnet (All-Tools) | Schmdit1 | 474648 | 2.38 |
| | Schmidt2 | 440566 | 2.26 |
| | CAR-T | 386379 | 1.96 |
| | Scharen. | 314562 | 2.18 |
| | Carnev. | 415469 | 2.14 |
| | Sanchez | 351780 | 1.70 |
| Claude 3 Haiku (No-Tool) | Schmidt1 | 271173 | 0.47 |
| | Schmidt2 | 269938 | 0.47 |
| | CAR-T | 264463 | 0.45 |
| | Scharen. | 89393 | 0.19 |
| | Carnev. | 263726 | 0.46 |
| | Sanchez | 230232 | 0.37 |
| Claude 3 Haiku (All-Tools) | Schmidt1 | 347435 | 0.61 |
| | Schmidt2 | 353799 | 0.61 |
| | CAR-T | 372581 | 0.67 |
| | Scharen. | 103950 | 0.23 |
| | Carnev. | 353511 | 0.61 |
| | Sanchez | 289256 | 0.51 |

Table 11: Token Usage and API Cost for Gene Perturbation by Model and Dataset

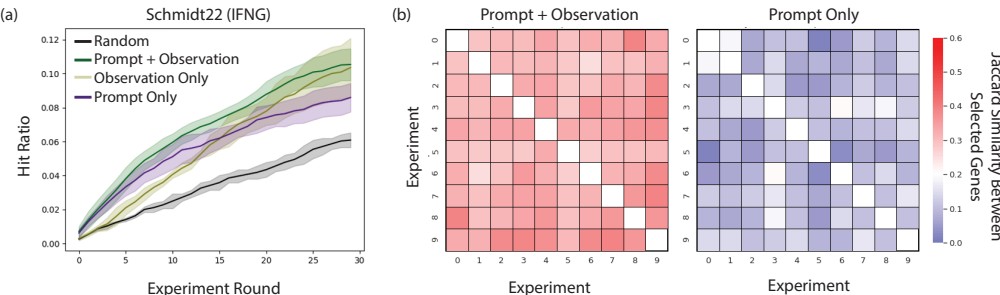

Figure 8: **Agent accounts for both prior knowledge and observations in decision-making** Three scenarios are considered: BioDiscoveryAgent (Claude v1) has access to task description and experimental observations (Prompt + Observation); the agent only has access to experimental observations (Observation Only); the agent only has access to the task description (Prompt Only). **(a)** Hit ratio at each experimental round across 30 rounds of experiments. 32 genes predicted per round. Each line corresponds to the average over 10 runs with error bars representing 1 standard deviation. **(b)** Jaccard similarity index between all predicted genes at 10 rounds of experimentation. Each cell corresponds to a different model run.

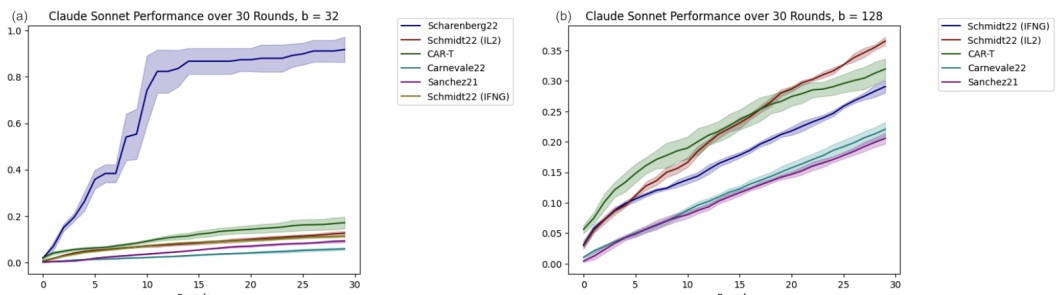

Figure 9: **Agent shows consistent performance for the first 30 rounds** BioDiscoveryAgent (Claude 3.5 Sonnet) is tested in two different settings to observe its performance over time: hit ratio at each experimental round across 30 rounds of experiments with **(a)** 32 genes predicted per round and **(b)** 128 genes per round. Each line corresponds to the average over 5 runs with error bars representing 1 standard deviation. Scharenberg et al. (2023) is only tested with a batch size of 32 due to smaller overall size of the dataset: 1061 perturbations

| Model | Schmidt1 | | | Schmidt2 | | | CAR-T[†] | | |
|---|---|---|---|---|---|---|---|---|---|
| | 10 | 20 | 30 | 10 | 20 | 30 | 10 | 20 | 30 |
| Random | 0.06 | 0.123 | 0.181±0.011 | 0.047 | 0.092 | 0.139±0.014 | 0.06 | 0.119 | 0.179±0.018 |
| **Baseline Models** | | | | | | | | | |
| Soft Uncertain | 0.061 | 0.123 | 0.192±0.008 | 0.049 | 0.100 | 0.148±0.012 | 0.054 | 0.127 | 0.186±0.027 |
| Top Uncertain | 0.077 | 0.146 | 0.216±0.009 | 0.068 | 0.137 | 0.200±0.011 | 0.074 | 0.139 | 0.204±0.013 |
| Margin | 0.070 | 0.143 | 0.211±0.009 | 0.068 | 0.130 | 0.192±0.014 | 0.075 | 0.136 | 0.209±0.018 |
| Coreset | 0.123 | 0.231 | 0.303±0.005 | 0.136 | 0.248 | 0.314±0.006 | 0.092 | 0.144 | 0.227±0.012 |
| Badge | 0.079 | 0.153 | 0.226±0.013 | 0.082 | 0.154 | 0.217±0.005 | 0.064 | 0.144 | 0.208±0.025 |
| K-Means (E) | 0.066 | 0.13 | 0.198±0.007 | 0.072 | 0.131 | 0.194±0.008 | 0.052 | 0.097 | 0.161±0.013 |
| K-Means (D) | 0.069 | 0.14 | 0.203±0.008 | 0.086 | 0.172 | 0.235±0.006 | 0.060 | 0.125 | 0.189±0.005 |
| **BioDiscoveryAgent** (No-Tools) | | | | | | | | | |
| Sonnet | **0.143** | 0.214 | 0.293±0.005 | **0.179** | **0.286** | **0.360±0.001** | **0.193** | **0.264** | **0.329±0.014** |
| **BioDiscoveryAgent** + Baseline Models | | | | | | | | | |
| Sonnet+Coreset | 0.141 | **0.247** | **0.314±0.006** | 0.147 | 0.256 | 0.324±0.005 | 0.142 | 0.234 | 0.301±0.007 |

Table 12: **Performance comparison for 1-gene perturbation experiments across 30 rounds (Batch size:128)**. Hit ratio for experimental rounds 5, 10, 20, 30 averaged over 10 runs, with 128 genes predicted in each round. Results shown for non-essential genes. The best performing model in each class is highlighted in gray. Schmidt1 refers to the screen measuring Interferon-$\gamma$ (IFNG) and Schmidt2 measures Interleukin-2 (IL-2) following perturbation (Schmidt et al., 2022). [†]CAR-T refers to an unpublished dataset.

| Model | Carnevale | | | Sanchez | | |
|---|---|---|---|---|---|---|
| | 10 | 20 | 30 | 10 | 20 | 30 |
| Random | 0.037 | 0.080 | 0.111±0.057 | 0.062 | 0.123 | 0.186±0.007 |
| **Baseline Models** | | | | | | |
| Soft Uncertain | 0.057 | 0.102 | 0.160±0.050 | 0.063 | 0.122 | 0.183±0.017 |
| Top Uncertain | 0.063 | 0.13 | 0.195±0.004 | 0.093 | 0.173 | 0.245±0.007 |
| Margins | 0.084 | 0.152 | 0.211±0.008 | 0.074 | 0.152 | 0.218±0.013 |
| Coreset | 0.069 | 0.136 | 0.204±0.004 | 0.106 | 0.181 | 0.252±0.008 |
| Badge | 0.065 | 0.123 | 0.162±0.074 | 0.064 | 0.135 | 0.207±0.013 |
| K-Means (E) | 0.060 | 0.116 | 0.184±0.003 | 0.066 | 0.136 | 0.199±0.005 |
| K-Means (D) | 0.056 | 0.115 | 0.174 ± 0.007 | 0.07 | 0.14 | 0.196±0.007 |
| **BioDiscoveryAgent** (No-Tools) | | | | | | |
| Sonnet | **0.083** | **0.154** | **0.218±0.008** | **0.116** | **0.217** | **0.303±0.002** |
| **BioDiscoveryAgent** + Baseline Models | | | | | | |
| Sonnet+Coreset | 0.073 | 0.136 | 0.207±0.006 | 0.104 | 0.187 | 0.257±0.006 |

Table 13: **Performance comparison for 1-gene perturbation experiments across 30 rounds (Batch size:128)**. Hit ratio for experimental rounds 5, 10, 20, 30 averaged over 10 runs, with 128 genes predicted in each round. Results shown for non-essential genes. The best performing model in each class is highlighted in gray.

| Model | Schmidt1 | | | Schmidt2 | | | CAR-T[†] | | |
|---|---|---|---|---|---|---|---|---|---|
| | 10 | 20 | 30 | 10 | 20 | 30 | 10 | 20 | 30 |
| Random | 0.016 | 0.032 | 0.049±0.007 | 0.016 | 0.034 | 0.052±0.005 | 0.018 | 0.035 | 0.052±0.005 |
| **Baseline Models** | | | | | | | | | |
| Soft Uncertain | 0.014 | 0.029 | 0.044±0.006 | 0.013 | 0.028 | 0.045±0.006 | 0.02 | 0.038 | 0.053±0.017 |
| Top Uncertain | 0.019 | 0.040 | 0.061±0.005 | 0.023 | 0.048 | 0.067 ± 0.004 | 0.022 | 0.036 | 0.063 ± 0.003 |
| Margins | 0.014 | 0.032 | 0.053±0.005 | 0.016 | 0.036 | 0.054±0.008 | 0.03 | 0.038 | 0.057±0.006 |
| Coreset | 0.032 | 0.065 | 0.095±0.004 | 0.033 | 0.067 | 0.107±0.004 | 0.033 | 0.054 | 0.069±0.007 |
| Badge | 0.02 | 0.04 | 0.062±0.008 | 0.031 | 0.057 | 0.079±0.003 | 0.016 | 0.031 | 0.051±0.018 |
| K-Means (E) | 0.016 | 0.029 | 0.041±0.003 | 0.019 | 0.039 | 0.059±0.004 | 0.007 | 0.019 | 0.034±0.008 |
| K-Means (D) | 0.017 | 0.032 | 0.045±0.003 | 0.015 | 0.030 | 0.042±0.003 | 0.010 | 0.017 | 0.034±0.005 |
| **BioDiscoveryAgent** (No-Tools) | | | | | | | | | |
| Sonnet | **0.076** | **0.108** | **0.127±0.006** | **0.087** | **0.116** | **0.143±0.010** | **0.085** | **0.152** | **0.179±0.025** |
| **BioDiscoveryAgent** + Baseline Models | | | | | | | | | |
| Sonnet+Coreset | 0.049 | 0.070 | 0.086 ± 0.004 | 0.042 | 0.074 | 0.115 ± 0.007 | 0.054 | 0.073 | 0.093 ± 0.009 |

Table 14: **Performance comparison for 1-gene perturbation experiments across 30 rounds (Batch size:32)**. Hit ratio for experimental rounds 5, 10, 20, 30 averaged over 10 runs, with 32 genes predicted in each round. Results shown for non-essential genes. The best performing model in each class is highlighted in gray. Schmidt1 refers to the screen measuring Interferon-$\gamma$ (IFNG) and Schmidt2 measures Interleukin-2 (IL-2) following perturbation (Schmidt et al., 2022). [†]CAR-T refers to an unpublished dataset.

| Model | Scharen.* | | Carnevale | | | Sanchez | | |
|---|---|---|---|---|---|---|---|---|
| | 10 | 20 | 10 | 20 | 30 | 10 | 20 | 30 |
| Random | 0.243 | 0.518 ±0.045 | 0.016 | 0.033 | 0.049±0.006 | 0.015 | 0.034 | 0.053±0.003 |
| **Baseline Models** | | | | | | | | |
| Soft Uncertain | 0.281 | 0.57 ±0.029 | 0.018 | 0.037 | 0.052±0.01 | 0.016 | 0.030 | 0.045±0.005 |
| Top Uncertain | 0.354 | 0.604 ±0.0 | 0.014 | 0.033 | 0.048±0.004 | 0.017 | 0.038 | 0.070±0.004 |
| Margins | 0.319 | 0.604 ±0.0 | 0.012 | 0.027 | 0.044±0.005 | 0.012 | 0.028 | 0.046±0.005 |
| Coreset | 0.358 | 0.587 ±0.014 | 0.013 | 0.03 | 0.05±0.005 | 0.033 | 0.067 | **0.096±0.005** |
| Badge | 0.363 | 0.587 ±0.024 | 0.013 | 0.03 | 0.05±0.005 | 0.022 | 0.042 | 0.061±0.006 |
| K-Means (E) | 0.237 | 0.543 ±0.021 | 0.013 | 0.023 | 0.036±0.005 | 0.017 | 0.035 | 0.050±0.005 |
| K-Means (D) | 0.388 | 0.585 ±0.018 | 0.012 | 0.027 | 0.045 ± 0.001 | 0.015 | 0.032 | 0.048±0.004 |
| **BioDiscoveryAgent** (No-Tools) | | | | | | | | |
| Sonnet | **0.551** | **0.871±0.051** | **0.022** | **0.043** | **0.059±0.004** | **0.034** | **0.072** | 0.095±0.007 |
| **BioDiscoveryAgent** + Baseline Models | | | | | | | | |
| Sonnet+Coreset | 0.394 | 0.602 ± 0.023 | 0.023 | 0.036 | 0.054 ± 0.004 | 0.024 | 0.051 | 0.079 ± 0.004 |

Table 15: **Performance comparison for 1-gene perturbation experiments across 30 rounds (Batch size:32)**. Hit ratio for experimental rounds 5, 10, 20, 30 averaged over 10 runs, with 32 genes predicted in each round. Results shown for non-essential genes. The best performing model in each class is highlighted in gray. In Scharenberg, baseline methods as implemented in Mehrjou et al. (2021) were unable to proceed beyond 24 rounds due to smaller size of dataset.

| Dataset | | Schmidt22 (IL2) | | | Schmidt22 (IFNG) | | |
|---|---|---|---|---|---|---|---|
| Rounds | | 10 | 20 | 30 | 10 | 20 | 30 |
| Model | Setting | Avg. Hit Ratio | | | | | |
| BioDiscoveryAgent | Observ. Only | 0.0449 | 0.0872 | 0.1037 | 0.0291 | 0.0702 | 0.1061 |
| BioDiscoveryAgent | Prompt Only | **0.0605** | 0.0846 | 0.1071 | 0.0484 | 0.0727 | 0.0872 |
| BioDiscoveryAgent | Prompt + Observ. | 0.0596 | **0.0908** | **0.1143** | **0.0559** | **0.0842** | **0.1077** |

Table 16: **Agent uses both prior knowledge and experimental observations to make predictions.** Three scenarios: BioDiscoveryAgent (Claude v1) has access to task description and experimental observations (Prompt + Observation); the agent only has access to experimental observations (Observation Only); the agent only has access to the task description (Prompt Only). **(a)** Hit ratio at each experimental round across 30 rounds of experiments. 32 genes predicted per round across 30 rounds.

## G  COMPLETE AGENT TRACE

In this section, we provide a few initial steps of the agent run equipped with all tools, and prompts seen by the agent at each step, as well as the agent's response. We used BioDiscoveryAgent (Claude v1) for this analysis since it shows high absolute score as well as high performance gain through tool use (Appendix Table 10).

```
==================prompt====================

Human: Step 0
 You are a scientist working on problems in drug discovery.

Research Problem: I'm planning to run a genome-wide CRISPR screen
    to identify genes that regulate the production of Interferon-
    gamma (IFNG). There are 18,939 possible  genes to perturb and
    I can only perturb 128 genes at a time. For each perturbation,
     I'm able to measure out the log fold change in Interferon-
    gamma (IFNG) normalized read counts which will be referred to
    as the score. I can only do a few rounds of experimentation.

Here is one strategy you can follow: Try to first focus a lot on
    trying very diverse genes to
get a sense of which types of genes affect the research problem
    the most. From the observations,
update your beliefs quickly and smartly and then double down on
    genes which you think shall be hits
in order to increase the cumulative hit rate.

Always respond in this format exactly:

1. Reflection: Thoughts on previous results and next steps.
2. Research Plan: The full high level research plan, with current
    status and reasoning behind each proposed approach. It should
    be at most 5 sentences.
3. Gene Search: Name a gene to search for 50 most dissimilar genes
     based on features. Only include the gene name itself after
    "2. Gene Search:".
4. Solution: Propose a list of predicted genes to test separated
    by commas in this format: 1. <Gene name 1>, 2. <Gene name 2>
    ...
Do not include any genes from this prompt (since they're already
    tested).

Now let's start!

 You have done some literature review till now and have the
    following information at your disposal which you may use to
    make your predictions:

Ubiquitin ligase STUB1 destabilizes IFNG-receptor complex to
    suppress tumor IFNG signaling.
 Here is a summary of information that can be extracted from the
    provided file to help with the research problem:
```

The file mentions that STUB1 is an E3 ubiquitin ligase that mediates proteasomal degradation of the IFNG-R1/JAK1 complex ( lines 2-3). This suggests that perturbing STUB1 may increase IFNG production.

The file also refers to a genome-wide CRISPR/Cas9 screen that identified regulators of IFNG-R1 cell surface abundance (line 2). Running a similar screen focused on IFNG production may identify critical genes to perturb.

In this segment, I cannot find specific gene targets or pathways to focus on for the initial experiments. The provided information is too high-level.

The strategy suggested in the research problem to try diverse genes first and then focus on hits is reasonable given the limited number of perturbations possible. Updating beliefs based on initial observations will help identify the most promising genes to study further.

Mitochondrial respiration contributes to the interferon gamma response in antigen-presenting cells.
 Here is a summary of the key information from the passage:

The passage describes a genome-wide CRISPR screen in murine macrophages to identify regulators of IFNG-inducible proteins MHCII, CD40 and PD-L1. The screen found that complex I of the mitochondrial respiratory chain is involved in the expression of all three proteins and IFNG signaling (lines 3-7).

Based on this, I would focus initial experiments on perturbing genes involved in mitochondrial respiration and the electron transport chain, specifically complex I. Perturbing these genes could identify regulators of IFNG and its downstream targets. Once initial hits are found, I would follow up by perturbing more genes in those pathways to increase the cumulative hit rate, as suggested in the research problem description.

In this segment, I cannot find specific genes to target or exact numbers of genes that can be perturbed. The passage only refers to a "series of genome-wide knockout screens" and " multiscreen approach" (lines 3-4).

Genome-Wide CRISPR Screen Reveals Cancer Cell Resistance to NK Cells Induced by NK-Derived IFN-G.
 Based on the information provided, here are some insights I can gather:

The research aims to identify genes that regulate IFNG production using a CRISPR screen. The genes can be perturbed in batches of 128 at a time out of 18,939 total genes. The log fold change in IFNG counts can be measured for each perturbation to determine scores. Given the large search space and limited rounds of experimentation, an effective strategy is to first try diverse genes to determine which types most impact IFNG

production. Then focus on the most promising genes to increase
  the cumulative hit rate.

In lines 1-2, the research describes using a "genome-wide CRISPR
  screen" and  "primary human NK cells" to identify "genes that
  regulate the vulnerability of leukemia cells". This indicates
  the genes of interest are involved in the immune system and
  response to NK cells.

In line 3, "loss of" indicates the genes enable resistance to NK
  cells, so the genes of interest may be involved in
  susceptibility to NK cytotoxicity.

Based on the additional literature review, the genes of interest
  are likely involved in the production of IFNG, an immune
  signaling molecule. The screen aims to determine which of the
  18,939 genes regulate IFNG levels when perturbed.

In this segment, I cannot find specific gene names or pathways
  referenced in the provided text. More details on the
  experimental procedure and results would be needed to
  determine promising genes or gene types to prioritize.

The key is to start broad, then use observations to focus on the
  most promising areas. An iterative approach updating beliefs
  and re-prioritizing genes is suggested to increase the
  cumulative discovery of genes impacting IFNG regulation.

Overexpressing Long Noncoding RNAs Using Gene-activating CRISPR.
 Here is a summary of the information provided:

The research aims to overexpress long noncoding RNAs (lncRNAs) in
  vitro using the CRISPR activation technique in order to study
  their role in disease. As an example, the lncRNA IFNG-AS1
  associated with inflammatory bowel disease is overexpressed in
   Jurkat T cells. The CRISPR activation technique targets
  transcription factors to the transcriptional start site of a
  gene to enable overexpression of lncRNA splice forms. This
  procedure involves guide RNA design, virus generation and
  transduction, and colony screening. Overexpression of IFNG-AS1
   by greater than 20-fold was observed in Jurkat T cells.

Based on the research problem:

In this segment, I cannot find specific genes that regulate IFNG
  production. The file describes a general technique to
  overexpress lncRNAs using CRISPR activation but does not
  specify genes that regulate IFNG.

The strategy to first test diverse genes to determine which types
  are most impactful and then focus on likely "hits" is
  reasonable. Updating beliefs based on initial observations and
   doubling down on promising genes can increase the cumulative
  hit rate.

The literature review indicates the CRISPR activation technique
  can be used to overexpress lncRNAs to study their role in
  disease. Overexpression of the lncRNA IFNG-AS1, associated

with inflammatory bowel disease, was achieved in Jurkat T
    cells. This provides a proof of concept for the technique in
    the context of the research problem.

In summary, the file describes a useful technique to
    systematically overexpress lncRNAs which could be applied to
    identify genes regulating IFNG production. The example of
    overexpressing IFNG-AS1 in Jurkat T cells demonstrates the
    potential of this approach. The proposed strategy of testing
    diverse genes and focusing on likely hits is reasonable for
    this type of genome-wide screen.

=================================================

Assistant:
==================claude-1 response (2000)======================
 1. Reflection: The literature review provides useful context on
     techniques and pathways that may be involved in regulating
     IFNG production. However, more details on specific genes and
     experimental results are needed to determine the most
     promising candidates for initial perturbations. The strategy
     to start broad and focus on likely hits based on observations
      is reasonable given the large search space.

2. Research Plan: Run an initial set of diverse perturbations
    across pathways potentially involved in IFNG regulation based
    on literature review. Measure IFNG levels for each to
    determine scores and identify promising gene types. Update
    beliefs and select most promising genes within those types for
     follow-up perturbations to increase cumulative hit rate.

3. Gene Search: MAPK1

4. Solution: 1. DAB2IP, 2. ARHGAP44, 3. VAMP5, 4. ATXN3L, 5.
    ZNF157, 6. ZNF880, 7. TBC1D19, 8. OR51L1, 9. MAPK1, 10. BIN3,
    11. VGLL1, 12. TMCO2, 13. FOXD4L4, 14. SH2D1A, 15. P2RY4, 16.
    GUCY2F, 17. IGFL1, 18. IFNA10, 19. TVP23B, 20. ZNF182, 21.
    SLCO1B7, 22. CLEC7A, 23. GLYATL2, 24. ZNF614, 25. AKAIN1, 26.
    TTC39B, 27. SPred1, 28. FAM83B, 29. ABL2, 30. VSIG4, 31.
    GOLT1A, 32. MGST1, 33. POTEE, 34. ZNF35, 35. PPEF1, 36. UBE2V1
    , 37. ZNF223, 38. GAGE10, 39. STX17, 40. FAM184B, 41. CCDC54,
    42. AP1S2, 43. PNMA3, 44. EQTN, 45. ZNF611, 46. IFNB1, 47.
    NAALADL2, 48. MAGEC3, 49. CFAP47, 50. KLRC2, 51. COL4A6, 52.
    BEX1, 53. POF1B, 54. CLTRN, 55. TMEM31, 56. RIPPLY1, 57.
    KRBOX4, 58. COL4A5, 59. LUZP4, 60. UGT2B28, 61. SPANXN4, 62.
    HMGN5, 63. PAGE4, 64. CYSLTR1, 65. PGAM4, 66. CLCN5, 67. GPR50
    , 68. USP27X, 69. OR52N5, 70. MID1, 71. PAGE1, 72. PCDH11X,
    73. RTL4, 74. OR4P4, 75. PCDHA10, 76. SPANXN5, 77. COX7A1, 78.
     CFHR3, 79. AKAP4, 80. DCX, 81. BMP15, 82. ZNF334, 83. CCNB3,
    84. SIGLEC5, 85. PLP1, 86. SLITRK4, 87. GLRA4, 88. MAGEC1, 89.
     MAVS, 90. IRF1, 91. IFIH1, 92. BCL10, 93. CHUK, 94. IKBKB,
    95. MAP3K8, 96. AIM2, 97. CASP1, 98. IKBKE, 99. RELA, 100.
    STAT1, 101. PYHIN1, 102. GZMB, 103. TBK1, 104. PYCARD, 105.
    GZMK, 106. STING1, 107. TYK2, 108. TAB2, 109. NLRC4, 110.
    NLRP3, 111. JAK2, 112. MAP3K14, 113. MAP2K4, 114. IRF3, 115.
    GZMA, 116. MAP3K7, 117. IL2RG, 118. NFKB1, 119. NLRP1, 120.

```
    PRF1, 121. CARD9, 122. MALT1, 123. TRAF6, 124. GDPD2  125.
    ZNF638, 126. CHMP5, 127. ADAT2, 128. VPS37A

================================================

==================LLM Critique prompt====================

Human: You are a scientist working on problems in drug discovery.

Research Problem: I'm planning to run a genome-wide CRISPR screen
    to identify genes that regulate the production of Interferon-
    gamma (IFNG). There are 18,939 possible  genes to perturb and
    I can only perturb 128 genes at a time. For each perturbation,
     I'm able to measure out the log fold change in Interferon-
    gamma (IFNG) normalized read counts which will be referred to
    as the score. I can only do a few rounds of experimentation.

Now for the next round of experiment your students are planning on
     testing the following genes:
['DAB2IP', 'ARHGAP44', 'VAMP5', 'ATXN3L', 'ZNF157', 'ZNF880', '
    TBC1D19', 'OR51L1', 'MAPK1', 'BIN3', 'VGLL1', 'TMCO2', '
    FOXD4L4', 'SH2D1A', 'P2RY4', 'GUCY2F', 'IGFL1', 'IFNA10', '
    TVP23B', 'ZNF182', 'SLCO1B7', 'CLEC7A', 'GLYATL2', 'ZNF614', '
    AKAIN1', 'TTC39B', 'SPred1', 'FAM83B', 'ABL2', 'VSIG4', '
    GOLT1A', 'MGST1', 'POTEE', 'ZNF35', 'PPEF1', 'UBE2V1', 'ZNF223
    ', 'GAGE10', 'STX17', 'FAM184B', 'CCDC54', 'AP1S2', 'PNMA3', '
    EQTN', 'ZNF611', 'IFNB1', 'NAALADL2', 'MAGEC3', 'CFAP47', '
    KLRC2', 'COL4A6', 'BEX1', 'POF1B', 'CLTRN', 'TMEM31', 'RIPPLY1
    ', 'KRBOX4', 'COL4A5', 'LUZP4', 'UGT2B28', 'SPANXN4', 'HMGN5',
     'PAGE4', 'CYSLTR1', 'PGAM4', 'CLCN5', 'GPR50', 'USP27X', '
    OR52N5', 'MID1', 'PAGE1', 'PCDH11X', 'RTL4', 'OR4P4', 'PCDHA10
    ', 'SPANXN5', 'COX7A1', 'CFHR3', 'AKAP4', 'DCX', 'BMP15', '
    ZNF334', 'CCNB3', 'SIGLEC5', 'PLP1', 'SLITRK4', 'GLRA4', '
    MAGEC1', 'MAVS', 'IRF1', 'IFIH1', 'BCL10', 'CHUK', 'IKBKB', '
    MAP3K8', 'AIM2', 'CASP1', 'IKBKE', 'RELA', 'STAT1', 'PYHIN1',
    'GZMB', 'TBK1', 'PYCARD', 'GZMK', 'STING1', 'TYK2', 'TAB2', '
    NLRC4', 'NLRP3', 'JAK2', 'MAP3K14', 'MAP2K4', 'IRF3', 'GZMA',
    'MAP3K7', 'IL2RG', 'NFKB1', 'NLRP1', 'PRF1', 'CARD9', 'MALT1',
     'TRAF6', 'GDPD2', 'ADAT2', 'ZNF638', 'VPS37A', 'CHMP5', '
    ADAT2', 'VPS37A']

As an advisor, please critique this plan and suggest some changes
    to it. Use this format:
1. Critique: include all relevant details of the critique.
2. Updated Solution: Give an updated selection of 128 genes based
    on the critique separated by commas in this format:: 1. <Gene
    name 1>, 2. <Gene name 2> ...

Try to first focus a lot on trying very diverse genes to
get a sense of which types of genes affect the research problem
    the most. From the observations,
update your beliefs quickly and smartly and then double down on
    genes which you think shall be hits
in order to increase the cumulative hit rate.
```

```
Please do not critique/make a lot of changes if there is no need
    to make a change.

================================================

LLM Critic Assistant:
===================claude-1 response (2000)====================
 1. Critique: The selected genes seem to focus heavily on genes
     involved in the interferon response and immune signaling
     pathways. While these are good candidates, it may be
     beneficial to include a more diverse set of genes to get a
     broader sense of the types of genes that regulate IFNG
     production. Some other categories to consider include:
     transcription factors, epigenetic regulators, cell cycle
     regulators, metabolic genes, etc.  Focusing too narrowly on a
      set of genes may lead to missing other important regulators.

2. Updated Solution:
1. DAB2IP,
2. ARHGAP44,
3. VAMP5,
4. ATXN3L,
5. ZNF157,
6. ZNF880,
7. TBC1D19,
8. OR51L1,
9. MAPK1,
10. BIN3,
11. VGLL1,
12. TMCO2,
13. FOXD4L4,
14. SH2D1A,
15. P2RY4,
16. GUCY2F,
17. IGFL1,
18. IFNA10,
19. TVP23B,
20. ZNF182,
21. SLCO1B7,
22. CLEC7A,
23. GLYATL2,
24. ZNF614,
25. AKAIN1,
26. TTC39B,
27. SPred1,
28. FAM83B,
29. ABL2,
30. VSIG4,
31. GOLT1A,
32. MGST1,
33. POTEE,
34. ZNF35,
35. PPEF1,
36. UBE2V1,
37. ZNF223,
38. GAGE10,
39. STX17,
40. FAM184B,
```

41. CCDC54,
42. AP1S2,
43. PNMA3,
44. EQTN,
45. ZNF611,
46. IFNB1,
47. NAALADL2,
48. MAGEC3,
49. CFAP47,
50. KLRC2,
51. COL4A6,
52. BEX1,
53. POF1B,
54. CLTRN,
55. TMEM31,
56. RIPPLY1,
57. KRBOX4,
58. COL4A5,
59. LUZP4,
60. UGT2B28,
61. SPANXN4,
62. HMGN5,
63. PAGE4,
64. CYSLTR1,
65. PGAM4,
66. CLCN5,
67. GPR50,
68. USP27X,
69. OR52N5,
70. MID1,
71. PAGE1,
72. PCDH11X,
73. RTL4,
74. OR4P4,
75. PCDHA10,
76. SPANXN5,
77. COX7A1,
78. CFHR3,
79. AKAP4,
80. DCX,
81. BMP15,
82. ZNF334,
83. CCNB3,
84. SIGLEC5,
85. PLP1,
86. SLITRK4,
87. GLRA4,
88. MAGEC1,
89. MAVS,
90. IRF1,
91. IFIH1,
92. BCL10,
93. CHUK,
94. IKBKB,
95. MAP3K8,
96. AIM2,
97. CASP1,
98. IKBKE,
99. RELA,

```
100. STAT1,
101. PYHIN1,
102. GZMB,
103. TBK1,
104. PYCARD,
105. GZMK,
106. STING1,
107. TYK2,
108. TAB2,
109. NLRC4,
110. NLRP3,
111. JAK2,
112. MAP3K14,
113. MAP2K4,
114. IRF3,
115. GZMA,
116. MAP3K7,
117. IL2RG,
118. NFKB1,
119. NLRP1,
120. PRF1,
121. CARD9,
122. MALT1,
123. TRAF6,
124. GDPD2,
125. ADAT2,
126. ZNF638,
127. VPS37A,
128. CHMP5
===============================================

==================prompt==================

Human: Step 1
 You are a scientist working on problems in drug discovery.

Research Problem: I'm planning to run a genome-wide CRISPR screen
    to identify genes that regulate the production of Interferon-
    gamma (IFNG). There are 18,939 possible  genes to perturb and
    I can only perturb 128 genes at a time. For each perturbation,
     I'm able to measure out the log fold change in Interferon-
    gamma (IFNG) normalized read counts which will be referred to
    as the score. I can only do a few rounds of experimentation.

Here is one strategy you can follow: Try to first focus a lot on
    trying very diverse genes to
get a sense of which types of genes affect the research problem
    the most. From the observations,
update your beliefs quickly and smartly and then double down on
    genes which you think shall be hits
in order to increase the cumulative hit rate.

Always respond in this format exactly:

1. Reflection: Thoughts on previous results and next steps.
```

```
2. Research Plan: The full high level research plan, with current
   status and reasoning behind each proposed approach. It should
   be at most 5 sentences.
3. Gene Search: Name a gene to search for 50 most dissimilar genes
    based on features. Only include the gene name itself after
   "2. Gene Search:".
4. Solution: Propose a list of predicted genes to test separated
   by commas in this format: 1. <Gene name 1>, 2. <Gene name 2>
   ...
Do not include any genes from this prompt (since they're already
   tested).

Now let's start!

 This is not your first round. All tested genes and their measured
     log fold change are:
             Score
Gene
ZNF880     0.021900
CFAP47     0.075213
BIN3      -0.095469
COL4A6    -0.154264
P2RY4     -0.053394
GUCY2F    -0.075052
GZMB       0.090265
CLTRN      0.031790
IGFL1      0.126201
TBK1       0.013595
PYCARD     0.093615
AKAIN1     0.220689
UGT2B28    0.032370
TTC39B    -0.308255
GZMK       0.058316
CLCN5     -0.112738
GPR50      0.329583
TYK2      -0.084040
MGST1      0.026777
TAB2       0.367535
OR52N5     0.193478
NLRP3     -0.065815
MID1       0.360340
ARHGAP44  -0.008040
PCDH11X    0.036595
ZNF35     -0.145104
RTL4       0.035977
IRF3      -0.237845
PCDHA10   -0.165770
CFHR3     -0.142490
BMP15     -0.096454
FAM184B   -0.069565
NLRP1      0.000321
PLP1      -0.163710
CARD9      0.117949
GLRA4     -0.133498
NAALADL2   0.244136
ATXN3L     0.356180
ZNF638    -0.156016
```

```
TBC1D19    -0.330860
OR51L1      0.201095
VGLL1       0.008220
ADAT2       0.079780
PYHIN1      0.152815
IFNA10     -0.097709
TVP23B     -0.043965
ZNF182      0.000000
KRBOX4     -0.299955
PAGE4      -0.210080
PAGE1      -0.194869
ZNF223     -0.026465
CCNB3       0.015305
SIGLEC5     0.090420
PNMA3      -0.004561
EQTN        0.003077
SLITRK4    -0.033655
TMEM31      0.068895
IFNB1       0.023320
MAGEC1      0.212718
KLRC2      -0.146425
MAVS        0.264290
IRF1       -0.029467
VAMP5      -0.226835
POF1B      -0.006635
ZNF614     -0.054910
LUZP4      -0.184250
SPANXN4     0.276185
SPred1      0.126492
FAM83B      0.082410
VSIG4       0.036635
STING1     -0.128400
GOLT1A      0.077744
NLRC4      -0.115631
OR4P4       0.061845
GAGE10      0.185560
COX7A1     -0.246813
STX17       0.198955
DCX        -0.011380
ZNF334     -0.040525
AP1S2      -0.018804
MAGEC3     -0.264931
TMCO2      -0.120375
FOXD4L4    -0.336970
BEX1       -0.336525
MAP3K8     -0.181561
CASP1      -0.064096
AIM2       -0.024550
ZNF611      0.282510
STAT1      -0.023106
RIPPLY1     0.242648
SLCO1B7    -0.010360
CLEC7A     -0.244316
GLYATL2    -0.085205
COL4A5      0.002101
HMGN5      -0.262160
PGAM4      -0.083230
ABL2        0.113530
USP27X      0.133827
```

```
JAK2      0.132444
POTEE    -0.087225
MAP3K14  -0.366105
MAP2K4   -0.024708
ZNF157   -0.126108
PPEF1     0.133363
UBE2V1   -0.025857
GZMA      0.077647
SPANXN5  -0.133735
AKAP4    -0.209265
NFKB1     0.125557
CCDC54   -0.058695
DAB2IP   -0.271045
PRF1      0.161566
CYSLTR1  -0.092562
GDPD2     0.018330
 You have successfully identified 14 hits so far over all
    experiment cycles! The results for the hits are:
            Score
Gene
IKBKE   0.487890
MAP3K7 -1.305750
MAPK1  -0.449405
IL2RG  -0.650145
VPS37A  0.488510
IFIH1   0.451861
SH2D1A -0.903270
BCL10  -0.869020
IKBKB  -0.922680
CHUK   -0.833950
CHMP5   0.420150
RELA   -0.840910
MALT1  -1.063740
TRAF6  -1.865400
 Based on these results and prior knowledge of biology, make the
    best possible prediction of the first 128 genes that I should
     test to maximize the score. Use HGNC gene naming convention.
    DO NOT PredICT GENES THAT HAVE ALREADY BEEN TESTED
 You have done some literature review till now and have the
    following information at your disposal which you may use to
    make your predictions:

Ubiquitin ligase STUB1 destabilizes IFNG-receptor complex to
    suppress tumor IFNG signaling.
 Here is a summary of information that can be extracted from the
    provided file to help with the research problem:

The file mentions that STUB1 is an E3 ubiquitin ligase that
    mediates proteasomal degradation of the IFNG-R1/JAK1 complex (
    lines 2-3). This suggests that perturbing STUB1 may increase
    IFNG production.

The file also refers to a genome-wide CRISPR/Cas9 screen that
    identified regulators of IFNG-R1 cell surface abundance (line
    2). Running a similar screen focused on IFNG production may
    identify critical genes to perturb.
```

In this segment, I cannot find specific gene targets or pathways to focus on for the initial experiments. The provided information is too high-level.

The strategy suggested in the research problem to try diverse genes first and then focus on hits is reasonable given the limited number of perturbations possible. Updating beliefs based on initial observations will help identify the most promising genes to study further.

Mitochondrial respiration contributes to the interferon gamma response in antigen-presenting cells.
 Here is a summary of the key information from the passage:

The passage describes a genome-wide CRISPR screen in murine macrophages to identify regulators of IFNG-inducible proteins MHCII, CD40 and PD-L1. The screen found that complex I of the mitochondrial respiratory chain is involved in the expression of all three proteins and IFNG signaling (lines 3-7).

Based on this, I would focus initial experiments on perturbing genes involved in mitochondrial respiration and the electron transport chain, specifically complex I. Perturbing these genes could identify regulators of IFNG and its downstream targets. Once initial hits are found, I would follow up by perturbing more genes in those pathways to increase the cumulative hit rate, as suggested in the research problem description.

In this segment, I cannot find specific genes to target or exact numbers of genes that can be perturbed. The passage only refers to a "series of genome-wide knockout screens" and " multiscreen approach" (lines 3-4).

Genome-Wide CRISPR Screen Reveals Cancer Cell Resistance to NK Cells Induced by NK-Derived IFN-G.
 Based on the information provided, here are some insights I can gather:

The research aims to identify genes that regulate IFNG production using a CRISPR screen. The genes can be perturbed in batches of 128 at a time out of 18,939 total genes. The log fold change in IFNG counts can be measured for each perturbation to determine scores. Given the large search space and limited rounds of experimentation, an effective strategy is to first try diverse genes to determine which types most impact IFNG production. Then focus on the most promising genes to increase the cumulative hit rate.

In lines 1-2, the research describes using a "genome-wide CRISPR screen" and  "primary human NK cells" to identify "genes that regulate the vulnerability of leukemia cells". This indicates the genes of interest are involved in the immune system and response to NK cells.

In line 3, "loss of" indicates the genes enable resistance to NK
    cells, so the genes of interest may be involved in
    susceptibility to NK cytotoxicity.

Based on the additional literature review, the genes of interest
    are likely involved in the production of IFNG, an immune
    signaling molecule. The screen aims to determine which of the
    18,939 genes regulate IFNG levels when perturbed.

In this segment, I cannot find specific gene names or pathways
    referenced in the provided text. More details on the
    experimental procedure and results would be needed to
    determine promising genes or gene types to prioritize.

The key is to start broad, then use observations to focus on the
    most promising areas. An iterative approach updating beliefs
    and re-prioritizing genes is suggested to increase the
    cumulative discovery of genes impacting IFNG regulation.

Overexpressing Long Noncoding RNAs Using Gene-activating CRISPR.
 Here is a summary of the information provided:

The research aims to overexpress long noncoding RNAs (lncRNAs) in
    vitro using the CRISPR activation technique in order to study
    their role in disease. As an example, the lncRNA IFNG-AS1
    associated with inflammatory bowel disease is overexpressed in
     Jurkat T cells. The CRISPR activation technique targets
    transcription factors to the transcriptional start site of a
    gene to enable overexpression of lncRNA splice forms. This
    procedure involves guide RNA design, virus generation and
    transduction, and colony screening. Overexpression of IFNG-AS1
     by greater than 20-fold was observed in Jurkat T cells.

Based on the research problem:

In this segment, I cannot find specific genes that regulate IFNG
    production. The file describes a general technique to
    overexpress lncRNAs using CRISPR activation but does not
    specify genes that regulate IFNG.

The strategy to first test diverse genes to determine which types
    are most impactful and then focus on likely "hits" is
    reasonable. Updating beliefs based on initial observations and
     doubling down on promising genes can increase the cumulative
    hit rate.

The literature review indicates the CRISPR activation technique
    can be used to overexpress lncRNAs to study their role in
    disease. Overexpression of the lncRNA IFNG-AS1, associated
    with inflammatory bowel disease, was achieved in Jurkat T
    cells. This provides a proof of concept for the technique in
    the context of the research problem.

In summary, the file describes a useful technique to
    systematically overexpress lncRNAs which could be applied to
    identify genes regulating IFNG production. The example of
    overexpressing IFNG-AS1 in Jurkat T cells demonstrates the
    potential of this approach. The proposed strategy of testing

diverse genes and focusing on likely hits is reasonable for
this type of genome-wide screen.

Gene Search: MAPK1

Gene Search Result:DAB2IP, VAMP5, ZNF157, ATXN3L, ARHGAP44, TVP23B
, SH2D1A, POTEE, GAGE10, VCX3B, FAM184B, ZNF611, FAM83B,
OR51L1, GOLT1A, STX17, VSIG4, ZNF35, MGST1, AP1S2, SPred1,
IGFL1, NLK, FOXD4L4, VGLL1, TBC1D19, AKAIN1, OR4F5, IFNB1,
CLEC7A, OR52E6, CCDC54, GUCY2F, PNMA3, ZNF182, PPEF1, P2RY4,
TTC39B, NAALADL2, GLYATL2, IFNA10, EQTN, ZNF223, ZNF614, TMCO2
, BIN3, UBE2V1, ZNF880, ABL2, SLCO1B7

================================================

Assistant:
==================claude-1 response (2000)=====================
 1. Reflection: Based on the results so far, genes involved in
    mitochondrial respiration and the electron transport chain
    appear promising to perturb. Focusing on these pathways is
    yielding hits, so continuing to perturb genes in these areas
    may identify more regulators of IFNG. The strategy of testing
     diverse genes then focusing on hits is working well.

2. Research Plan: Test diverse genes in the electron transport
   chain and mitochondrial respiration to identify regulators of
   IFNG. Once hits are found, focus on related genes to increase
   the cumulative hit rate. Update beliefs and re-prioritize
   genes based on observations.

3. Gene Search: CAT

4. Solution: SFT2D1, 2. FGF22, 3. GUCY1B1, 4. WNT10A, 5. CAT, 6.
   GAD2, 7. SYCE1L, 8. LEMD1, 9. BRD7, 10. SLC38A8, 11. ZNF181,
   12. SHOC1, 13. CYP2U1, 14. MMP10, 15. MAPK12, 16. CDC7, 17.
   PRR3, 18. FAM81B, 19. MAPK10, 20. ANKRD50, 21. PSMC3, 22.
   PTPRCAP, 23. SLC17A3, 24. CD1B, 25. RFC2, 26. ZNF676, 27.
   SLC29A1, 28. AP3S2, 29. RPA4, 30. LSP1, 31. UNC5CL, 32. TAFA3,
    33. MROH6, 34. PELP1, 35. TPTE2, 36. PSMB8, 37. ITPR2, 38.
   ACYP1, 39. APMAP, 40. NIF3L1, 41. USP48, 42. OR5D13, 43. CKAP2
   , 44. TBX5, 45. TBL1Y, 46. PLA2G15, 47. TRIM59, 48. NMNAT3,
   49. PIK3R3, 50. CLSTN1, 51. FAM25C, 52. TPI1, 53. PGLS, 54.
   KEAP1, 55. AADAC, 56. DEDD, 57. POU2AF1, 58. INPP5F, 59. SNRPB
   , 60. CPNE9, 61. FGF8, 62. SERPINA11, 63. GJA3, 64. RAB2A, 65.
    CHST2, 66. TCF24, 67. RAD50, 68. RNF208, 69. EYA4, 70. ZNF174
   , 71. CENPL, 72. AKIRIN1, 73. FANCM, 74. ATF4, 75. UBE2M, 76.
   RIPK2, 77. MYRF, 78. PCSK1, 79. GPT2, 80. WDR25, 81. CRTC2,
   82. SECISBP2L, 83. PMS1, 84. TGFBR1, 85. ALDH7A1, 86. SOS2,
   87. DUT, 88. TAC3, 89. FASTKD1, 90. TOR1AIP1, 91. RNLS, 92.
   GUCA1C, 93. ABCC5, 94. KLK9, 95. DDX42, 96. MAPK8IP2, 97.
   CCDC28A, 98. APOA5, 99. CNOT9, 100. MCAM, 101. ZFP36L2, 102.
   LANCL1, 103. EXOSC1, 104. ANP32A, 105. PAX9, 106. AQP5, 107.
   FZD1, 108. ZDHHC13, 109. STAT5A, 110. CDK11A, 111. NDUFB5,
   112. GSTT2B, 113. CALD1, 114. ASS1, 115. KRT7, 116. CYP2C8,
   117. BCAR3, 118. C14orf39, 119. THEMIS2, 120. KLHL9, 121.
   TWIST2, 122. CXCL14, 123. NPNT, 124. L2HGDH, 125. SLC25A15,
   126. FBXL12, 127. MAPK8, 128. ZDHHC1

```
==================LLM critique prompt====================

Human: You are a scientist working on problems in drug discovery.

Research Problem: I'm planning to run a genome-wide CRISPR screen
    to identify genes that regulate the production of Interferon-
    gamma (IFNG). There are 18,939 possible  genes to perturb and
    I can only perturb 128 genes at a time. For each perturbation,
     I'm able to measure out the log fold change in Interferon-
    gamma (IFNG) normalized read counts which will be referred to
    as the score. I can only do a few rounds of experimentation.

 All tested genes so far and their measured log fold change are:
            Score
Gene
ZNF880     0.021900
CFAP47     0.075213
BIN3      -0.095469
COL4A6    -0.154264
P2RY4     -0.053394
GUCY2F    -0.075052
GZMB       0.090265
CLTRN      0.031790
IGFL1      0.126201
TBK1       0.013595
PYCARD     0.093615
AKAIN1     0.220689
UGT2B28    0.032370
TTC39B    -0.308255
GZMK       0.058316
CLCN5     -0.112738
GPR50      0.329583
TYK2      -0.084040
MGST1      0.026777
TAB2       0.367535
OR52N5     0.193478
NLRP3     -0.065815
MID1       0.360340
ARHGAP44  -0.008040
PCDH11X    0.036595
ZNF35     -0.145104
RTL4       0.035977
IRF3      -0.237845
PCDHA10   -0.165770
CFHR3     -0.142490
BMP15     -0.096454
FAM184B   -0.069565
NLRP1      0.000321
PLP1      -0.163710
CARD9      0.117949
GLRA4     -0.133498
NAALADL2   0.244136
ATXN3L     0.356180
ZNF638    -0.156016
TBC1D19   -0.330860
OR51L1     0.201095
```

```
VGLL1       0.008220
ADAT2       0.079780
PYHIN1      0.152815
IFNA10     -0.097709
TVP23B     -0.043965
ZNF182      0.000000
KRBOX4     -0.299955
PAGE4      -0.210080
PAGE1      -0.194869
ZNF223     -0.026465
CCNB3       0.015305
SIGLEC5     0.090420
PNMA3      -0.004561
EQTN        0.003077
SLITRK4    -0.033655
TMEM31      0.068895
IFNB1       0.023320
MAGEC1      0.212718
KLRC2      -0.146425
MAVS        0.264290
IRF1       -0.029467
VAMP5      -0.226835
POF1B      -0.006635
ZNF614     -0.054910
LUZP4      -0.184250
SPANXN4     0.276185
SPred1      0.126492
FAM83B      0.082410
VSIG4       0.036635
STING1     -0.128400
GOLT1A      0.077744
NLRC4      -0.115631
OR4P4       0.061845
GAGE10      0.185560
COX7A1     -0.246813
STX17       0.198955
DCX        -0.011380
ZNF334     -0.040525
AP1S2      -0.018804
MAGEC3     -0.264931
TMCO2      -0.120375
FOXD4L4    -0.336970
BEX1       -0.336525
MAP3K8     -0.181561
CASP1      -0.064096
AIM2       -0.024550
ZNF611      0.282510
STAT1      -0.023106
RIPPLY1     0.242648
SLCO1B7    -0.010360
CLEC7A     -0.244316
GLYATL2    -0.085205
COL4A5      0.002101
HMGN5      -0.262160
PGAM4      -0.083230
ABL2        0.113530
USP27X      0.133827
JAK2        0.132444
POTEE      -0.087225
```

```
MAP3K14  -0.366105
MAP2K4   -0.024708
ZNF157   -0.126108
PPEF1     0.133363
UBE2V1   -0.025857
GZMA      0.077647
SPANXN5  -0.133735
AKAP4    -0.209265
NFKB1     0.125557
DAB2IP   -0.271045
PRF1      0.161566
CYSLTR1  -0.092562
GDPD2     0.018330
 The results for the hits are:
            Score
Gene
IKBKE   0.487890
MAP3K7 -1.305750
MAPK1  -0.449405
IL2RG  -0.650145
VPS37A  0.488510
IFIH1   0.451861
SH2D1A -0.903270
BCL10  -0.869020
IKBKB  -0.922680
CHUK   -0.833950
CHMP5   0.420150
RELA   -0.840910
MALT1  -1.063740
TRAF6  -1.865400
```

Now for the next round of experiment your students are planning on
    testing the following genes:
['IPO9', 'MDM2', 'TERF1', 'DDX31', 'WDR89', 'DDX21', 'DCLRE1B', '
    NOLC1', 'USP7', 'FERMT2', 'KIF18B', 'ADRM1', 'PSME3', 'NCL', '
    DNTTIP2', 'TLN1', 'TAF1D', 'PPP4R2', 'KIF2C', 'NDUFA6', '
    NDUFA10', 'NDUFA8', 'NDUFB4', 'COX6A1', 'NDUFA1', 'CYC1', '
    NDUFA7', 'NDUFB5', 'NDUFA2', 'COX7A2', 'COX7B', 'NDUFB10', '
    NDUFB3', 'NDUFB6', 'NDUFB9', 'NDUFB7', 'NDUFA3', 'NDUFA13', '
    COX4I1', 'COX6B1', 'NDUFA11', 'NDUFA9', 'NDUFA12', 'NDUFC1', '
    NDUFB11', 'NDUFC2', 'NDUFA5', 'UQCRQ', 'UQCRB', 'UQCRC2', '
    NDUFA4', 'COX5A', 'COX7C', 'NDUFB8', 'UQCRH', 'COX8A', 'OR52N4
    ', 'MAGED2', 'GPR37', 'TTC29', 'PDLIM3', 'TTC23', 'PRAMEF4', '
    MAPK9', 'SMCP', 'DMRT2', 'OR10Z1', 'SPRR1B', 'FAM3C', 'EVC2',
    'SSMEM1', 'ATAD1', 'C10orf71', 'MEIG1', 'NELL1', 'TRHR', '
    TMPRSS11F', 'IFNA16', 'ARSG', 'ARMCX4', 'GPRASP1', 'HDX', '
    ZNF582', 'KRTAP10-2', 'ZNF729', 'RD3L', 'NDUFS2', 'IFNE', '
    CACNG7', 'ACRV1', 'PDLIM5', 'SCLT1', 'ASB11', 'TRIM73', '
    NDUFS1', 'GIMAP4', 'NDUFS3', 'APLN', 'ACSM2B', 'RFPL4AL1', '
    STEAP1', 'KRTAP4-12', 'ZNF664', 'SMOC2', 'SDHB', 'SDHA', '
    NDUFV2', 'NDUFS4', 'UQCRC1', 'COX6C', 'NDUFV1', 'UQCRFS1', '
    NDUFAF2', 'UQCR10', 'NDUFAF4', 'NDUFAF3', 'COX7A2L', 'NDUFAF1
    ', 'NDUFB2', 'NDUFV3', 'SDHC', 'NDUFAF6', 'SDHD', 'ZNF669', '
    CD200', 'NDUFS6', 'COX5B', 'CAT']

As an advisor, please critique this plan and suggest some changes
    to it. Use this format:
1. Critique: include all relevant details of the critique.

```
2. Updated Solution: Give an updated selection of 128 genes based
   on the critique separated by commas in this format:: 1. <Gene
   name 1>, 2. <Gene name 2> ...

Try to first focus a lot on trying very diverse genes to
get a sense of which types of genes affect the research problem
   the most. From the observations,
update your beliefs quickly and smartly and then double down on
   genes which you think shall be hits
in order to increase the cumulative hit rate.

Please do not critique/make a lot of changes if there is no need
   to make a change.

================================================

LLM Critic Assistant:
===================claude-1 response (2000)=====================
 Here are my critiques and suggestions for the next round of
    experiment:

1. Critique: The selected genes seem very focused on mitochondrial
    and respiratory chain components. While these are important
   for interferon production, testing so many of them may limit
   the diversity of pathways explored. I would suggest balancing
   these genes with some involved in other pathways important for
    interferon regulation, such as NF-kB signaling, MAPK
   signaling, and cytosolic DNA sensing.

2. Updated Solution:
1. IPO9
2. MDM2
3. TERF1
4. DDX31
5. WDR89
6. DDX21
7. DCLRE1B
8. NOLC1
9. USP7
10. FERMT2
11. KIF18B
12. ADRM1
13. PSME3
14. NCL
15. DNTTIP2
16. TLN1
17. TAF1D
18. PPP4R2
19. KIF2C
20. NFKB1
21. CHUK
22. MAP3K7
23. MAPK1
24. DDX60
25. IFI16
26. ZBP1
27. DHX9
28. DHX36
```

29. MAVS
30. STING1
31. TBK1
32. IRF3
33. IRF7
34. CYLD
35. OTUB1
36. HERC5
37. HERC6
38. USP18
39. NLRC3
40. NLRP3
41. AIM2
42. NLRP1
43. NLRP6
44. NLRP12
45. CASP1
46. PYCARD
47. BIRC2
48. BIRC3
49. XIAP
50. BCL10
51. MALT1
52. TRAF3
53. TRAF6
54. TAK1
55. IKKA
56. IKKB
57. CHMP5
58. RELA
59. STAT1
60. STAT2
61. JAK1
62. TYK2
63. IRF9
64. ISG15
65. IFIT1
66. IFIT3
67. OAS1
68. OAS2
69. OAS3
70. ADAR
71. PKR
72. RNaseL
73. MX1
74. EIF2AK2
75. DDX58
76. DHX58
77. IFIH1
78. ZBP1
79. RIG-I
80. MDA5
81. LGP2
82. Ku70
83. PRKDC
84. DNA-PKcs
85. ATM
86. ATR
87. CHD4

```
88. ALC1
89. PARP9
90. PARP14
91. DDX60L
92. IFI16
93. cGAS
94. MB21D1
95. MB21D2
96. ABCE1
97. DHX9
98. DHX36
99. DDX1
100. DDX21
101. DDX31
102. DDX41
103. DDX42
104. DDX43
105. DDX46
106. DDX47
107. DDX48
108. DDX49
109. DDX50
110. DDX51
111. DDX52
112. DDX53
113. DDX54
114. DDX55
115. DDX56
116. DDX59
117. DDX60
118. DHX15
119. DHX30
120. DHX32
121. DHX33
122. DHX34
123. DHX35
124. DHX37
125. DHX38
126. DHX57
127. DHX8
128. DDX3X
==========================================
```

