# OpenReview forum: "BioDiscoveryAgent: An AI Agent for Designing Genetic Perturbation Experiments"
_ICLR.cc/2025/Conference — ICLR 2025 Poster_

### Official Review · Reviewer_biwg · 2024-10-31

**Soundness:** 4
**Presentation:** 4
**Contribution:** 3
**Rating:** 8
**Confidence:** 4

**Summary:**

The authors propose a new, LLM-based approach for designing gene perturbation experiments, where the goal is to efficiently select a subset of genes that result in a desired phenotype when perturbed. Their approach is based on the creation of BioDiscoveryAgent, an LLM-based agent that can generate, critique, and refine hypotheses about which genes should be selected. They show that BioDiscoveryAgent achieves a large performance improvement over baselines on the task of predicting relevant genetic perturbations across six datasets.

**Strengths:**

1. **Significant contribution to an important problem.** The problem of efficiently designing genetic perturbation experiments is important, with wide-reaching applications to drug target discovery, gene therapy, immunotherapy, etc. The authors’ experiments show that their proposed method yields a considerable performance improvement over baselines on the task of identifying relevant gene perturbations.
2. **Novel application of LLMs.** While using LLMs for experiment design is not a new idea, to my knowledge, it has not yet been explored for the design of gene perturbation experiments. The author’s approach is considerably different from existing approaches for this task, which primarily focus on Bayesian optimization.
3. **Experiments are thorough and produce interesting insights.** The authors not only show that their method obtains strong performance, they also conduct a series of experiments that shed light on why their method is able to perform well. In particular, the experiments that examine the similarity of the predicted genes across datasets and across experimental rounds are very interesting. The prompt/observation ablations are also very helpful in understanding which parts of the method contribute to the strong performance.
4. **Clear presentation.** The paper is clearly written. The motivation, contribution, and experiment design are all clearly and thoroughly described. It is written in a way such that it is understandable even for someone with a limited background in biology.

**Weaknesses:**

1. **Lack of technical/methodological novelty**: The two main parts of the method are (1) task-specific prompts and (2)  an algorithm that involves repeatedly calling an LLM with prompts that are adapted based on the outputs at previous steps. This basic process is shared by other work on LLM agents. However, I don’t consider this a major weakness, since the goal of the paper is to design an approach that works well for gene perturbation experiments, and the authors’ approach both works well and is novel within this context.
2. **Claims about interpretability seem overstated**: The authors claim multiple times that their approach offers the benefit of interpretability. For example, they claim that by having the model output a “research plan” and “reflection” in addition to the "solution", this can help to increase interpretability and “rule out predictions that may be hallucinations or not well-motivated.” The problem with this logic is that it assumes that the model’s stated reasoning/reflections are indicative of the true reasoning the model used to arrive at its solutions. However, prior work has shown that an LLM’s stated reasoning/explanations may be misleading/unfaithful to its true decision making process (e.g., [1]). In light of these findings, the author’s claims about interpretability seem overstated to me. I think the paper would be strengthened if the authors discussed the risk that these “reflection” outputs could be misleading. In addition, the authors claim that the LLM’s use of the literature search tool enhances interpretability. However, if the LLM incorrectly summarizes an article or hallucinates information that is not present in the article, this could actually harm interpretability (because it would be misleading). While the user could go to the article and check the correctness of LLM summaries, this is a time-consuming process, so it may not be realistic to expect a user to do this.

[1] Turpin, Miles, et al. "Language models don't always say what they think: unfaithful explanations in chain-of-thought prompting." Advances in Neural Information Processing Systems 36 (2024).

**Questions:**

1. Why don’t you include all baselines in the 2-gene perturbation experiments (i.e., why do you only compare to random sampling)?

---

> ### Author Response · Authors · 2024-12-02
> **Novelty of our approach**
>
> We are very grateful to the reviewer for their constructive feedback and provide some comments below
>
> > Technical/methodological novelty:
>
> We thank the reviewer for recognizing the novelty of our contribution.
>
> - Our approach intentionally emphasizes simplicity to maximize robustness and generalizability. We view this simplicity as a strength, particularly given one of the core advantages of using large language models: the ability to employ intuitive and interpretable natural language-based prompts.
> - Moreover, the simplicity of our approach is also what enables its broad generalization with no direct training. Our approach is not specifically trained on the data that it is applied on, yet it still performed roughly 21% and 47% better than other baselines on the same metrics.
> - While we acknowledge that there are numerous ways to further improve the agent’s performance, these do not detract from the core argument of our paper: establishing the value of large language models as experiment design agents capable of informed decision-making based on both existing knowledge and new evidence.
>
> Finally, we also present a tangible way forward for further enhancing these agents. We believe that the most effective approach to augment the agent’s performance without compromising on its generalizability is through the use of tools. Tools enhance capabilities without requiring fine-tuning or re-training that can reduce the robustness of the model to new settings. We designed several new tools and include a detailed analysis of the benefit of their use across different LLM backbones. We highlight cases where tools consistently improve performance and others where this is not the case. We present a novel relationship between the benefit of tool use and the size of the LLM.

---

> ### Author Response · Authors · 2024-12-02
> **Interpretability**
>
> We thank the reviewer for the very useful and constructive feedback on how to validate and make use of model interpretability.
>
> > Claims about interpretability seem overstated... I think the paper would be strengthened if the authors discussed the risk that these “reflection” outputs could be misleading.
>
> We agree that language models are able to hallucinate and produce outputs that are not correct. However, as a tool for augmenting interpretability in the setting of experiment design this may not necessarily be a deal breaker. The purpose of providing an explanation behind model reasoning in this scenario is to guide the researcher on potential directions for further exploration and validation as opposed to providing the right answer to the researcher’s problem. We agree that it would be beneficial to clarify this in the text.
>
> **Keyword frequency experiment**
>
> We performed a quick analysis of the keywords from the model reasoning text to test whether the responses were meaningful or not:
> - Ran BioDiscoverAgent 30 times using Claude v1 for the Schmidt (IL2) dataset
> - Combined the model reasoning logs (Reflection, Research Plan etc.)
> - Removed all gene names and scores
> - Identified 50 keywords from each run (using word frequency)
> - Linked each word to hit ratio at the tenth round
> - Computed the mean hit ratio for each word across all runs where it was used.
>
> Interestingly, the top 10 terms, ranked by hit ratio, were enriched for concepts directly related to MAPK signaling and immune signaling, both closely tied to IL-2 function. Interestingly, terms related to inflammation, another aspect of IL-2 signaling, were associated with lower hit ratios.
>
> While this does not prove that the model reasoning was always correct, it does show that a relationship exists between the different strategies that the agent takes and the reasoning that is generated, and that it is not random.
>
> | Word            | Score           |
> |-----------------|-----------------|
> | **mapk**        | 0.100152905199 |
> | **cytokine**    | 0.098623853211 |
> | **transcription** | 0.095565749235 |
> | **regulators**  | 0.089194699286 |
> | **factors**     | 0.084097859327 |
> | **signal**      | 0.077981651376 |
> | **tranduction** | 0.077981651376 |
> | **regulation**  | 0.076889471385 |
> | **immune**      | 0.074705111402 |
> | **signaling**   | 0.071865443425 |
> | gene            | 0.071738022426 |
> | genes           | 0.070689249588 |
> | protein         | 0.070336391437 |
> | kinases         | 0.070336391437 |
> | treg            | 0.070336391437 |
> | regulatory      | 0.070336391437 |
> | fold            | 0.070336391437 |
> | kinase          | 0.070336391437 |
> | pathway         | 0.069189602446 |
> | ephrinb         | 0.067278287462 |
> | activation      | 0.066513761468 |
> | expression      | 0.066131498471 |
> | cell            | 0.065888240200 |
> | immunology      | 0.065749235474 |
> | genetic         | 0.064220183486 |
> | regulator       | 0.064220183486 |
> | receptor        | 0.064220183486 |
> | inflammatory    | 0.064220183486 |
> | resting         | 0.063455657492 |
> | transcriptome   | 0.062691131498 |
> | prkcch          | 0.061162079511 |
> | inflammation    | 0.059633027523 |
> | modulating      | 0.059633027523 |
>
>
>
> > If the LLM incorrectly summarizes an article or hallucinates information that is not present in the article, this could actually harm interpretability
>
> While we agree that it is possible for the agent to hallucinate when referencing the literature, we counteract this by prompting the agent to provide not just literature citations but also line numbers from those specific papers. For instance, in Appendix G, lines 1616-1628: the agent provides specific line numbers for all its references
>
> >The file mentions that STUB1 is an E3 ubiquitin ligase that mediates proteasomal degradation of the IFNG-R1/JAK1 complex ( lines 2-3)
>
> >The file also refers to a genome-wide CRISPR/Cas9 screen that identified regulators of IFNG-R1 cell surface abundance (line 2).

---

> ### Author Response · Authors · 2024-12-02
> **Additional 2-gene baselines**
>
> > Why don’t you include all baselines in the 2-gene perturbation experiments (i.e., why do you only compare to random sampling)?
>
> We thank the reviewer for their question.
>
> - **No combinatorial baseline**: The methods that we compare against for 2-gene perturbation prediction all make use of a feature vector for each individual gene. There is no direct solution for how to adapt these methods to represent a combination of genes to better enable the downstream 2-gene perturbation task. While the acquisition functions used in baseline methods could be repurposed, we would have to create a new baseline model for handling 2-gene representations as well as design an approach for representing 2 genes.
> - **Learning combinatorial effect from few samples very challenging**: Additionally, unlike single-gene based models, this approach would need to account for interaction phenotypes between genes. Effectively learning these combinatorial interaction effects is highly challenging with the limited number of samples available—in our case: 160 samples collected across 5 rounds of experimentation. Given the approximately 100,000 possible 2-gene combinations, sampling from such a vast space is unlikely to yield enough strong interaction phenotypes for effective model training.

---

> > ### Comment · Reviewer_biwg · 2024-12-03
> >
> > I thank the authors for the time and effort spent responding to my review. My score remains the same.

---

### Official Review · Reviewer_WoKr · 2024-11-02

**Soundness:** 3
**Presentation:** 3
**Contribution:** 2
**Rating:** 6
**Confidence:** 3

**Summary:**

This paper presents an LLM-based agent to design gene perturbation experiments for drug targeting, using the LLM's inherent knowledge, previous observations and external tools to improve hit rates over various baselines.

**Strengths:**

1. The motivation is clear, and the method design seems intuitive;
2. The experiments are thourough, including different LLMs, both seen and unseen datasets, single and double gene settings, and different tool utilization. The results with the strongest LLM shows a large margin over traditional methods, supporting the effectiveness of their approach.
3. Detailed prompts were provided in the Appendix, which is good for reproducibility.
4. The writing is clear and easy to follow.

**Weaknesses:**

1. Given the application scenario of the problem, most of the prompting techniques (e.g. to include previous outcomes, to reflect and plan for the next step) are quite intuitive. I do not find anything particularly novel in this part. In particular, you included the literal representation of the experiment outcomes into the prompt for the agent's observation, which is basically an array of numerial data. I wonder if this is the best way to do it.
2. The tool utilization techniques (e.g. searching for genes in the same pathway, searching PubMed articles) are reasonable, but they sometimes does not improve or even hurts the performance. This is concerning when happening with strong LLMs such as Claude 3.5 Sonnet, if we want to explore an upper bound on the abilities of the agent. Moreover, there lacks an explanation why the effects of tools differs with LLMs. From an application perspective, it would be good to predict when using tools helps and when it does not.
3. There lacks an intuitive explanation why using critiques could benefit the agent's predictions, given that the critique is yet another LLM that is not necessary stronger than the actor, and can make mistakes with the same probablity. Unless it is prompted in a way complementary to the actor, or is accessible to some information unavailable to the actor.
4. In general, the paper makes several reasonable attempts to enhance the agent, yet their method does not seem refined enough to contribute substantially to the agent methodology in this application scenario. There also lacks an in-depth discussion on the inconsistency in their empirical findings.

**Questions:**

1. (Same as Weakness 2) Can you provide an intuitive explanation why the effect of using tools differs with LLMs?
2. (Same as Weakness 3) Can you explain a bit the method used by the critique agent, and why you think it may improve performance?
3. You mentioned in the Discussion section that much of the improvements brought by the agent are observed in the early stages, which I think may be related to a better cold start compared to traditional methods. Do you think we can probably use the cold starts from agents to enhance traditional methods?

---

> ### Author Response · Authors · 2024-11-23
> **Simplicity is a strength of our approach**
>
> >Given the application scenario of the problem, most of the prompting techniques (e.g. to include previous outcomes, to reflect and plan for the next step) are quite intuitive. I do not find anything particularly novel in this part. In particular, you included the literal representation of the experiment outcomes into the prompt for the agent's observation, which is basically an array of numerial data.
>
> We thank the reviewer for their feedback. Our approach intentionally emphasizes simplicity to maximize robustness and generalizability. We view this simplicity as a strength, particularly given one of the core advantages of using large language models: the ability to employ intuitive and interpretable natural language-based prompts. Even though our approach is in many ways simpler and more intuitive than the baselines we compare against, it performed roughly 21\% better on the same metrics.
>
> **Core contribution is the novel application of LLM based agents to biological experiment design**: We demonstrate for the first time the use of agents in the closed-loop design of such experiments, highlighting their potential in optimizing the efficiency of CRISPR-based perturbation screens. While our approach is straightforward, it is robust, interpretable, and generalizable. By combining deep biological knowledge with the ability to parse experimental observations, BioDiscoveryAgent significantly outperforms recently published models ([Mehrjou et al., ICLR 2022](https://iclr.cc/virtual/2022/poster/6889) ; [Lyle et al., 2023](https://arxiv.org/abs/2312.04064)) despite not being directly trained for this task.
>
> While we acknowledge that there are numerous ways to further improve the agent’s performance, these do not detract from the core argument of our paper: establishing the value of large language models as experiment design agents capable of informed decision-making based on both existing knowledge and new evidence. Moreover, we believe that the most effective approach to augment the performance of the agent without compromising on its generalizability is through the use of tools. Tools enhance capabilities without requiring fine-tuning or re-training that can reduce the robustness of the model to new settings. Our paper includes a detailed analysis of the benefit of tool use across different LLM backbones.

---

> ### Author Response · Authors · 2024-11-23
> **1. Trends in benefits of tool use, what are their causes, how to get around them**
>
> > The tool utilization techniques are reasonable, but they sometimes does not improve or even hurts the performance...
>
> **Benefit of tool use correlates with size of LLM**: We thank the reviewer for their valuable feedback. We agree that tool use is critical for enhancing the applicability of LLM-based agents in practice. For this reason, we included a detailed breakdown of the benefit of using different tools with different LLMs. Our findings indicate that larger LLMs tend to benefit less from existing tools. In Table 4, we present a novel analysis showing that tool usage does not always improve performance and, intriguingly, appears to inversely correlate with model size (estimated via price per token). To our knowledge, this is the first paper to provide such an analysis.
>
> **For smaller LLMs, tools consistently improve performance** in every case when using all tools, with only 2 exceptions where the performance remains approximately the same. As an example of the consistency of this effect: when using all tools, a smaller LLM, Claude v1 outperforms all machine learning baselines, even though it is unable to do so across all datasets without tools
>
> | Model              | Schmidt1 (All/N-E) | Schmidt2 (All/N-E) | CAR-T (All/N-E) | Scharenberg (All/N-E) | Carnev. (All/N-E) | Sanchez (All/N-E) |
> |---------------------|--------------------|---------------------|-----------------|-----------------------|-------------------|-------------------|
> | BioDiscoveryAgent (Claude v1, All-Tools)  | **0.095** / **0.068**      | **0.122** / **0.114**       | **0.114** / **0.116**   | **0.333** / **0.306**         | **0.054** / **0.042**     | 0.058 / **0.055**     |
> | Coreset            | 0.072 / 0.066      | 0.102 / 0.084       | 0.069 / 0.059   | 0.243 / 0.197         | 0.047 / 0.038     | **0.061** / 0.054     |
>
> > Moreover, there lacks an explanation why the effects of tools differs with LLMs.
>
> **Reasoning**: In order to further address reviewer concerns, we have included in the paper more discussion on why the benefit of tool use varies with LLMs:
>
> - **Larger LLMs may not need tools to identify hit genes only predicted by smaller LLMs when using tools**: To better understand this phenomenon, we investigated the underlying factors. In Appendix Figure 7, we measure the overlap between predictions made by Claude-Haiku with tools and those made by the larger Claude-Sonnet without tools. For 3 out of 6 datasets, we observe that 15% to 27% of the hit genes identified using tools in Claude-Haiku are already predicted natively by Claude-Sonnet without tool use. This suggests that the difference in tool effectiveness between smaller and larger models may stem from the larger models' superior ability to retrieve relevant information independently. In contrast, smaller models like Claude-Haiku and GPT-3 rely more heavily on tools to quickly access and process such information. However, the percentage overlap is still low, highlighting avenues for further performance improvement with better tool design.
>
> - **Model confidence**: Recent work ([Wu et al. NeurIPS 2024](https://arxiv.org/abs/2404.10198)) suggests that a model's likelihood of incorporating retrieved content from a tool is influenced by its confidence in its initial response. Specifically, the less confident a model is, the more likely it is to adopt new information from a tool. For larger models, which tend to exhibit higher confidence in their responses, tools may need to be augmented to improve their ability to influence these models and increase the likelihood of their adopting new information.
>
> - **Tools show great promise for further improvement in agent performance**: Overall, our findings led us to state in line 431 that the limitation lies not in the models themselves but in the tools, which must be adapted and enhanced to better complement larger LLMs.  Indeed, our results present clear justification for why it would be beneficial to devote further research into the development of tools that are well-suited for larger models and why there is a high likelihood for further improvements in model performance.
>
> >From an application perspective, it would be good to predict when using tools helps and when it does not.
>
> **Recommendations**: We have updated the paper to include the following recommendations:
>
> - When using lower cost LLMs like Claude v1, Claude 3 Haiku and GPT3.5, tool use consistently improves performance. Based on Appendix Table 10, the all-tools setting should give the best result.
> - When using larger LLMs like Claude 3.5 Sonnet or GPT-o1-mini, then tool use can hurt performance and should not be used
> - If literature citations is important, then including the literature search tool can be beneficial with a minor cost in performance
> - If further understanding of model decision-making is helpful then the conversation with the critique agent can provide deeper context and critical analysis.

---

> ### Author Response · Authors · 2024-11-23
> **2. Why would a critique agent improve performance**
>
> We thank the reviewer for this important question. We have included elements of the following discussion in the Appendix to provide further reasoning on why the critic agent was used.
>
> **Prompt engineering is important**: Past research into how diverse prompting strategies result in very different outcomes using the same LLM ([White et al. 2023](https://arxiv.org/abs/2302.11382)). Indeed, the field of prompt engineering has emerged around the importance of carefully defined prompts for achieving desired goals. For instance in Figure 3, we observe that even prompting without observation is responsible for a significant proportion of improvement from the random baseline.
>
> **Self-critique impacts performance**: Research into AI-based self critique and reinforcement has shown a significant impact in LLM responses. For example, [Bai et al. 2022](https://arxiv.org/abs/2212.08073) introduce RL from AI feedback (RLAIF) where they used AI-based feedback on the responses from the same AI model to successfully finetune a different model. Another example is [Shinn et al. 2023](https://arxiv.org/abs/2303.11366) where agents verbally reflect on task feedback, then maintain their own reflections in memory to enable better decision-making in subsequent trials.
>
> **Critic agent improves BioDiscoveryAgent performance and enhances interpretability**: In our case, the critic agent is crucial in this sense because it is prompted to challenge the selections made by the actor and thus allows the agent to consider a more diverse set of genes that it may have known about but needed a different prompting strategy to retrieve. By conditioning on the existing gene list, the new prompt is able to search beyond the most obvious choices for that specific round. For instance, in Table 9, in the case of Scharenberg and Carnevale datasets, we observe a meaningful improvement in performance upon including the agent.
>
> Another important benefit of the agent is it adds an additional layer of interpretability. In lines 1789-1797, the critique agent highlights the lack of diversity in the original set of genes that were proposed and criticizes the model for only focusing on interferon response and immune signaling pathways.

---

> ### Author Response · Authors · 2024-11-24
> **3. Tackling cold start in traditional approaches using an agent**
>
> > You mentioned in the Discussion section that much of the improvements brought by the agent are observed in the early stages, which I think may be related to a better cold start compared to traditional methods. Do you think we can probably use the cold starts from agents to enhance traditional methods?
>
> We thank the reviewer for this very valuable suggestion. Indeed, for our future work we are very interested in further exploring the intersection between conventional machine learning methods that perform better in the domain of large data with agentic approaches that can tackle the cold start problem.
>
> **BioDiscoveryAgent (Sonnet) + Coreset Experiment**: To test the reviewer’s proposed idea, we combined the best-performing traditional approach, Coreset, with the best-performing agent, BioDiscoveryAgent using the Claude 3.5 Sonnet backbone. For the first round, we used the genes predicted by Claude Sonnet, and in all subsequent rounds, we used the genes predicted by the Coreset model.
>
> | Model              | Schmidt1 (All/N-E) | Schmidt2 (All/N-E) | CAR-T (All/N-E) | Scharenberg (All/N-E) | Carnev. (All/N-E) | Sanchez (All/N-E) |
> |---------------------|--------------------|---------------------|-----------------|-----------------------|-------------------|-------------------|
> | BioDiscoveryAgent (Sonnet)  | **0.095** / **0.107**      | 0.104 / **0.122**       | **0.130** / **0.133**   | **0.326** / **0.292**         | 0.042 / **0.044**     | **0.066** / **0.063**     |
> | Coreset            | 0.072 / 0.066      | 0.102 / 0.084       | 0.069 / 0.059   | 0.243 / 0.197         | **0.047** / 0.038     | 0.061 / 0.054     |
> | BioDiscoveryAgent (Sonnet) + Coreset   | 0.092 / 0.079      | **0.133** / 0.092       | 0.111 / 0.101   | 0.215 / 0.160         | **0.047** / 0.038     | 0.063 / 0.054     |
>
> **Results show consistent improvement over Coreset but doesn't outperform BioDiscoveryAgent alone**: The results show that incorporating the agent allows the Coreset approach to better overcome its cold start problem, achieving an overall higher score across all datasets except Scharenberg. This exception is reasonable, as BioDiscoveryAgent did not show strong first-round performance for that dataset. Interestingly, for the Schmidt2 (IL2) dataset, we observed improved performance over the best agent in the case of all genes (but not for non-essential genes). However, in all other datasets, BioDiscoveryAgent with the Claude 3.5 Sonnet backbone still outperformed this hybrid approach.
>
> **Implications**: This results highlights the exciting potential of combining agents with traditional machine learning methods in experiment design. We believe that incorporating agents into such pipelines will become essential—not only for addressing cold start problems with prior knowledge but also for improving decision-making, comparing new evidence with previous results, and enhancing interpretability at each stage.

---

> ### Author Response · Authors · 2024-11-25
> **4. Contributions of our agent to closed loop biological experiment design and consistency in results**
>
> > their method does not seem refined enough to contribute substantially to the agent methodology in this application scenario.
>
> We thank the reviewer for their feedback. The primary aim of our paper is to demonstrate the utility of LLM-powered agents in the closed loop design of biological experiments as opposed to the currently dominant Bayesian optimization paradigm.
>
> - **First to apply agents to closed loop biological experiment design**: Our application of agents to the genetic perturbation scenario is entirely novel and, to our knowledge, has not been proposed before. While we agree the agent framework can be further refined, this was not the primary focus of our work.
> - **Outperforms all published baselines**: We benchmarked against all formally published models for this task, including the Bayesian optimization baselines from GeneDisco [Mehrjou et al, ICLR 2022](https://iclr.cc/virtual/2022/poster/6889) and DiscoBax [Lyle et al. ICML 2023](https://arxiv.org/abs/2312.04064). Our approach demonstrates a 21% improvement in detecting hit genes and a 46% improvement when only considering non-essential genes, a much harder task.
> - **Interpretability at every stage**: BioDiscoveryAgent's predictions are highly interpretable at every stage, which is not a feature in any of the baseline models.
> - **Introduce 2-gene setting**: Our work introduces closed-loop experiment design for 2-gene perturbation settings, which present a significantly larger search space and greater scientific value. Existing machine learning baselines cannot directly address this due to the need for a novel representation of gene combinations.
> - **Suite of tools to expand agent's capabilities**: The agent we develop is extensible with various different tools that add novel capabilities such as accessing tabular datasets and providing literature citations. We highlight how tool use significantly improve model performance for smaller LLMs. In Table 4, our paper presents a novel relationship between the size of LLMs and the benefit of tools use.
> - **Validation on unpublished data**: To enhance the rigor of our validation, we include an unpublished dataset (CAR-T) that is generated in a wet-lab by one of the authors of this paper.
>
> > There also lacks an in-depth discussion on the inconsistency in their empirical findings.
>
> We thank the reviewer for their comment. It's unclear what they mean by inconsistency in the empirical findings.
>
> **Consistency in agent performance**: While it’s true that there is variation in performance across different model backbones, we don’t think the performance varies randomly. Claude 3.5 Sonnet is nearly always the best performing LLM and o1-mini and o1-preview perform equally well and are both second best. To provide further clarity, we added a new table (Table 8) ranking each agent across datasets. Claude 3.5 Sonnet achieved the lowest average rank of 1.75, followed by o1-mini and o1-preview, both with an average rank of 3.0. The similar performance of the o1 models suggests that the variation is not random.
>
> **The best agent consistently outperforms the best traditional machine learning model**: Furthermore, if we just restrict our result to BioDiscoveryAgent (Claude 3.5 Sonnet) and the best performing traditional method (Coreset), the agent outperforms in every single comparison, with only one exception.
>
> | Model              | Schmidt1 (All/N-E) | Schmidt2 (All/N-E) | CAR-T (All/N-E) | Scharenberg (All/N-E) | Carnev. (All/N-E) | Sanchez (All/N-E) |
> |---------------------|--------------------|---------------------|-----------------|-----------------------|-------------------|-------------------|
> | BioDiscoveryAgent (Claude 3.5 Sonnet)  | **0.095** / **0.107**      | **0.104** / **0.122**       | **0.130** / **0.133**   | **0.326** / **0.292**         | 0.042 / **0.044**     | **0.066** / **0.063**     |
> | Coreset            | 0.072 / 0.066      | 0.102 / 0.084       | 0.069 / 0.059   | 0.243 / 0.197         | **0.047** / 0.038     | 0.061 / 0.054     |
>
> **Consistency in tool use**: The purpose of introducing tools in the paper was not to show that they always improve agent performance but rather to explore more deeply how tool use varies for different LLM backbones and how they can be enhanced further.
> - For smaller LLMs such as Claude v1, GPT3.5 Turbo, we observe an improvement in performance in every case when using all tools, with only 2 exceptions where the performance remains approximately the same.
> - As an example of the consistency of this effect: when using all tools, a smaller LLM, Claude v1 outperforms all machine learning baselines, even though it is unable to do so across all datasets without tools.
> - For larger LLMs, as stated in the text, our results indicate that our current tools do not improve performance and in some cases can hurt performance.
>
> We appreciate the reviewers feedback and have incorporated additional labeling in Table 3 and Appendix 7 to further clarify this distinction.

---

> ### Author Response · Authors · 2024-11-25
> **Summary Response for Reviewer WoKr**
>
> We thank the reviewer WoKr for their valuable feedback and evaluation of our work. We are glad that the reviewer found our work clear and easy to follow. They appreciated the range of diverse experiments as well as the large margin of improvement we show over published baselines. We addressed the following comments that WoKr raised about our work:
>
> **Novelty and core contribution**: We described how our core contribution was the application of LLMs to closed loop biological experiment design. Our approach does not involve any model training or feature engineering, works in both 1-gene and 2-gene settings and is interpretable at each stage. Its simplicity ensures broad applicability, achieving significant performance improvements: a 21% increase in detecting hit genes and a 46% improvement in the harder task of identifying non-essential hit genes.
>
> **Variation in the benefit of tool use with different LLMs**: Our paper presents a novel analysis on how tools can benefit different LLMs differently depending on their size. We include new analysis on why tools benefit different LLMs to varying degrees. Factors such as retrieval capabilities and model confidence appear to play a role, potentially explaining why smaller LLMs consistently benefit from tools why larger LLMs do not. Drawing on prior research, we also described on how critique agents enhance performance.
>
> **Agent + traditional ML approach**: We implemented reviewer WoKr’s excellent suggestion to combine the agent with the best-performing baseline model. This hybrid approach consistently outperformed the baseline yet was still less effective than the agent alone.
>
> We believe our responses comprehensively address the reviewer’s concerns and hope they will consider raising their score.

---

> > ### Comment · Reviewer_WoKr · 2024-11-27
> > **Response to author rebuttal**
> >
> > I thank the authors for their efforts in responding to my questions. The added experiments and discussions are interesting and useful, which I highly recommend adding to the modified paper. Below are my remaining concerns:
> > - Given the substantial improvement in LLM abilities in the recent year, the fact that this simple-yet-effective agent method outperforms previous more complex ones is not very informative, since the biggest factor (LLM) was not controlled;
> > - There is reasonable doubt that the inability of the current strongest LLMs in constantly benefiting from tool usage may reflect suboptimal method design;
> > - Your explanation of why adding a critique is helpful is still not convining enough to me, especially when the critique only reviews the agent's response itself and not provided with external feedback. However, this is an open problem that has been puzzling me for a long time, and I don't expect a better answer.
> >
> > That being said, I feel this is a complete and decent work that proves the feasiblity of agent-based methods in gene perturbation experiment design, and probably the first work in this application scenario. Additioanlly, I appriciate the authors' honesty and openness in presenting some of the sub-optimal results for future discussion and improvement, which does contribute to the community. So I decide to raise the score.

---

> ### Author Response · Authors · 2024-11-27
> **Response to comment on controlling for LLMs and critique agents**
>
> We thank the reviewer for their thoughtful comments and very constructive feedback that improved the quality of the paper. We are grateful that they revised their score
>
> - **Controlling for the impact of the LLM**: We acknowledge that the LLM is the primary contributor to the agent's performance. You will note that, in the paper, the term BioDiscoveryAgent is almost always accompanied by the specific LLM backbone being used in that context. Additionally, we performed extensive evaluations and ablations using a range of LLMs varying in size and originating from different institutions. We hope these experiments provided clarity on the specific ways the LLM influenced overall performance relative to other components of the agent infrastructure such as tools. If the reviewer has recommendations for additional experiments or baselines to better control for the LLM's impact, we would be happy to incorporate them.
>
> - **Tool use**: We agree that there is room for a lot of exploration on how tools are developed for new and larger LLMs. Our work highlights some of the challenges in this area, particularly how current tools may not fully account for the enhanced retrieval capabilities of larger models or their higher confidence in their predictions. As the reviewer noted, we chose to include this analysis because it productively contributes to the conversation around agent use and how best to augment it. In fact, if we have only included results from a smaller model like Claude-v1, we would have still consistently outperformed the baselines while also consistently benefitting from the tools. Thus, it was our goal to be exhaustive and highlight the benefits and disadvantages of each LLM backbone and tool specification.
>
> - **Critique agent**: We agree that the critique agent is a non-intuitive concept but has been made use of in many other recent papers. A recent review on the use of agents in biomedical research ([Gao et al. Cell 2024, Empowering biomedical discovery with AI agents](https://www.sciencedirect.com/science/article/pii/S0092867424010705)) highlighted the importance of multi-agent systems for scientific research. Interestingly, they specifically noted how prompting a single LLM to take on different roles can be highly effective:
> > "a single LLM, programmed to fulfill multiple roles, can provide a more practical and effective solution than developing specialized models. By assigning specific roles, the agents can replicate the specialized knowledge of experts across various fields ... Early results in clinical medicine question-answering suggest that assigning specific roles, such as clinicians, to GPT-4 can achieve better performance in terms of accuracy on multiple-choice benchmarks compared with using domain-specialized LLMs like BioGPT..'
>
> We once again thank the reviewer for their thoughtful engagement in the rebuttal process. If our responses have adequately addressed their concerns, we hope they will consider further raising their score. Alternatively, we would appreciate any additional questions or suggestions for further analysis that can be added to the paper before the deadline.

---

### Official Review · Reviewer_qGCq · 2024-11-04

**Soundness:** 3
**Presentation:** 2
**Contribution:** 2
**Rating:** 8
**Confidence:** 4

**Summary:**

The authors present a LLM agent that can design single-cell experiments. At every step the agent uses literature and Reactome to narrow down a list of genes until it converges, taking the output of the previous step as context.

**Strengths:**

The authors present an interesting method of using LLMs to help guide perturbation experiments. The authors present extensive experiments with LLMs to pick the LLM best suited to coordinate the tasks to decide which genes will be perturbed. The LLM has access to literature and Reactome to make the decisions of the next gene to perturb. This work follows an increasing movement to design LLM agents to automate parts of the science process and, upon addressing of my points, represents a fruitful direction the community should head towards.

**Weaknesses:**

**Major points:**

1. I am confused what does this paper add when there exists models like GEARS which can predict gene expression that results from perturbations? I realize the authors would argue that this model helps designs experiments as seen by its higher hit ratio but with this logic doesn't GEARS have a very high hit ratio, in one round of in-silico experiments getting many hits?

2. I am also confused by the metric, what about the genes that the model suggests that are not hits? Do we not care about precision? The genes that the model suggests that are not hits should also be considered. The more of these the worse off.

3. In line 1855 in the sample agent conversation where does the agent get log-fold change? Does it have a tool that gives it access to this information? From the main text it was just literature and reactome?

4. From Appendix Table 7 it is not clear to me that adding tools helps, a lot of the confidence intervals overlap? Is the difference in the numbers significant here? Also these differences vary by LLM which makes it hard to discern which to do for a particular task.

5. What is the agent adding that cannot be done by simply doing a more traditional pathway enrichment analysis and google?

**Minor points:**

1. Two-gene perturbation prediction is not a new problem. GEARS introduces this problem already.

**Questions:**

No questions.

---

> ### Author Response · Authors · 2024-11-21
> **1. GEARS is not a baseline that can be applied in this setting**
>
> >I am confused what does this paper add when there exists models like GEARS which can predict gene expression that results from perturbations?
>
> We thank the reviewer for their feedback. While GEARS ([Roohani et al. Nature Biotech, 2023](https://www.nature.com/articles/s41587-023-01905-6)) is a machine learning model for predicting gene expression following perturbation, it is designed for a completely different setting and is not applicable to the tasks addressed in this paper.
>
> GEARS requires specialized single-cell Perturb-Seq data for training. Perturb-Seq captures gene expression changes in all genes within a cell after perturbation, yielding a high-dimensional readout (typically in $\mathbb{R}^{18000}$) for each individual cell. In contrast, general CRISPR screen experiments, which are the focus of this paper, produce scalar readouts ($\mathbb{R}$), such as cell vitality or the expression level of a single gene. Additionally, these readouts are typically aggregated across cells, rather than recorded at the single-cell level.
>
> Data from conventional CRISPR screens is far more abundant, less expensive to generate, and is central to this study. Thus, GEARS cannot be applied to the datasets or tasks considered here.
>
> >I realize the authors would argue that this model helps designs experiments as seen by its higher hit ratio but with this logic doesn't GEARS have a very high hit ratio, in one round of in-silico experiments getting many hits?
>
> The concept of ‘hits’ and ‘hit ratios’ does not apply to GEARS. The goal of single-cell perturbation models like GEARS is to match a target gene expression profile, not to identify outliers or hits. Furthermore, what constitutes a ‘hit’ in the high-dimensional space of Perturb-Seq experiments is not clearly defined in the literature. Notably, the terms ‘hit’ or ‘hit ratio’ are not mentioned even once in the [GEARS paper](https://www.nature.com/articles/s41587-023-01905-6).
>
> >I am confused what does this paper add
>
> **Contributions of this paper**:
> - BioDiscoveryAgent introduces the first LLM agent-based approach for designing genetic perturbation experiments. Specifically, we address CRISPR perturbation screens with scalar phenotypic readouts, as described above.
> - We focus on iterative experimental design, aiming to optimize the detection of phenotypic hits across multiple experimental rounds. To our knowledge, we have included all other formally published models for this task: the Bayesian optimization baselines from GeneDisco ([Mehrjou et al. ICLR, 2022](https://iclr.cc/virtual/2022/poster/6889)) and DiscoBax ([Lyle at el. ICML, 2023](https://arxiv.org/abs/2312.04064)). We demonstrate significant performance improvement over these baselines without requiring any model training.
> - The agent we develop is extensible with various different tools that add novel capabilities such as accessing tabular datasets and providing literature citations. Moreover, the agent's predictions are highly interpretable at every stage, which is not a feature in any of the baseline models that we compare against.
> - It is also important to clarify that sequential experimental design is not the same as a one-step predictive models such as GEARS. Closed loop experimentation requires the design of an explicit acquisition function to define the sampling strategy. This function guides data generation and also helps balance exploration with exploitation. In the case of BioDiscoveryAgent, the agent implicitly defines its own acquisition function. We use the `Research Plan` entry in its response to encourage it to follow a strategy across experimental rounds.

---

> > ### Comment · Reviewer_qGCq · 2024-11-23
> > **Response to Point 1**
> >
> > I thank the authors for the clarification. The distinction between the two application areas is clear. Though I do believe it is possible to use GEARS for this application, I agree doing so would require modifying the architecture of GEARS and retraining for this application which is beyond the scope of this work.

---

> ### Author Response · Authors · 2024-11-21
> **2. Measuring precision or recall is equivalent in this setting**
>
> >I am also confused by the metric, what about the genes that the model suggests that are not hits? Do we not care about precision? The genes that the model suggests that are not hits should also be considered. The more of these the worse off.
>
> We thank the reviewer for the question. Detecting hits is a proxy for how well the model is learning to identify true biological phenotype over successive rounds of experimentation. In our experiments, we have a fixed number of genes predicted by the agent in each round. Since we always measure the final score cumulatively at a pre-specified round, the relative results are entirely the same whether we measure precision or recall, as the total number of predicted genes is constant.
>
> In line 152, we define the hit ratio as $\frac{|\mathcal{G}_a|}{|\mathcal{G}_p|}$ where $\mathcal{G}_a$ is the cumulative set of hits identified by the agent up to round $t$ and $\mathcal{G}_p$ is the set of all true hits for that phenotype and is fixed per dataset. This is equivalent to **recall**, as stated in the text.
>
> For **precision**, since we have the same number of predictions for each model, it is equivalent to simply scaling the recall by a constant, $ \frac{|\mathcal{G}_a|}{T \cdot B}$ where $B$ is the fixed number of genes predicted in each round and $T$ is the total number of rounds. In other words, this is just the number of hits detected divided by the batch size multiplied by the number of rounds.
>
> Since both $\mathcal{G}_p$ and $T \cdot B$ are identical across datasets, the relative ranking of methods will remain unchanged regardless of whether precision or recall is used.

---

> > ### Comment · Reviewer_qGCq · 2024-11-23
> > **Response to point 2**
> >
> > I thank the authors for the clarification, that makes sense. I have no further questions about the performance metrics.

---

> ### Author Response · Authors · 2024-11-21
> **3. Log fold change is part of the standard experimental output in Figure 1**
>
> >In line 1855 in the sample agent conversation where does the agent get log-fold change? Does it have a tool that gives it access to this information? From the main text it was just literature and reactome?
>
> We thank the reviewer for the question. The log-fold change in cytokine expression represents the desired phenotype for the specific perturbation experiment. This phenotypic outcome, obtained from the experiment conducted in the previous round, is explicitly provided in the prompt for the subsequent round. This process is explained in the main text as follows:
>
> - **Lines 173-174**: The paper explains that the experimental results from prior rounds are incorporated into the subsequent round.
> - **Lines 236-238**: We detail the experimental format, specifying that the phenotype induced by perturbations is retrieved at each round.
> - **Lines 250-252**: For the IFNG and IL2 datasets, we clarify that the phenotype refers to the change in production of key cytokines.
> - **Figure 1(b) and Appendix A**: The exact prompt text is provided, explicitly stating:
> >After you select the genes to perturb, I will experimentally measure the log-fold change in Interferon-gamma (IFNG).
>
> Thus, the log-fold change is derived directly from the experimental results provided in the prompt, not from a separate tool.

---

> > ### Comment · Reviewer_qGCq · 2024-11-23
> > **Response to Point 3**
> >
> > I thank the authors for the clarification; it is clear to me how logfold is integrated.

---

> ### Author Response · Authors · 2024-11-21
> **4. Tool use does not help larger LLMs but more consistent benefit seen in smaller LLMs**
>
> >From Appendix Table 7 (Now Table 9 in revised submission) it is not clear to me that adding tools helps, a lot of the confidence intervals overlap? Is the difference in the numbers significant here?
>
> **The purpose of Appendix Table 7 (Now Table 9 in revised submission) and Table 3, 4 was not to show that tool use always improves agent performance** but rather to explore more deeply how tool use varies for different LLM backbones.
> - **In Table 4**, we present an interesting relationship between model size and benefit of tool use. Specifically, the benefit of tool use is more pronounced in smaller models with lower cost per token compared to larger ones.
>
> - **In Appendix Table 7 (Now Table 9) and Table 3**, we delve deeper into this contrast by examining two specific examples. One involves a smaller model, Claude 3 Haiku, and the other tests a larger model, Claude 3.5 Sonnet, which has a higher price per token. For Claude-3.5-Sonnet, as stated in lines 370-372, we do not believe the current tools are showing value and indeed the performance improvements are within the error bars. On the other hand, for the smaller model, Claude 3 Haiku, we observe a significant improvement in performance that exceeds the confidence intervals.
>
> - We appreciate the reviewers feedback and have incorporated additional labeling in Table 3 and Appendix 7 to indicate that Claude 3 Haiku is showing consistent improvement in performance through tool use while Claude 3.5 Sonnet is not. This can help the reader more easily discern the key message from the Table.
>
> > Also these differences vary by LLM which makes it hard to discern which to do for a particular task.
>
> The All-Tools setting consistently demonstrates higher performance in the case of the smaller models. For larger models, we find that no-tools is usually the most reliable setting and recommend further research in how to optimally design new tools to augment their performance.

---

> > ### Comment · Reviewer_qGCq · 2024-11-23
> > **Response to Point 4**
> >
> > I thank the authors for the clarification. The revisions to the manuscript, especially being clear that tools only help smaller LLMs is needed and addresses my concern.

---

> ### Author Response · Authors · 2024-11-21
> **5. Traditional pathway enrichment and gene search does not outperform random baseline**
>
> > 5. What is the agent adding that cannot be done by simply doing a more traditional pathway enrichment analysis and google?
>
> - We thank the reviewer for their comment. We have added an additional human baseline in Table 1, which consists of Google search for pathways involved in each test (Table 6) followed by enrichment analysis as suggested by the reviewer. The result showed little to no improvement from the random baseline for five out of the six datasets.
>
> - This makes sense given the known shortcomings of traditional pathway enrichment analysis, such as assumption of gene independence ([Tamayo et al., Stat Methods Med Res. 2016](https://journals.sagepub.com/doi/10.1177/0962280212460441?url_ver=Z39.88-2003&rfr_id=ori:rid:crossref.org&rfr_dat=cr_pub%20%200pubmed)) or overrepresentation of highly connected genes ([Markowetz et al., PLoS Comp. Bio. 2010](https://journals.plos.org/ploscompbiol/article?id=10.1371/journal.pcbi.1000655)), which makes the enrichment analysis quite a limited indicator for hits by itself.
>
> - Moreover, this highlights the complexity of the genetic perturbation experiment design task. This task demands a meaningful and informative representation of gene relationships as well as the ability to weigh experimental evidence against prior biological knowledge to identify the most promising hits in a given scenario—capabilities that our agent uniquely integrates.

---

> > ### Comment · Reviewer_qGCq · 2024-11-23
> > **Response to Point 5**
> >
> > I thank the authors for this experiment. I am surprised that human performance was very similar to random. Can the authors explain what exactly the procedure was for the human baseline? For example, if I google genes involved with CAR-T proliferation I get 10 genes. If a human does their own literature review in addition to pathway and enrichment analysis, this should be better than random.

---

> ### Author Response · Authors · 2024-11-21
> **We introduce the task of 2-gene perturbation for closed-loop experiment design**
>
> > Two-gene perturbation prediction is not a new problem. GEARS introduces this problem already.
>
> We thank the reviewer for their feedback.
> - As stated in the abstract, our paper introduces combinatorial perturbation prediction in the context of **closed loop experimental design**. There has not been any work, to our knowledge, that has performed combinatorial perturbation experiments sequentially in a closed feedback loop. As stated above, it is important to distinguish between the sequential experimental design that requires an acquisition function and balancing exploration with exploitation versus one-step predictive models that do not do so.
> - GEARS does not propose any solution for designing a sequence of combinatorial perturbation experiments to optimize a desired phenotype. While it's true that GEARS could be used in a sequential experiment design framework but it would require the design of a suitable acquisition function on top of GEARS to design the next experiment.
> - Moreover, GEARS is not designed to be trained on data from conventional CRISPR screens that do not have a transcriptome-wide readout such as Perturb-Seq experiments. Indeed, within the specific context of Perturb-Seq, it is possible to build an LLM agent that leverages GEARS as a tool and uses its prediction to design the next 2-gene perturbation experiment. However, Perturb-Seq data is very limited. The largest 2-gene perturbation dataset for Perturb-seq is from [Norman et al. Science 2019](https://www.science.org/doi/10.1126/science.aax4438) and consists of 131 combinations. The combinatorial dataset we used in this paper ([Horlbeck et al. Cell 2018](https://www.cell.com/cell/pdf/S0092-8674(18)30735-9.pdf)) tests over 100,000 unique combinations (Line 259).

---

> > ### Comment · Reviewer_qGCq · 2024-11-23
> > **Response to point 6**
> >
> > I thank the authors for the clarification. I agree with them that this is a new problem in the context of closed loop experimental design. I again contend that GEARS can be modified to do this task, but I realize that is outside the scope of this work.

---

> ### Comment · Reviewer_qGCq · 2024-11-23
> **Additional points of Review**
>
> I thank the authors for the effort they put in to address my concerns about the manuscript. I will raise my score to a 5, but there are still a few concerns that keep me from accepting this paper. They are the following:
>
> 1. My comment about the human baseline, see my response.
>
> 2. Literature search with an agent is not an easy task to do. There has been a flurry of recent works trying to tackle this problem with varied success. How often does the agent return literature that is not relevant and how does this effect performance? I imagine the agent will not do well in applications where there is not a lot of literature information already which is arguably the applications where these screens would most likely be most useful.
>
> This blends into another point that I had. The authors claim that there is no information leakage because of dates studies are published or the fact that the CAR-T study is not published yet. However, information leakage still exists. If I search "genes involved with CAR-T" proliferation I get many studies that talk about genes that are potentially relevant. This could explain why large LLMs do not need tools as they have seen this literature already and don't need to be represented what they were already trained on.
>
> The way to address this is to either (1) perform an experiment in an application area where there is little to no literature or (2) show that the agent can identify genes relevant to the application area that no other literature has ever mentioned as being relevant. This would help show that the agent can help in the scenarios this agent is most likely to be used: in application areas where little to no knowledge is known about the genes that are relevant.
>
> 3. Figure 9 is great but presents some confusion. In the original results I see some of the baselines rise to almost match the performance of the agent by iteration 5. Can you show how these baselines do as you extend the number of iterations to 30? This is just to make sure they don't become better than the agent at iteration 10, so as to solidify the improvement in performance is not due to selecting a certain number of iterations where the agent did better.
>
> I see the argument the authors have presented about how increasing the number of iterations is not relevant as this is not what is done experimentally. If we do 5 iterations, scanning 100 genes each we get 500 genes tested, but some of these papers talk about doing 100,000 or so gene perturbations. So extending to 30 iterations, scanning 100 genes each does not reach the amount of genes scanned by typical experiments? Correct me if I am wrong.
>
> The point of the above statement is to argue that the authors should run an experiment where they go to completion, i.e. keep running the methods until the hit-rate becomes 1. Then we can truly know if one method is better than the other, since we can see which method gets to a 100% hit rate the fastest.
>
> I am willing to raise my score to an accept if these points are addressed.

---

> ### Author Response · Authors · 2024-11-24
> **1. Why manually designing these experiments is difficult**
>
> We thank the reviewer for responding to our rebuttal and continuing to engage constructively with our work. We are very grateful for their feedback that is helping to strengthen our work.
>
> > I am surprised that human performance was very similar to random. For example, if I google genes involved with CAR-T proliferation I get 10 genes. If a human does their own literature review in addition to pathway and enrichment analysis, this should be better than random.
>
> It is understandable that this task may seem like it is simply identifying biologically relevant genes from a list but it is in fact significantly more complex.
>
> **Complexity of the task**: If the task were as simple as a gene search, we would observe a much higher number of hit genes identified in the first round. The agent could simply select all genes annotated for related biological functions. However, this is not the case. Even for the best models, only 12–14 out of 128 predicted genes typically exhibit a biologically significant (hit) phenotype in the first round.
>
> **Reason for this complexity**: Biological systems are influenced by numerous intricate factors, many of which are not fully captured by pathway labels, which are very often noisy and imprecise. These systems exhibit redundancy and feedback mechanisms, meaning that perturbing a single gene in a pathway does not always disrupt its function. Moreover, the phenotypes we're interested in are the strongest ones not just those that are different from baseline. Additionally, gene regulatory relationships can vary significantly between donors, cell types, cell states. This inherent complexity underscores the importance of conducting large-scale perturbation experiments under precise conditions. While prior biological knowledge from online sources and databases can provide a useful starting point, experimental probing offers deeper insights into which gene sets are truly impacting downstream phenotypes in the specific system being studied.
>
> **Why googling is not enough**: This is why simply googling for genes that are involved in CAR-T cell proliferation is not the same as actually performing a perturbational experiment to validate the hypothesis. Moreover, the phenotype that we test in our experiment is much more precisely defined than just genes involved in CAR-T proliferation. See Appendix B for task definition:
>
> > Identify genes that upon inhibition allow cells to resist T-cell exhaustion, under the HA GD2 CAR (chimeric-antigenic receptor) condition. The measurement we were making was the log fold change in normalized sgRNA read counts compared to the non-targeting control, 22 days after perturbation
>
> **Example illustrating complexity of the task**: As an example, we present to you a simple experiment. For the CAR-T dataset, we randomly selected 25 genes from the list of true hits consisting of over 750 hit genes. We pasted this list in GPT-4o and asked the LLM the following questions:
>
> - Q. **Can you identify any common pathways here?**
> - A. It searched the internet and identified several pathways that were relevant to subsets of genes in this list. These include Immune Response and Inflammation, Metabolic Processes, Protein Synthesis and Translation, Cell Cycle and Apoptosis, Lipid Metabolism and Signaling, Glycosylation and Protein Modification, DNA Repair and Replication and a few more.
>
> As you can see, there isn’t any existing annotation that clearly implicates the majority of these genes as related to CAR-T function. Then we asked a more pointed question:
>
> - Q. **Is this gene list associated with CAR-T cells?**
> - A. Out of the 25, it identified 4 that could be linked with CAR-T cell function directly.
>
> The point we’re making is that simply searching for genes related to CAR-T proliferation is not the same as running an experiment to identify which genes are most effective at doing so in a specific setting. If the results were so obvious then by looking at the correct solution the LLM should immediately have identified all or at least a majority of the hit genes as being associated with CAR-T cells.
>
> > Can the authors explain what exactly the procedure was for the human baseline?
> **Procedure we followed for human baseline**:
> - In the case of the CAR-T dataset, we first googled Reactome pathways that are involved with CAR-T proliferation, we got three significant pathways. We built our first prediction of 128 genes by selecting genes that are involved the most in these pathways.
> - Then for the subsequent rounds, we incorporate information gained from the experiment as well. We perform enrichment analysis with the Reactome database on the set of genes that are identified as hits: this will return pathways where there are most overlapping genes with significant p-value. Once we retrieve these pathways, similar to the first round sampling, we look for other genes that are most involved in these pathways. The human baseline for the other datasets followed this exact same procedure.

---

> ### Author Response · Authors · 2024-11-24
> **2. Why memorizing past papers wouldn't enable the current agent performance**
>
> We thank the reviewer for their important comments on accessing the literature and its impact on agent performance
>
> > How often does the agent return literature that is not relevant and how does this effect performance? I imagine the agent will not do well in applications where there is not a lot of literature information already which is arguably the applications where these screens would most likely be most useful.
>
> We agree with the reviewer that in the case of systems that are less well studied in such settings (e.g. neurons in the Sanchez dataset) and where there is less information in the literature, the model does struggle more than in other cases. However, the same could be said about humans working in a new field as well. Were human scientists to make predictions in a new area, we would also make more mistakes than if we were to speak about well-studied systems. The bigger point is that doing better in well studied systems is not indicative of information leakage or memorization, it could also very likely just be an area where the agent's understanding is more sound and it can answer questions more effectively. However, we agree with the reviewer that it warrants further investigation. This is what led us to use our own unpublished datasets in very specific settings of CAR-T cell activation that had not previously been tested before in the literature. Compared to the past models that we benchmark against in this space, none of them generated their own data.
>
> > However, information leakage still exists. If I search "genes involved with CAR-T" proliferation I get many studies that talk about genes that are potentially relevant.
>
> As described in the response to the previous question, simply searching for "genes involved with CAR-T proliferation" is very unlikely to identify genes that produce hit phenotypes (or the strongest phenotypic effects). This is also because the exact phenotype we tested in our experiment was much more specific: genes that upon inhibition allow cells to resist T-cell exhaustion, under the HA GD2 CAR (chimeric-antigenic receptor) condition.
>
> Moreover, each of the studies we used for experimental datasets is an independent peer reviewed journal publication because the experimental data it contains (i.e. the hits they identify) presents new information not equivalent to what's already been seen in past papers or experiments. Please see the previous response for why google searching genes doesn't adequately account for the complexity of the task.
>
> > The way to address this is to either (1) perform an experiment in an application area where there is little to no literature
>
> We thank the reviewer for this suggestion but given the time left, we do not think it would be feasible to perform a new experiment which takes several months. Moreover, we already include newly generated data in our paper for a setting of CAR-T cell proliferation that has not been previously tested in the literature.
>
> > or (2) show that the agent can identify genes relevant to the application area that no other literature has ever mentioned as being relevant. This would help show that the agent can help in the scenarios this agent is most likely to be used: in application areas where little to no knowledge is known about the genes that are relevant.
>
> We thank the reviewer for this suggestion. For the BioDiscoveryAgent model, we took the set of all hits predicted at round 5 and asked GPT-4o to identify if any of these had been linked with CAR-T cells before.
>
> Out of 33 total hits, it found a direct link for 4 of them, a possible indirect link for 7 of them and for all the remaining 22 hit genes, it said "As of now, there is limited or no direct evidence linking these genes to CAR-T cell research or therapy." We also tried a few Google searches and were unable to make meaningful connections for these genes to previously published papers on CAR-T cells. In case the reviewer wants to run their own experiments, then some of these genes are FOXG1, SDHB, SSRP1, TBX5, TCF12. Finally, it is worth emphasizing that even if papers were found mentioning a gene’s potential relevance to CAR-T cells, such signals are often highly noisy and unlikely to explain the strong experimental performance we observe.
>
> We hope that this is sufficient evidence that BioDiscoveryAgent is not simply relying on memorization to predict new genes. Instead, it effectively integrates prior knowledge from the literature and other source in combination with experimental outcomes to make reasoned predictions.

---

> > ### Comment · Reviewer_qGCq · 2024-11-25
> > **Response to author rebuttal**
> >
> > I thank the authors for the clarification and extra experiments. The CAR-T dataset is key for demonstrating the approach's utility. I would recommend the authors implement my other suggestions of focusing on unique, undiscovered genes in the other datasets to further demonstrate the utility in any further submission of this work. I raise my score to 6. I think it warrants acceptance, but my hesitation lies in my last point about the number of rounds versus baseline/agent performance.

---

> ### Author Response · Authors · 2024-11-27
> **3. BioDiscoveryAgent outperforms baselines in longer runs of 30 iterations**
>
> >Can you show how these baselines do as you extend the number of iterations to 30?
>
> We thank the reviewer for their useful feedback. We apologize for the delay as we had to train over 900 models (6 datasets x 7 baseline models x 10 iterations x 2 batch sizes) across 30 rounds of experimentation. We present the results from our analysis in Tables 12, 13, 14 and 15 in the revised manuscript.
>
> | **Model**         | **Schmidt1 (10 rounds)** | **Schmidt1 (20)** | **Schmidt1 (30)** | **Schmidt2 (10)** | **Schmidt2 (20)** | **Schmidt2 (30)** | **CAR-T (10)** | **CAR-T (20)** | **CAR-T (30)** |
> |--------------------|-------------------|-------------------|-------------------|-------------------|-------------------|-------------------|----------------|----------------|----------------|
> | Random            | 0.06             | 0.123            | 0.181            | 0.047            | 0.092            | 0.139            | 0.06           | 0.119          | 0.179          |
> | **Baseline Models**                                                                                                                                    |
> | Soft Uncertain    | 0.061            | 0.123            | 0.192            | 0.049            | 0.100            | 0.148            | 0.054          | 0.127          | 0.186          |
> | Top Uncertain     | 0.077            | 0.146            | 0.216            | 0.068            | 0.137            | 0.200            | 0.074          | 0.139          | 0.204          |
> | Margin            | 0.070            | 0.143            | 0.211            | 0.068            | 0.130            | 0.192            | 0.075          | 0.136          | 0.209          |
> | Coreset           | 0.123            | **0.231**            | **0.303**            | 0.136            | 0.248            | 0.314            | 0.092          | 0.144          | 0.227          |
> | Badge             | 0.079            | 0.153            | 0.226            | 0.082            | 0.154            | 0.217            | 0.064          | 0.144          | 0.208          |
> | K-Means (E)       | 0.066            | 0.130            | 0.198            | 0.072            | 0.131            | 0.194            | 0.052          | 0.097          | 0.161          |
> | K-Means (D)       | 0.069            | 0.140            | 0.203            | 0.086            | 0.172            | 0.235            | 0.060          | 0.125          | 0.189          |
> | **BioDiscoveryAgent (Sonnet)**        | **0.143**        | 0.214            | 0.293            | **0.179**        | **0.286**        | **0.360**        | **0.193**      | **0.264**      | **0.329**      |
>
> **Experiment**: We present the results for hit ratio at round 10, 20 and 30. In each round the agent predicts 128 genes. Results include the ML baseline methods including the coreset which performs best. Due to space constraints we share below the result for 3 datasets. We evaluate the results using the hit ratio over non-essential genes. In Table 14 and 15, we include results for all datasets with error bars.
>
> **Results**:
> - Across all but one dataset, BioDiscoveryAgent outperforms all baselines.
> - As noted correctly by the reviewer, in Schmidt1 (IFNG), we did observe previously that the best ML baseline (coreset) begins to rise around round 5 and indeed it outperforms BioDiscoveryAgent marginally in rounds 20, 30. However, in all other cases, BioDiscoveryAgent more accurately identifies genes relevant to the specific phenotype.
> - In addition, in Table 14 we show that by including the new approach of Agent + coreset, we outperform coreset in every case including for the IFNG dataset. Thus, the agent's predictions are not redundant with those made by the baseline method.
> - To be even more rigorous, we tested longer runs both with a batch size of 128 (Tables 12, 13) and 32 (Tables 14, 15). Smaller batch sizes show larger improvement which reflects better low-data generalization capabilities of the agent.
>
> > scanning 100 genes each does not reach the amount of genes scanned by typical experiments?
>
> This is a good question and describes an important challenge in the design of these experiments.
> - Experimentalists aim to maximize coverage, ideally perturbing ~20,000 genes in the 1-gene setting and >100,000 in combinatorial settings. However, cost is a major limitation. While genome-scale (20,000-gene) single-gene screens are common, they can't be applied for every experiment and are far more costly in complex contexts like mice or primary cells. Exhaustive combinatorial experiments across large gene sets are currently impossible.
> - The trade-off is to forgo a 100% hit rate and run smaller experiments in batches but with fewer iterations to quickly identify strong hit genes.
>
> We once again thank the reviewer for their thoughtful engagement in the rebuttal process. If our response has adequately addressed their concerns, we hope they will consider further raising their score.

---

> > ### Comment · Reviewer_qGCq · 2024-11-27
> > **Response to additional iterations**
> >
> > This is great! I thank the authors for running these experiments I am now convinced of the  strength of the approach. As a result, I raise my score. I thank the authors for putting so much effort in the rebuttal.

---

### Official Review · Reviewer_kidb · 2024-11-04

**Soundness:** 3
**Presentation:** 3
**Contribution:** 2
**Rating:** 5
**Confidence:** 4

**Summary:**

This work proposes BioDiscoveryAgent, a reflection-based LLM agent designed to conduct gene perturbation experiments. It iteratively predicts gene perturbation outcomes, selects genes to perturb, receives experimental results, and continues with subsequent trials based on updated predictions. The study evaluates the effectiveness of BioDiscoveryAgent using experiments on six cell perturbation datasets.

**Strengths:**

Backbone Testing: The experiments are conducted across multiple backbone models, enhancing the method's credibility.

Comprehensive Pipeline: The LLM agent is equipped with a variety of tools, resulting in a well-rounded approach.

Clear Writing: The writing is clear, making the method and results easy to understand.

**Weaknesses:**

Insufficient Baseline Comparisons: The baseline methods are somewhat distant from the proposed approach. Comparisons with interactive LLM agents, such as ReAct or multi-turn frameworks like reflection-based methods or LLMs as optimizers, would strengthen the evaluation. Additionally, as the method incorporates retrieval augmentation, a comparison with Retrieval-Augmented Generation (RAG) based on the same corpus would be valuable.

**Questions:**

Potential Information Leakage: Could the use of literature and gene search cause information leakage? Specifically, is there a risk that the task query might already be covered in the literature or database, thus affecting the validity of the results?

Linear Performance Curve: Figures 2 and 3(a) show an almost linear improvement in performance across up to 30 turns, without any indication of saturation. This is surprising, as LLM-based multi-turn trials typically reach saturation more quickly. Could you elaborate on why this linear improvement occurs and when, if at all, it might begin to saturate?

Method Validity: The experimental results in Table 1 indicate that different backbones show nearly random performance variations, and the tool usage results in Table 3 also exhibit inconsistent improvements across backbones, which raises concerns about the validity of the proposed method. One possible reason for this could be the low 1-turn accuracy, leading to a low signal-to-noise ratio. Given that the linear performance curve suggests the LLM can effectively address this task with more turns, it would be beneficial to compare the performance of different backbones and tool usage in a multi-turn setting. This could provide more stable results and more insight about backbone’s influence.

---

> ### Author Response · Authors · 2024-11-23
> **Proposed baseline methods are already part of BioDiscoveryAgent**
>
> >Insufficient Baseline Comparisons: The baseline methods are somewhat distant from the proposed approach. Comparisons with interactive LLM agents, such as ReAct or multi-turn frameworks like reflection-based methods or LLMs as optimizers, would strengthen the evaluation.
>
> We thank the reviewer for their valuable feedback. We would like to point out that our method is essentially a mix of ReAct/reflection/LLM as optimizers/RAG
> - For each step, we prompt the LLM to generate a reasoning process (including reflection) and then action as in ReAct
> - We use a separate AI Critic stage as in reflection-based methods
> - We supply all previous gene testing results (action score pairs) as in the ‘[LLM as optimizer](https://arxiv.org/pdf/2309.03409)’ paper
> - And we concatenate retrieved literature review and gene search results to the context as in RAG.
> Our ablation studies (Table 3) can be seen as evaluating the effectiveness of these different individual ingredients/approaches. We thank the reviewer for pointing out this perspective and will add this view to our paper.
>
> Additionally, we would like to point out that this paper focuses on demonstrating the promise of LLM agents as a general paradigm against the currently dominant Bayesian optimization paradigm for this task in the experimental biology research community. We agree that the LLM agent framework can be further enhanced, but that was not the primary goal for this paper. We are excited to further optimize this LLM agent implementation in future works after convincing the research community with this initial promising result.

---

> ### Author Response · Authors · 2024-11-23
> **We include unpublished data to prevent information leakage**
>
> > Potential Information Leakage: Could the use of literature and gene search cause information leakage? Specifically, is there a risk that the task query might already be covered in the literature or database, thus affecting the validity of the results?
>
> We thank the reviewer for their question.
>
> **Inclusion of unpublished data**: To avoid information leakage, we included a dataset that was unpublished and generated by the authors of this paper. This is the CAR-T dataset and is also one of the datasets where BioDiscoveryAgent performs best showing a performance improvement of 88% over the best performing machine learning baseline.
>
> **Some datasets were published after LLM cut-off dates**: Additionally, we carefully selected datasets released as recently as possible to minimize potential data leakage during training. While this approach may not entirely eliminate data leakage for all models, it is effective for models with earlier knowledge cutoff dates. For instance, the knowledge cutoff date for Claude v1 is estimated to be 2021, and for GPT-3.5, it is January 2022. These cutoff dates precede the publication dates of Carnevale et al. (2022), Scharenberg et al. (2023), and Schmidt et al. (2022). Notably, Claude v1 with all tools is among the best-performing models across these datasets.
>
> **BioDiscoveryAgent predicted hit genes not easily traced in the literature**: To test how novel the agent's predictions were, we took the set of all hits predicted by BioDiscoveryAgent (Claude 3.5 Sonnet) at round 5 for the CAR-T dataset and asked GPT-4o to identify if any of these had been linked with CAR-T cells before.
>
> Out of 33 total hits, it found a direct link for 4 of them, a possible indirect link for 7 of them and for all the remaining 22 hit genes, it said "As of now, there is limited or no direct evidence linking these genes to CAR-T cell research or therapy." We also tried a few Google searches and were unable to make meaningful connections for these genes to previously published papers on CAR-T cells. In case the reviewer wants to run their own experiments, then some of these genes are FOXG1, SDHB, SSRP1, TBX5, TCF12.
>
> We hope that this is sufficient evidence that BioDiscoveryAgent is not simply relying on memorization to predict new genes. Instead, it effectively integrates prior knowledge from the literature and other source in combination with experimental outcomes to make reasoned predictions.

---

> ### Author Response · Authors · 2024-11-23
> **Misunderstanding in performance metrics**
>
> > Linear Performance Curve: Figures 2 and 3(a) show an almost linear improvement in performance across up to 30 turns, without any indication of saturation. This is surprising, as LLM-based multi-turn trials typically reach saturation more quickly. Could you elaborate on why this linear improvement occurs and when, if at all, it might begin to saturate?
>
> We thank the reviewer for this question and apologize if our metrics were unclear.
>
> - **Performance improvement is not linear**: There may have been some misunderstanding regarding the performance score, which is the hit ratio or recall. In Figure 2 and 3(a), the performance ‘improvement’ is not linear. The performance improvement saturates after the initial few rounds.
> - **The performance score (hit ratio) is cumulative**: The hit ratio is not calculated just for the new hits for a given round but for all hits from all rounds up to that specific round. Thus, a linear curve would indicate roughly constant performance in each round (See line 152-153).
> - **Definition**: The hit ratio is defined as $\text{hit ratio} = \frac{|\mathcal{G}_a|}{|\mathcal{G}_p|} $ where  $ \mathcal{G}_a  = \cup_1^T B_t^+$. $B_t^+$ represents the hits identified in round  $t$ and $T$ is the total number of rounds. Thus, it combines all hits detected so far across all previous rounds. The denominator, $\mathcal{G}_p$, is the total number of possible hits for that dataset.
>
> Example: in Figure 2, for the first model on the top left, Schmidt22 (IFNG), the average number of new hits identified in each round were the following:
> - round 1: 13 hits
> - round 2: 9 hits
> - round 3: 10 hits
> - round 4: 8 hits
>
> Thus, even though the graph may appear to be linearly increasing, the number of new hits in each round are roughly the same. It is only the cumulative performance (hit ratio) increases with each round.
>
> **Performance curves over longer runs**: In response to the reviewer's feedback, we also include a new figure showing performance curves over extended runs of 30 rounds. These longer runs reveal a similar trend: a linear increase in the hit ratio with slight saturation in the later stages. This saturation is most evident in the case of Scharenberg22, where the dataset is smaller, and approximately 40% of all possible hit genes are predicted by round 20.
>
> **Experimental feasibility of longer runs**: We also wish to emphasize that, from a biological perspective, running a screening experiment over 30 rounds is often impractical due to technical and cost constraints. Therefore, our primary analysis focuses on 5–10 rounds with a batch size of about 100 genes, balancing experimental feasibility with economic considerations.

---

> ### Author Response · Authors · 2024-11-23
> **Agent performance across LLMs shows clear trends and is not random**
>
> >Method Validity: The experimental results in Table 1 indicate that different backbones show nearly random performance variations, and the tool usage results in Table 3 also exhibit inconsistent improvements across backbones
>
> **Consistency in agent performance**: We thank the reviewer for their comment. While it’s true that there is variation in performance across different model backbones, we don’t think the performance varies randomly. Claude 3.5 Sonnet is nearly always the best performing LLM and o1-mini and o1-preview perform equally well and are both second best.
>
> To provide further clarity, we added a new table (Table 8) ranking each agent across datasets. Claude 3.5 Sonnet achieved the lowest average rank of 1.75, followed by o1-mini and o1-preview, both with an average rank of 3.0. The similar performance of the o1 models suggests that the variation is not random. These models, while distinct, share similar architectures and likely utilize comparable training data, explaining their consistent performance.
>
> | Model              | Average Rank |
> |---------------------|--------------|
> | Claude 3 Haiku     | 7.25         |
> | GPT-3.5-Turbo      | 8.67         |
> | Claude v1          | 6.00         |
> | o1-mini            | _3.08_       |
> | Claude 3 Sonnet    | 4.58         |
> | **Claude 3.5 Sonnet** | **1.75**     |
> | GPT-4o             | 6.33         |
> | o1-preview         | _3.00_       |
> | _Claude 3 Opus_    | 3.67         |
>
> **The best agent consistently outperforms the best traditional machine learning model**: Furthermore, if we just restrict our result to BioDiscoveryAgent (Claude 3.5 Sonnet) and the best performing traditional method Coreset, the agent outperforms in every single comparison, with only one exception.
>
> | Model              | Schmidt1 (All/N-E) | Schmidt2 (All/N-E) | CAR-T (All/N-E) | Scharenberg (All/N-E) | Carnev. (All/N-E) | Sanchez (All/N-E) |
> |---------------------|--------------------|---------------------|-----------------|-----------------------|-------------------|-------------------|
> | BioDiscoveryAgent (Claude 3.5 Sonnet)  | **0.095** / **0.107**      | **0.104** / **0.122**       | **0.130** / **0.133**   | **0.326** / **0.292**         | 0.042 / **0.044**     | **0.066** / **0.063**     |
> | Coreset            | 0.072 / 0.066      | 0.102 / 0.084       | 0.069 / 0.059   | 0.243 / 0.197         | **0.047** / 0.038     | 0.061 / 0.054     |
>
> **Consistency in tool use**: Table 4 demonstrates that the benefits of tool usage in LLMs follow a clear trend correlated with model size.
>
> - For smaller LLMs such as Claude v1, GPT3.5 Turbo, we observe an improvement in performance in every case when using all tools, with only 2 exceptions where the performance remains approximately the same.
> - As an example of the consistency of this effect: when using all tools, a smaller LLM, Claude v1 outperforms all machine learning baselines, even though it is unable to do so across all datasets without tools
>
> | Model              | Schmidt1 (All/N-E) | Schmidt2 (All/N-E) | CAR-T (All/N-E) | Scharenberg (All/N-E) | Carnev. (All/N-E) | Sanchez (All/N-E) |
> |---------------------|--------------------|---------------------|-----------------|-----------------------|-------------------|-------------------|
> | BioDiscoveryAgent (Claude v1, All-Tools)  | **0.095** / **0.068**      | **0.122** / **0.114**       | **0.114** / **0.116**   | **0.333** / **0.306**         | **0.054** / **0.042**     | 0.058 / **0.055**     |
> | Coreset            | 0.072 / 0.066      | 0.102 / 0.084       | 0.069 / 0.059   | 0.243 / 0.197         | 0.047 / 0.038     | **0.061** / 0.054     |
>
> - For larger LLMs, as stated in the text, our results indicate that our current tools do not improve performance and in some cases can hurt performance.
>
> > One possible reason for this could be the low 1-turn accuracy, leading to a low signal-to-noise ratio.
>
> It is unclear what the reviewer means by 1-turn accuracy. The performance is usually highest in the first round. This may be related to the misunderstanding on the definition of the hit ratio as described in the previous response.
>
> > Given that the linear performance curve suggests the LLM can effectively address this task with more turns, it would be beneficial to compare the performance of different backbones and tool usage in a multi-turn setting.
>
> Again, we believe this comment is related to misunderstanding the cumulative nature of the hit ratio metric. In any case, we also include a new figure showing performance over 30 rounds. These longer runs reveal similar performance across all agents with slight saturation in the later stages. This saturation is most evident in the case of Scharenberg22, where the dataset is smaller, and approximately 40% of all possible hit genes are predicted by round 20.

---

> ### Author Response · Authors · 2024-11-25
> **Summary Response for Reviewer kidb**
>
> We thank the reviewer kidb for their valuable feedback and evaluation of our work. We are glad that the reviewer found our work clear and easy to follow.We addressed the following comments that kidb raised about our work:
>
> **Insufficient baseline comparisons**: We clarified to the reviewer that BioDiscoveryAgent already incorporates many of the proposed agent based frameworks including ReAct, reflection, LLM as optimizers and RAG. While we agree that our framework can be further enhanced, our primary goal in this paper was to demonstrate the utility of LLMs as agents for closed loop biological experiment design.
>
> **Tackling information leakage**: To mitigate information leakage, we carefully selected recently published datasets and included an unpublished dataset to ensure that the agent has not encountered related data before. Additionally, we conducted an experiment using predicted hit genes from BioDiscoveryAgent and queried GPT-4o to determine if those genes had been previously implicated in a similar phenotype. The LLM was unable to identify any such references for most of the genes, highlighting that BioDiscoveryAgent is not merely memorizing past results.
>
> **Performance metrics and method validity**: We clarified in the text that our performance metric is cumulative and not calculated independently for each round. We also highlight the consistency in ranking of LLMs both across datasets as well as in outperforming the best machine learning baseline. We also illustrate consistency in the benefit of tool use for smaller LLMs.
>
> We encourage the reviewer to let us know if anything is still unclear. We believe our responses comprehensively address the reviewer’s concerns and hope they will consider raising their score.

---

> > ### Comment · Reviewer_kidb · 2024-11-26
> >
> > Thank you for the detailed feedback.
> > While some of my questions have been addressed, the major concern regarding the novelty and baselines remains. The baselines presented in the study are fairly basic machine learning methods, such as K-means and multi-layer perceptron (GeneDisco and DiscoBax). As the authors stated, "our method is essentially a mix of ReAct, reflection, LLM as optimizers, and RAG." This further heightens concerns about the contribution and novelty of the work, as it appears that the paper primarily applies existing techniques (ReAct/reflection/LLM as optimizers/RAG) to a gene perturbation dataset in a multi-turn task setting.
> > Given these concerns, I will maintain my current score.

---

> ### Author Response · Authors · 2024-11-26
> **Response to reviewer kidb**
>
> >The baselines presented in the study are fairly basic machine learning methods
>
> We thank the reviewer for their comments. The baselines that we make use of in this paper were published in [ICLR in 2022](https://openreview.net/forum?id=-w2oomO6qgc) and in [ICML in 2023](https://arxiv.org/abs/2312.04064). These are the state of the art approaches in the field at the moment. If the reviewer has suggestions for new baselines, then we would be happy to include them. In fact, we went even further to include new baselines based on human search and using an agent + Bayesian optimization approach.
>
> > it appears that the paper primarily applies existing techniques (ReAct/reflection/LLM as optimizers/RAG) to a gene perturbation dataset
>
> We are confused by this response because the reviewer's original criticism earlier was that we were not applying ReAct, reflection, LLM as optimizers or RAG approaches. We also perform ablation experiments to compare against simpler approaches using only a subset of these methods. Moreover, the simplicity of our approach is its strength and is also what enables its broad generalization with no direct training.
>
> Lastly, we are happy to hear that we addressed the remaining 3 of the reviewers core comments. To reiterate the reviewer raised concerns about
>
> - **Potential Information Leakage** which we clarified was not the case thanks to the inclusion of an unpublished datasets and further validation using GPT-4o and web search to check that predicted genes were not from the literature
> - **Linear Performance Curve** which we clarified was not the case since the reviewer appeared to have misunderstood our performance metrics
> - **Method Validity** which we showed held true across datasets, tasks and tool use
>
> We hope the reviewer will let us know what specific baselines we can include to address any remaining concerns. In any case, given that the majority of their original concerns have been addressed, we hope they will consider revising their score.

---

### Official Review · Reviewer_MuqZ · 2024-11-04

**Soundness:** 3
**Presentation:** 3
**Contribution:** 2
**Rating:** 5
**Confidence:** 3

**Summary:**

This paper introduces BioDiscoveryAgent,an AI-based tool for automating the design of genetic perturbation experiments. Leveraging large language models (LLMs) and biological knowledge, BioDiscoveryAgent identifies genes for perturbation to induce specific phenotypes, such as cell growth, without requiring custom machine learning models or acquisition functions.
Contribution: 1. BioDiscoveryAgent innovates in closed-loop experimental design by using LLMs to propose genetic perturbations in each experimental round, adapting its selections based on prior results and biological insight. 2. The agent outperforms Bayesian optimization baselines by identifying 21% more relevant genes across multiple datasets and achieves a 46% improvement on the more challenging task of identifying non-essential genes. 3. It incorporates external tools like literature search, code execution, and critique from a secondary agent, enhancing interpretability and decision-making.

**Strengths:**

Strengths of this paper includes:
1. BioDiscoveryAgent introduces a unique approach to closed-loop experimental design, leveraging LLMs to suggest gene perturbations and adapt based on feedback from prior rounds. This paper claims this application of LLMs is novel in experimental biology, particularly for genetic perturbation studies.
2. The agent demonstrates superior performance compared to traditional Bayesian optimization methods. It identify 21% more relevant genes and 46% more non-essential gene hits.
3. BioDiscoveryAgent can draw from diverse information sources with high adaptability and versatility.

**Weaknesses:**

1. Although the model demonstrates strong performance on simulated and existing datasets, the paper lacks validation through real-world, wet-lab experiments.
2. BioDiscoveryAgent’s effectiveness is highly dependent on the underlying LLMs’ pre-existing biological knowledge and reasoning abilities. Limitations in the LLM’s training data or biological understanding could restrict the agent's performance.
3. The use of large-scale LLMs, along with additional tools like literature search and gene databases, results in considerable computational overhead.
4. While the agent performs well overall, the paper observes that the greatest performance improvements occur in the early rounds of experimentation, likely due to the LLM’s pre-existing knowledge. However, as rounds progress, these benefits decline, suggesting potential limitations in its ability to learn effectively from experimental feedback over time.

**Questions:**

See weakness.

---

> ### Author Response · Authors · 2024-11-22
> **1. Agent utility validated through biologically meaningful computational benchmarks**
>
> > Although the model demonstrates strong performance on simulated and existing datasets, the paper lacks validation through real-world, wet-lab experiments.
>
> We thank the reviewer for their feedback and agree that the strongest validation of our approach would come from real-world wet-lab experiments. To this end, we have engaged with collaborators and initiated experimental work, though the timeline for obtaining results is expected to span several months.
>
> The primary aim of our paper is to demonstrate the utility of LLM-powered agents in the computational design of biological experiments. While our current evaluation relies on computational validation, we have shown clear improvements in performance compared to published baseline models and well-known Bayesian optimization benchmarks for experiment design tasks. It is worth noting that the prior works we compare against ([Mehrjou et al., ICLR 2022](https://iclr.cc/virtual/2022/poster/6889); [Lyle et al., ICML 2023](https://arxiv.org/abs/2312.04064)) also relied solely on computational validation for benchmarking. However, we have gone beyond past work to expand the scope and rigor of our validation.
>
> - **Diverse datasets**: This includes integrating new datasets that represent a broader diversity in cell types (T cells, pancreatic cells, neurons, epithelial cells, CAR-T cells), perturbation types (CRISPRi, CRISPRa), and phenotypic responses (cytokine expression, cell viability, cell proliferation).
> - **Unpublished data**: One of these datasets (CAR-T) is unpublished and generated in a wet-lab by one of the authors of this paper.
> - **2-gene setting**: Additionally, we were the first to incorporate a 2-gene perturbation screen based on our interactions with experimental biologists. The 2-gene setting is far more challenging given the larger search space and also more valuable scientifically.
> - **Predicting non-essential genes**: We also extend the evaluation to focus specifically on non-essential genes and observed a significant 46% improvement over baselines. Essential genes are known to show strong phenotypic effects upon perturbation in any experiment. By restricting the analysis to non-essential genes we focused specifically on biology that is directly relevant to the experiment.
> - **Enhanced interpretability**: We present a detailed example of model interpretability ensuring that our predictions are not only accurate but also valuable and actionable in real-world contexts.
> - **Realistic human baseline**: Lastly, we have also included a human baseline that involves a web search followed by regular pathway enrichment analysis to highlight the practical value added by our approach over manual selection and conventional bioinformatics.

---

> ### Author Response · Authors · 2024-11-22
> **2. BioDiscoveryAgent leverages both biological knowledge and experimental observations**
>
> > 2. BioDiscoveryAgent’s effectiveness is highly dependent on the underlying LLMs’ pre-existing biological knowledge and reasoning abilities. Limitations in the LLM’s training data or biological understanding could restrict the agent's performance.
>
> We thank the reviewer for their feedback. While it is true that the agent depends on biological knowledge for performance improvement, this is also its unique strength. Moreover, the agent’s ability to learn from experimental observations limits overdependence on prior knowledge.
>
> - **Biological knowledge is a unique strength of BioDiscoveryAgent**: Compared to baselines, ours is the only approach explicitly leveraging biological knowledge in experiment design tasks. This strength is evidenced by the agent’s strong performance during initial rounds, where prior knowledge plays a critical role. That said, we acknowledge that stronger LLMs with deeper biological understanding would further enhance performance on this task.
>
> - **Prior knowledge used in combination with experimental observations**: The true strength of BioDiscoveryAgent lies in its ability to balance prior knowledge with evidence derived from experimental results. As shown in Figure 3, running BioDiscoveryAgent using only a prompt without access to experimental observations leads to strong initial performance but declining results over time. In contrast, an agent given both the prompt and experimental observations consistently outperforms across all experimental rounds. This demonstrates how the integration of prior knowledge and experimental observations enhances robustness, allowing BioDiscoveryAgent to excel even when prior knowledge is insufficient.
>
> - **Generalizable platform for experiment design that does not need to be explicitly trained**: Furthermore, BioDiscoveryAgent introduces a broadly generalizable paradigm for genetic perturbation experiment design that doesn’t require specific fine-tuning for each dataset. Combined with the extensive interpretability of the agent’s actions at each stage, we would argue that this significantly strengthens the robustness of our proposed approach over existing baseline methods that are trained specifically on individual datasets using mostly black box methods with very little to no prior biological knowledge. This also reduces the risk of overfitting to noise in the experimental data as compared to the other baseline methods that are specifically trained on each dataset.

---

> ### Author Response · Authors · 2024-11-22
> **3. BioDiscoveryAgent has much lower local compute cost than ML baselines**
>
> > 3. The use of large-scale LLMs, along with additional tools like literature search and gene databases, results in considerable computational overhead.
>
> **Tool use incurs minimal cost overhead**: We agree that large-scale LLMs can incur significant computational overhead. However, gene search tools which show the greatest improvements in our ablation studies (Table 3) introduces minimal additional compute since tool use primarily involves simple lookups rather than intensive computations. We conducted an analysis of token usage and API cost for two LLMs (Claude 3.5 Sonnet, Claude 3 Haiku) and included the result in Table 11 of the paper. We also added a section in the main text discussing this analysis. We found that while usage of tools with Claude 3 Haiku increases token usage, it also reduces the average number of trials it takes for the model to produce a valid gene list, so there is only a slight net increase in the token price.
>
> **Generalizable experiment design tool with low computational cost**: The strength of our approach is that it does not require any LLM training or fine-tuning. We use API calls to pre-trained LLMs running on external servers and don’t run any LLM inference locally. In this sense, the agent runs were much cheaper to run computationally than the machine learning baselines which required several GPU-hours overall for training. Moreover, leveraging pre-trained LLMs allows BioDiscoveryAgent to perform well on a wide range of new datasets without requiring dataset-specific tuning or retraining.

---

> ### Author Response · Authors · 2024-11-22
> **4. Agent performance is consistent over longer experiments**
>
> > 4. While the agent performs well overall, the paper observes that the greatest performance improvements occur in the early rounds of experimentation, likely due to the LLM’s pre-existing knowledge. However, as rounds progress, these benefits decline, suggesting potential limitations in its ability to learn effectively from experimental feedback over time.
>
> **Ability to learn from both prior knowledge and experimental observations**: We thank the reviewer for their insightful comment. While it is true that the model’s prior knowledge provides a significant boost in predictive performance during the earlier rounds, experimental readouts continue to contribute to predictive performance in the later stages. In Figure 3, we present an analysis demonstrating the impact of these two distinct information sources on the model’s performance. We conducted three experiments:
> - **Agent with access only to the prompt (no experimental observations)**: This agent exhibited a high hit rate initially, but its performance plateaued quickly.
> - **Agent with access only to experimental observations**: This agent performed poorly at the start but quickly learned from the observations, eventually surpassing the agent with access to only the prompt.
> - **Agent with access to both the prompt and experimental observations**: This agent consistently outperformed both agents limited to just the prompt or the observations.
> These results highlight that the model meaningfully leverages both prior knowledge and experimental observations in its decision-making process.
>
> **Performance remains consistent over extended runs**: In response to the reviewer’s feedback, we conducted additional trials extending up to 30 rounds for each dataset, with batch sizes of 32 and 128. The results, presented in Figure 9, show no strong evidence of a decline in performance (even when total number of predicted genes had reached 3840). From a biological perspective, running a screening experiment over 30 rounds is often impractical due to technical overhead and cost constraints. This is why our analysis primarily focuses on 5–10 rounds with a batch size of approximately 100 genes, which strikes a balance between the utility of the series of experiments and economic viability.

---

> ### Author Response · Authors · 2024-11-27
> **4. Agent outperforms baselines across longer experiments**
>
> We wish to further update this response with experimental results. Based on feedback from the reviewers, we trained over 900 models (6 datasets x 7 baseline models x 10 iterations x 2 batch sizes) across 30 rounds of experimentation. We present the results from our analysis in the new Tables 12, 13, 14 and 15 in the revised manuscript.
>
> | **Model**         | **Schmidt1 (10 rounds)** | **Schmidt1 (20)** | **Schmidt1 (30)** | **Schmidt2 (10)** | **Schmidt2 (20)** | **Schmidt2 (30)** | **CAR-T (10)** | **CAR-T (20)** | **CAR-T (30)** |
> |--------------------|-------------------|-------------------|-------------------|-------------------|-------------------|-------------------|----------------|----------------|----------------|
> | Random            | 0.06             | 0.123            | 0.181            | 0.047            | 0.092            | 0.139            | 0.06           | 0.119          | 0.179          |
> | **Baseline Models**                                                                                                                                    |
> | Soft Uncertain    | 0.061            | 0.123            | 0.192            | 0.049            | 0.100            | 0.148            | 0.054          | 0.127          | 0.186          |
> | Top Uncertain     | 0.077            | 0.146            | 0.216            | 0.068            | 0.137            | 0.200            | 0.074          | 0.139          | 0.204          |
> | Margin            | 0.070            | 0.143            | 0.211            | 0.068            | 0.130            | 0.192            | 0.075          | 0.136          | 0.209          |
> | Coreset           | 0.123            | **0.231**            | **0.303**            | 0.136            | 0.248            | 0.314            | 0.092          | 0.144          | 0.227          |
> | Badge             | 0.079            | 0.153            | 0.226            | 0.082            | 0.154            | 0.217            | 0.064          | 0.144          | 0.208          |
> | K-Means (E)       | 0.066            | 0.130            | 0.198            | 0.072            | 0.131            | 0.194            | 0.052          | 0.097          | 0.161          |
> | K-Means (D)       | 0.069            | 0.140            | 0.203            | 0.086            | 0.172            | 0.235            | 0.060          | 0.125          | 0.189          |
> | **BioDiscoveryAgent (Sonnet)**        | **0.143**        | 0.214            | 0.293            | **0.179**        | **0.286**        | **0.360**        | **0.193**      | **0.264**      | **0.329**      |
>
> **Experiment**: We present the results for hit ratio at round 10, 20 and 30. In each round the agent predicts 128 genes. Results include the ML baseline methods including the coreset which performs best. Due to space constraints we share below the result for 3 datasets. We evaluate the results using the hit ratio over non-essential genes. In Table 14 and 15, we include results for all datasets with error bars.
>
> **Results**:
> - Across all 6 datasets with just one exception, BioDiscoveryAgent outperforms all baselines.
> - As noted correctly by the reviewer, in Schmidt1 (IFNG), we did observe previously that the best ML baseline (coreset) begins to rise around round 5 and indeed it outperforms BioDiscoveryAgent marginally in rounds 20, 30. However, in all other cases, BioDiscoveryAgent more accurately identifies genes relevant to the specific phenotype.
> - In addition, in Table 14 we show that by including the new approach of Agent + coreset, we outperform coreset in every case including for the IFNG dataset. Thus, the agent's predictions are not redundant with those made by the baseline method.
> - To be even more rigorous, we tested longer runs both with a batch size of 128 (Tables 12, 13) and 32 (Tables 14, 15). Smaller batch sizes show larger improvement which reflects better low-data generalization capabilities of the agent.

---

> ### Author Response · Authors · 2024-11-27
> **Summary Response for Reviewer MuqZ**
>
> We thank the reviewer MuqZ for their valuable feedback and evaluation of our work. We are very pleased to see that they appreciate the novelty of our application of LLM based agents to closed loop experiment design in biology. They also acknowledge the versatility of our method and its significant improvement in performance over baselines. We addressed the following comments that MuqZ raised about our work:
>
> - **Biologically relevant validation**: We described how our core contribution was the application of LLMs to closed loop biological experiment design. While we significantly outperform all computational baselines in the literature, we enhance our evaluation by making it more biologically relevant. Through discussions with biologists, we include more diverse datasets, a 2-gene perturbation task, evaluation on non-essential genes (harder than evaluating on all genes), a realistic human baseline as well as making use of our own unpublished data.
>
> - **Reliance on prior knowledge**: BioDiscoveryAgent is definitely dependent on prior knowledge for making its predictions and this is also its core strength and what makes it generalizable without any model training. However, we also show using experiments that its capabilities draw on both its prior knowledge as well information obtained from experimental evidence. The agent that has access to both sources of information outperforms one with either one.
>
> - **Compute requirements in comparison to baseline models**: We include a Table showing how prompting LLMs hosted externally actually lowers the local compute requirements compared to baseline machine learning models that generally need GPUs for training.
>
> - **Consistency in agent performance**: We train 900 additional experiments to show how agent performance remains consistent over longer runs. By evaluating all models at 10, 20 and 30 rounds of experimentation, we show that BioDiscoveryAgent consistently outperforms all other baselines across 5 out of the 6 datasets.
>
> We believe our responses comprehensively address the reviewer’s concerns and hope they will consider raising their evaluation if satisfied with these clarifications. If there are any further questions, we would be grateful if the reviewers could get back soon so we can incorporate the feedback before the submission deadline tomorrow.

---

> > ### Comment · Reviewer_MuqZ · 2024-11-27
> >
> > Thank you for your detailed response, which addressed some of my concerns. However, my major concerns remain:
> >
> > 1. Novelty:
> > The current approach relies heavily on the pretrained LLM to design genetic perturbation experiments, making the LLM a crucial component of the experiment design loop.  Thus I have concern about the significance of this work's novelty, as it builds upon existing LLM capabilities.
> >
> > 2.Robustness
> > The work assumes that the LLM is well-trained and has adequate prior knowledge. However, I worry about scenarios where:
> > The LLM lacks knowledge in certain genetic areas. The LLM was trained on corrupted or biased data, which could negative impact experiment design. How to detect and prevent such biased prior knowledge in LLMs from increasing experimental costs.
> >
> > Due to these concerns, I will maintain my current score.

---

> ### Author Response · Authors · 2024-12-04
> **Response to reviewer (Novelty)**
>
> We thank the reviewer for their comments
>
> 1. **Novelty**
>
> > I have concern about the significance of this work's novelty, as it builds upon existing LLM capabilities.
>
> **Simplicity is a strength of our approach**:
>
> Our approach intentionally emphasizes simplicity to maximize robustness and generalizability. We view this simplicity as a strength, particularly given one of the core advantages of using large language models: the ability to employ intuitive and interpretable natural language-based prompts.
>
> While it is true that our approach builds on existing LLM capabilities, its novelty lies in demonstrating how these models can be effectively repurposed as experiment design agents. Specifically, our method does not rely on direct training with the data it is applied to. Yet, it outperforms other baselines that are specifically trained on this task by approximately 21% and 47% on key metrics, showcasing its capability to generalize without fine-tuning. We believe this result is sufficiently novel and valuable for the community.
>
> **Tool use augments agent performance over LLM**
>
> Additionally, our work goes beyond simply leveraging pretrained LLM capabilities. We establish a tangible pathway to further enhance these agents through the use of tools, which augment the agent’s performance without compromising generalizability. Tools extend the functionality of LLMs, enabling them to account for both existing knowledge and new evidence in a more robust manner. Our paper includes a detailed analysis of tool use across various LLM backbones, highlighting where tools consistently improve performance and where challenges remain. Importantly, we uncover a novel relationship between the size of the LLM and the benefits of tool use, providing a framework for future work in this area.
>
> **Experiment to illustrate novelty over a regular LLM prompt**:
>
> As an experiment to highlight novelty in BioDiscoveryAgent predictions over a regular LLM, we took the set of all hits predicted at round 5 and asked GPT-4o to identify if any of these had been linked with CAR-T cells before.
>
> Out of 33 total hits, it found a direct link for 4 of them, a possible indirect link for 7 of them and for all the remaining 22 hit genes, it said "As of now, there is limited or no direct evidence linking these genes to CAR-T cell research or therapy." We also tried a few Google searches and were unable to make meaningful connections for these genes to previously published papers on CAR-T cells. In case the reviewer wants to run their own experiments, then some of these genes are FOXG1, SDHB, SSRP1, TBX5, TCF12.
> Thus BioDiscoveryAgent effectively integrates prior knowledge from the literature and other source in combination with experimental outcomes to make reasoned predictions.

---

> ### Author Response · Authors · 2024-12-04
> **Response to Reviewer (Robustness)**
>
> 2. **Robustness**
>
> > How to detect and prevent such biased prior knowledge in LLMs from increasing experimental costs.
>
> - Our agent combines both prior knowledge and data gathered from experimental rounds, preventing over-reliance on any single source of information. This balanced approach leads to superior performance compared to cases where only prior knowledge or only experimental data is used (Figure 3).
>
> - The agent provides detailed reasoning for each prediction, enabling scientists to verify the relevance and validity of its outputs. Additionally, when using the literature search tool, the agent supplements its reasoning with citations, including line numbers from referenced papers (Appendix D), ensuring greater transparency and accountability.
>
> - We agree with the reviewer that in the case of systems that are less well studied and where there is less information in the literature, the model does struggle more than in other cases. An example of this could be the Sanchez dataset in our paper which  performs perturbation experiments in neurons, a less well-studied system compared to T cells. Indeed, the model's prior knowledge appears to benefit it less but it still performs comparably to existing baseline models by leveraging information from the experimental data.

---

### Author Response · Authors · 2024-12-04
**Overall Summary**

We thank the reviewers for all their valuable feedback. It has really helped improve our approach and strengthen the rigor of our analysis and evaluation.

**Improvements in revised submission**

In response to reviewer feedback:
- **Results**: We have added 7 entirely new tables of results and 2 figures.
- **Baselines**: We incorporate 2 brand new baselines (Human baseline, AI agent + traditional ML baseline).
- **Experiments**: We ran over 900 new experiments to show consistent performance improvement over 30 rounds of experimentation.
- **Analyses**: We also incorporate new analysis looking at performance ranks of different LLM backbones, analysis of the novelty of agent predictions using GPT-4o and recommendations for tool selection under different LLMs.


**Contributions in our work**

**Approach**
- **First to apply agents to closed loop biological experiment design**: Our application of LLM-based agents to this genetic perturbation scenario, and to closed loop biological experimentation broadly, has not been proposed before (based on our knowledge). We focus on iterative experimental design, aiming to optimize the detection of phenotypic hits across multiple experimental rounds.
- **Simple, intuitive approach**: Our approach intentionally emphasizes simplicity to maximize robustness and generalizability. We view this simplicity as a strength and it is what enables its broad generalization with no direct training on individual datasets.
- **Leveraging both prior knowledge and experimental data**: We show that BioDiscoveryAgent draws on both prior knowledge as well information obtained from each experimental round. The agent that has access to both sources of information outperforms the agent with only access to either one.

**Evaluation**
- **Outperforms all published baselines**: We benchmarked against all formally published models for this task, including the Bayesian optimization baselines from GeneDisco [Mehrjou et al, ICLR 2022](https://iclr.cc/virtual/2022/poster/6889) and DiscoBax [Lyle et al. ICML 2023](https://arxiv.org/abs/2312.04064) that are specifically trained for this task. Our approach demonstrates a 21% improvement in detecting hit genes and a 46% improvement when only considering non-essential genes, a much harder task.
- **Biologically relevant evaluation**: We significantly expand the evaluation framework from past papers with more biologically relevant tasks, metrics and datasets. We include a broader diversity in cell types (T cells, pancreatic cells, neurons, epithelial cells, CAR-T cells), perturbation types (CRISPRi, CRISPRa), and phenotypic responses (cytokine expression, cell viability, cell proliferation).
- **Unpublished data**: To enhance the rigor of our validation, we include an unpublished dataset (CAR-T) that is generated in a wet-lab by one of the authors of this paper.
- **Introduce 2-gene setting**: We introduce closed-loop experiment design for 2-gene perturbation settings. This presents a significantly larger search space and greater scientific value.
- **Predicting non-essential genes**: We further extend the evaluation to focus specifically on non-essential genes (that are more biologically relevant) and observed a significant 46% improvement over baselines.
- **Realistic human baseline**: We have also included a human baseline that involves a web search followed by regular pathway enrichment analysis to highlight the practical value added by our approach over manual selection and conventional bioinformatics.

**Capabilities and Enhancements**

- **Enhanced interpretability**: BioDiscoveryAgent's predictions are highly interpretable at every stage, which is not a feature in any of the baseline models. We include analysis in the rebuttal to show that model reasoning is not random and aligns with known biology.
- **Suite of tools to expand agent's capabilities**: We design three new classes of tools that impact model performance: literature search for referencing and citing relevant literature, gene search for accessing information from tabular datasets and agent critique.
- **Novel relationship between benefit of tool use and size of LLM**: We present a new finding on how tool use significantly improved model performance for smaller LLMs but showed limited benefit in the case of larger LLMs.

**Implications**

Our work offers some of the first compelling evidence that language models can significantly benefit researchers in the design of biological experiments. We emphasize both the strengths of an agent-based approach and its current limitations. Notably, our experiments suggest that tool use holds the greatest potential for further augmenting LLM-based agents while preserving model generalization. Furthermore, there is considerable promise in expanding the use of machine learning models as tools, as demonstrated by our new Sonnet+Coreset baseline. However, these tools must be better designed to account for the advanced capabilities of larger models.

---

### Meta-Review · Area_Chair_1fjC · 2024-12-21

**Metareview:**

The paper introduces BioDiscoveryAgent, an AI-based tool leveraging large language models (LLMs) to design genetic perturbation experiments, "reason" about their outcomes, and compare and contrast hypotheses explaining them. The agent identifies genes for perturbation to induce specific phenotypes, such as cell growth, without requiring custom machine learning models or acquisition functions. The authors demonstrate BioDiscoveryAgent’s effectiveness across multiple datasets, including a novel one introduced here, showing significant improvements in predicting relevant genetic perturbations compared to existing Bayesian optimization baselines.

Strengths:
1. **Empirical Performance**: The main selling point of the paper is the effectiveness of the method; BioDiscoveryAgent shows substantial improvements over baselines across multiple settings.
2. **Comprehensive Evaluation**: The paper includes extensive experiments, covering multiple datasets (seen and unseen), different LLM backbones, and various tool utilizations.
3. **Presentation**: The writing is clear and easy to follow, the results are well presented.
4. **System Interpretability***: The agent provides detailed reasoning trace for its predictions, enhancing transparency and aiding in the validation of its outputs. However, whether those explanations are truly faithful to the model's internal mechanisms is unclear (and an open problem, as pointed out by reviewer  biwg).


Weaknesses:
1. **Limited Novelty in Methodology**: While the specific application and setting is novel, the underlying methods, such as using LLMs and tools, is certainly not new, with a very rich literature in the past year.
2. **Dependence on underlying LLM**: The agent’s performance heavily relies on the pre-existing knowledge and reasoning capabilities of the LLMs, which may limit its effectiveness in less-studied areas, obfuscates its operation (compared to alternative non-LLM bio methods), and makes it susceptible to known failure modes of LLMs (hallucinations, lack of robustness to prompt variation, etc).
3. **Unclear Interpretation**: Certain aspects of the method, design choices, and trends in results are missing explanation or discussion.
4. **No Wet-lab Validation**: The paper lacks real-world, wet-lab validation of the agent’s predictions, relying solely on computational benchmarks.

During the rebuttal period, the authors made significant efforts to address the reviewers’ concerns, including running additional experiments, clearing up many misunderstandings, providing thoughtful additional intuition, and improving the clarity of their presentation. Their diligent and meticulous rebuttal prompted two reviewers to increase their scores, and greatly mitigated weakness 3. Although weakness 4 is important, I do not consider it a deal-breaker considering the costs associated with wet-lab validation and the scope of this conference.

Overall, this paper addresses an important problem and provides a promising direction for the use of AI agents in scientific discovery. While some fundamental issues regarding novelty and generalizability remained, the overall positive feedback and the substantial empirical gains justify the decision to accept the paper, with the understanding that the additional results and discussion generated during the rebuttal will be included in the final version of the paper.

**Additional Comments On Reviewer Discussion:**

This paper generated a remarkably active discussion phase, with a staggering 52 forum replies exchanged between authors and reviewers. While these discussions are too long to describe here in detail, the main points and counterpoints were:

* Reviewer MuqZ raised concerns about the novelty and robustness of the approach. The authors addressed these by highlighting the simplicity and generalizability of their method and providing additional experiments to show consistent performance.
* Reviewer kidb questioned the baseline comparisons and potential information leakage. The authors clarified the inclusion of unpublished data and the complexity of the task, and provided additional human baseline comparisons.
* Reviewer qGCq suggested the inclusion of more baselines and raised concerns about the number of iterations. The authors conducted extensive additional experiments and demonstrated consistent performance improvements, which prompted the reviewer to increase their score three times, to 5, 6 and eventually 8.
* Reviewer WoKr highlighted the need for more refined methods and explanations for tool use. The authors provided detailed analyses on the benefits of tool use and the role of the critique agent, and implemented a hybrid approach combining the agent with traditional ML methods. The response prompted the reviewer to increase their score as well.
* Reviewer biwg appreciated the novelty and thoroughness of the work but suggested discussing the risks of interpretability claims. The authors acknowledged these risks and provided additional analyses to support their claims.

While I understand the desire to have wet-lab validation, I weighed those concerns less heavily for the reasons described above. I weighted the leakage concerns highly, but the authors' response to that matter seems to have mostly addressed that concern (as we all know, data leakage is incredibly hard to prove/disprove in any setting involving LLMs).

---

### Decision · Program_Chairs · 2025-01-22

Accept (Poster)